# Understanding the robustness difference between stochastic gradient descent and adaptive gradient methods

**Avery Ma**                                           *ama@cs.toronto.edu*
*University of Toronto*
*Vector Institute*

**Yangchen Pan**                                       *yangchen.pan@eng.ox.ac.uk*
*University of Oxford*

**Amir-massoud Farahmand**                             *farahmand@cs.toronto.edu*
*University of Toronto*
*Vector Institute*

**Reviewed on OpenReview:** *https://openreview.net/forum?id=ed8SkMdYFT*

## Abstract

Stochastic gradient descent (SGD) and adaptive gradient methods, such as Adam and RMSProp, have been widely used in training deep neural networks. We empirically show that while the difference between the standard generalization performance of models trained using these methods is small, those trained using SGD exhibit far greater robustness under input perturbations. Notably, our investigation demonstrates the presence of irrelevant frequencies in natural datasets, where alterations do not affect models' generalization performance. However, models trained with adaptive methods show sensitivity to these changes, suggesting that their use of irrelevant frequencies can lead to solutions sensitive to perturbations. To better understand this difference, we study the learning dynamics of gradient descent (GD) and sign gradient descent (signGD) on a synthetic dataset that mirrors natural signals. With a three-dimensional input space, the models optimized with GD and signGD have standard risks close to zero but vary in their adversarial risks. Our result shows that linear models' robustness to $\ell_2$-norm bounded changes is inversely proportional to the model parameters' weight norm: a smaller weight norm implies better robustness. In the context of deep learning, our experiments show that SGD-trained neural networks have smaller Lipschitz constants, explaining the better robustness to input perturbations than those trained with adaptive gradient methods. Our source code is available at https://github.com/averyma/opt-robust.

## 1 Introduction

Adaptive gradient methods, such as Adam (Kingma & Ba, 2015) and RMSProp (Hinton et al., 2012), are a family of popular techniques to optimize machine learning (ML) algorithms. They are an extension of the traditional gradient descent method, which uses the gradient of a differentiable objective function to update the model's parameters in the direction that improves the objective. To speed up the optimization procedure, the adaptive gradient methods introduce a coordinate-wise learning rate to adjust the update for each parameter based on its individual gradient. Previous empirical work investigates the difference in the standard generalization between models trained using SGD and adaptive gradient methods (Wilson et al., 2017; Agarwal et al., 2020), while recent efforts have focused on understanding the implicit bias of SGD (Gunasekar et al., 2017; Soudry et al., 2018; Lyu & Li, 2020) and adaptive gradient algorithms (Qian & Qian, 2019; Wang et al., 2021).

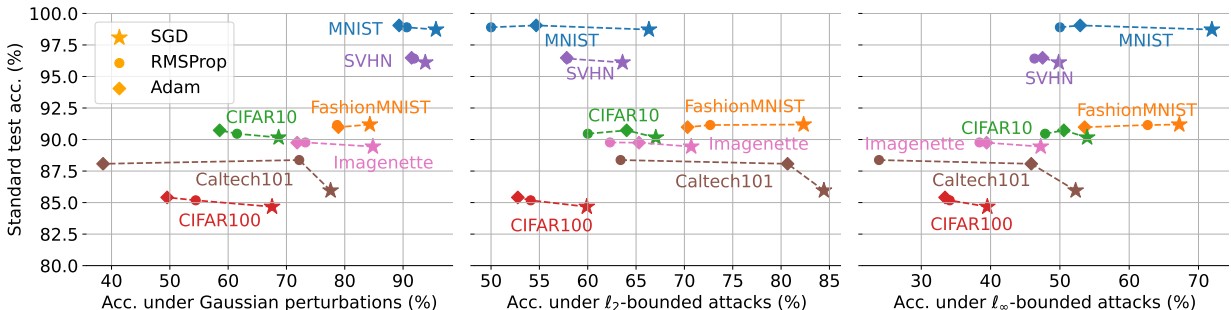

Figure 1: **Comparison between models trained using SGD, Adam, and RMSProp across seven benchmark datasets.** Each colored triplet denotes models on the same dataset. Models trained by different algorithms have similar standard generalization performance, but there is a distinct robustness difference as measured by the test data accuracy under Gaussian noise, $\ell_2$ and $\ell_\infty$ bounded adversarial perturbations (Croce & Hein, 2020). Results are averaged over three independent model initializations and trainings.

Nevertheless, our result shows that in practice such a gap in the standard generalization is relatively small, in contrast to the difference between the robustness of models trained using those algorithms. While more ML-based systems are deployed in the real world, the models' robustness, their ability to maintain their performance when faced with noisy or corrupted inputs, has become an important criterion. There is a large volume of literature on developing specialized methods to improve the robustness of neural networks (Silva & Najafirad, 2020), yet practitioners still simply use standard methods to train their models. In fact, a recent survey shows that only 3 of the 28 organizations have developed their ML-based systems with the improvement in robustness in mind (Kumar et al., 2020). Therefore, this motivates us to understand the effect of optimizers on the robustness of models obtained in the standard training regime. In particular, we focus on models trained using SGD and adaptive gradient methods. Note that our primary focus lies in understanding the robustness difference, and robustification falls outside the scope of our work.

## 1.1 The Robustness Difference between Models Trained by Different Algorithms

As a first step, we compare how models, trained with SGD, Adam, and RMSProp, differ in their standard generalization and robustness on seven benchmark datasets (LeCun, 1998; Xiao et al., 2017; Krizhevsky & Hinton, 2009; Netzer et al., 2011; Howard; Fei-Fei et al., 2004). In our experiments, we evaluate standard generalization using the accuracy of the trained classifier on the original test dataset. To measure robustness, we consider the classification accuracy on the test dataset perturbed by Gaussian noise, $\ell_2$ and $\ell_\infty$ bounded adversarial perturbations (Croce & Hein, 2020). We follow the default Pytorch configuration to train all the models and sweep through a wide range of learning rates. The final model is selected with the highest validation accuracy. Models are trained in a vanilla setting in which data augmentations are limited to random flipping. Additional discussions on batch normalization (Ioffe & Szegedy, 2015), data augmentation, optimization schedules, and network designs are detailed in Appendix B. Appendix C presents comprehensive results from the experiment depicted in Figure 1, including the approach for selecting perturbations for each dataset. While our primary experiments are centered around models based on convolutional neural networks, within the computer vision domain, we also extend our analysis to include results from experiments on Vision Transformers (Dosovitskiy et al., 2021) and an audio dataset (Warden, 2018). The results of these additional experiments are consistent with the findings presented in this section and are detailed in Appendix C. Additionally, visualizations of the perturbations can be found in Appendix G.

Figure 1 compares the models trained with SGD and the adaptive gradient methods, pointing to two important observations. First, the relatively small vertical differences among the three models, on a given dataset, show that the models have similar standard generalization performance despite being trained by different algorithms. On the other hand, we observe, under all three types of perturbations, a large horizontal span with SGD always positioned on the far right side among the three. This indicates that models trained by SGD significantly outperform models trained by the other two in terms of their robustness against perturbations.

## 1.2 Contributions

Previous optimization work often studies how the structure of the dataset affects the dynamics of learning. For example, some focus on a dataset with different feature strengths (Amari et al., 2021; Pezeshki et al., 2021), while others assume a linearly separable dataset (Wilson et al., 2017; Gunasekar et al., 2017; Soudry et al., 2018). In our work, we investigate how the frequency characteristics of the dataset impact the robustness of models trained by SGD and adaptive gradient methods. We make the following contributions:

- We demonstrate that natural datasets contain irrelevant frequencies, which, when removed, have negligible effects on standard generalization performance (Sec. 3.1).

- We also observe that neural networks trained by different algorithms can have very different robustness against perturbations in the direction of the irrelevant frequencies (Sec. 3.2).

- Those observations lead to our claim that models only need to learn how to correctly use relevant information in the data to optimize the training objective, and because their use of the irrelevant information is under-constrained, it can lead to solutions sensitive to perturbations (Sec. 3).

- Our analysis of linear models on least square regression shows that linear models' robustness to $\ell_2$-norm bounded changes is inversely proportional to the model parameters' weight norm: a smaller weight norm implies better robustness (Sec. 4.1).

- We study the learning dynamics of GD and signGD, a memory-free version of Adam and RMSProp, with linear models. With a three-dimensional input space, the analysis shows that models optimized with GD exhibit a smaller weight norm compared to their signGD counterparts (Sec. 4.2).

- To generalize this result in the deep learning setting, we demonstrate that neural networks trained by Adam and RMSProp often have a larger Lispchitz constant and, consequently, are more prone to perturbations (Sec. 5).

Specifically, in the analysis of linear models, we design a least square regression task using a synthetic dataset whose frequency representation mimics the natural datasets. This setting allows us to i) mathematically define the standard and adversarial population risks, ii) design a learning task that has multiple optima for the standard population risk, each with a different adversarial risk, and iii) theoretically analyze the learning dynamics of various algorithms.

## 2 Background

In this section, we briefly review the essential background to help understand our work. Specifically, we discuss formulations of adaptive gradient methods, previous work on the adversarial robustness of the model, and methods of representing signals in the frequency domain.

### 2.1 Optimizations with Adaptive Gradient Algorithms

Consider the empirical risk minimization problem with an objective of the form $\mathcal{L}(w) = \frac{1}{N}\sum_{n=1}^{N} \ell(X_n, Y_n; w)$, where $w \in \mathbb{R}^d$ is a vector of weights of a model, $\{(X_n, Y_n)\}_{n=1}^{N}$ is the training dataset and $\ell(x, y; w)$ is the point-wise loss quantifying the performance of the model on data point $(x, y)$. A common approach in training machine learning models is to reduce the loss via SGD which iteratively updates the model based on a mini-batch of data points drawn uniformly and independently from the training set:

$$g(w) = \frac{1}{|\mathcal{B}|}\sum_{n \in \mathcal{B}} \nabla_w \ell(X_n, Y_n; w), \tag{1}$$

where $\mathcal{B} \subset \{1, ..., N\}$ denotes the minibatch and has a size of $|\mathcal{B}| \ll N$. The update rule of SGD is $w(t + 1) = w(t) - \eta(t)g(w(t))$, where $\eta(t) \in \mathbb{R}^+$ denotes the learning rate.[1]

---

[1]From this point forward, subscripts denote vector/matrix coordinates, and numbers in parentheses denote update iterations unless otherwise specified.

A family of adaptive gradient methods has been used to accelerate training by updating the model parameters based on a coordinate-wise scaling of the original gradients. Methods such as Adam and RMSprop have demonstrated significant acceleration in training deep neural networks (Wilson et al., 2017). Many adaptive gradient methods can be written as

$$
\begin{aligned}
m(t+1) &= \beta_1 g(w(t)) + (1 - \beta_1) m(t) \\
v(t+1) &= \beta_2 g(w(t))^2 + (1 - \beta_2) v(t) \\
w(t+1) &= w(t) - \eta(t) \frac{m(t+1)}{\sqrt{v(t+1)} + \epsilon},
\end{aligned}
\tag{2}
$$

where $g(w(t))$ is the stochastic estimate of gradient used by SGD (1), $m$ and $v$ are the first and second-order memory terms with their strength specified by $\beta_1$ and $\beta_2$, and $\epsilon$ is a small constant used to avoid division-by-zero. Such a general form has been widely used to study the dynamics of adaptive gradient algorithm (Wilson et al., 2017; da Silva & Gazeau, 2020; Ma et al., 2022b). For example, Adam corresponds to $\beta_1, \beta_2 \in (0, 1)$, and RMSProp is recovered when $\beta_1 = 1$ and $\beta_2 \in (0, 1)$. Notice that such updates rely on the history of past gradients, and this makes the precise understanding and analysis of adaptive gradient methods more challenging (Duchi et al., 2011). Recent work analyzes the learning dynamics of adaptive gradient methods by separately considering the direction and the magnitude of the update (Kingma & Ba, 2015; Balles & Hennig, 2018; Ma et al., 2022b). As a simple example, to demonstrate how adaptive gradient methods can potentially accelerate learning compared to the vanilla SGD, consider a memory-free version of (2) with $\beta_1 = \beta_2 = 1$ and $\epsilon = 0$. It is easy to see that the update rule in (2) becomes sign gradient descent:

$$
\begin{aligned}
w(t+1) &= w(t) - \eta(t) \operatorname{sign}(g(w(t))) \\
&= w(t) - \vec{\eta}(t) \odot g(w(t)),
\end{aligned}
\tag{3}
$$

where $\odot$ denotes Hadamard product, $\vec{\eta}(t) \in \mathbb{R}^d$ is a coordinate-wise learning rate based on the absolute value of the weight, i.e., $\vec{\eta}(t) = \frac{\eta(t)}{|g(w(t))|}$. Therefore, $\vec{\eta}(t)$ accounts for the magnitude of the weight and a larger learning rate is used for parameters with smaller gradients.

In general, gradient-sign-based optimization methods are not successful in training deep learning models (Riedmiller & Braun, 1993; Ma et al., 2022b), nevertheless, methods such as signGD can shed light on the learning dynamics of adaptive gradient methods (Karimi et al., 2016; Balles & Hennig, 2018; Moulay et al., 2019). For example, recent work by Ma et al. (2022b) studies the behavior of adaptive gradient algorithms in the continuous-time limit. They demonstrate that under a fixed $\beta_1$ and $\beta_2$, the memory effect for both Adam and RMSprop diminishes and the continuous-time limit of the two algorithms follows the dynamics of signGD flow. In this work, the deep learning models on which we observe the robustness difference are trained using Adam and RMSProp, with the exception of Sec. 4, where we focus on signGD, a memory-free version of Adam and RMSProp, and gradient descent to help us understand the robustness gap between models trained using SGD and adaptive gradient methods in a simple setting.

## 2.2 Robustness of ML Models

An important assumption of most modern ML models is that samples from the training and testing dataset are independent and identically distributed (i.i.d.); however, samples collected in the real world rarely come from an identical distribution as the training data, as they are often subject to noise. It is known that ML models can achieve impressive success on the original testing dataset, but exhibit a sharp drop in performance under perturbations (Szegedy et al., 2014). Such an observation has posed concerns about the potential vulnerabilities for real-world ML applications such as healthcare (Qayyum et al., 2020), autonomous driving (Deng et al., 2020) and audio systems (Li et al., 2020). Models' robustness performance has become an important secondary metric in the empirical evaluation of new training methods, such as data augmentations (Zhang et al., 2018; Hendrycks et al., 2020; Verma et al., 2019; Ma et al., 2022a) and robust optimization techniques (Zhai et al., 2021). Generally, the robustness property of models is assessed by examining the model performance under multiple perturbations (Ding et al., 2020; Shen et al., 2021; Kuang et al., 2018). A wide variety of approaches have been proposed to improve the robustness of the model through regularizations

(Goodfellow et al., 2015; Simon-Gabriel et al., 2019; Wen et al., 2020; Ma et al., 2020; Foret et al., 2021; Wei et al., 2023), data augmentation (Madry et al., 2018; Rebuffi et al., 2021; Gowal et al., 2021; Ma et al., 2022a), and novel network architectures (Wu et al., 2021; Ma et al., 2021; Huang et al., 2021). However, most industry practitioners are yet to come to terms with improving security in developing ML systems (Kumar et al., 2020). Since SGD, Adam, and RMSProp have been the go-to optimizer in both academic and industrial settings, this motivates us to understand the robustness of models trained by them and built on the standard training pipelines, i.e., minimizing some losses on the original training set.

## 2.3 Frequency Representation of Signals

Natural signals are highly structured (Schwartz & Simoncelli, 2001). They often consist of statistically significant (or insignificant) patterns with a large amount of predictability (or redundancy). Such a phenomenon has been observed in both natural images (Ruderman, 1994; Simoncelli, 1997; Huang & Mumford, 1999) and natural audio signals (McAulay & Quatieri, 1986; Attias & Schreiner, 1996; Turner, 2010). To understand the structure of signals and identify patterns from them, one technique is to decompose the signal into multiples of "harmonics" or "overtones": a superposition of periodic waves with varying amplitudes and in varying phases. For example, Fourier (1822) first proposed to analyze complicated heat equations using well-understood trigonometric functions, a method now called the Fourier transformation. This new representation allows us to precisely study the structure and the magnitude of any repeating patterns presented in the original waveform. For the understanding of digital signals, such a process is called discrete-time signal processing (Oppenheim et al., 2001).

Many discrete harmonic transformations exist, such as the discrete Fourier transform, the discrete cosine transform (DCT) (Ahmed et al., 1974) and the wavelet transform (Mallat, 1999). The analysis in this work utilizes the type-II DCT, but other techniques can be applied as well and we expect similar results. Concretely, consider a $d$-dimensional signal $x \in \mathbb{R}^d$ in the spatial domain. The same signal can be alternatively represented as a discrete sum of amplitudes multiplied by its cosine harmonics: $\tilde{x}_k = \sum_{j=0}^{d-1} x_j \cos\left[\frac{\pi}{d}\left(j + \frac{1}{2}\right)k\right]$ for $k = 0, ..., d-1$, where the transformed signal $\tilde{x}$ has a frequency-domain representation.[2] Because DCT is linear, it can be carried out using a matrix operation, i.e., $\tilde{x} = Cx$, where $C$ is a $d \times d$ DCT transformation matrix with values specified by

$$C_{kj}^{(d)} = \sqrt{\frac{\alpha_k}{d}} \cos\left[\frac{\pi}{d}\left(j + \frac{1}{2}\right)k\right], \tag{4}$$

where $\alpha_0 = 1$ and $\alpha_k = 2$ for $k > 0$. In particular, $\tilde{x}$ can be written as a matrix-vector product between the transformation matrix $C$ and the column vector $x$:

$$\begin{bmatrix} \tilde{x}_0 \\ \tilde{x}_1 \\ \vdots \\ \tilde{x}_{d-1} \end{bmatrix} = \begin{bmatrix} \sqrt{\frac{1}{d}} & \sqrt{\frac{1}{d}} & \cdots & \sqrt{\frac{1}{d}} \\ \sqrt{\frac{2}{d}}\cos\frac{\pi(2(0)+1)(1)}{2d} & \sqrt{\frac{2}{d}}\cos\frac{\pi(2(1)+1)(1)}{2d} & \cdots & \sqrt{\frac{2}{d}}\cos\frac{\pi(2(d-1)+1)(1)}{2d} \\ \vdots & \vdots & \vdots & \vdots \\ \sqrt{\frac{2}{d}}\cos\frac{\pi(2(0)+1)(d-1)}{2d} & \sqrt{\frac{2}{d}}\cos\frac{\pi(2(1)+1)(d-1)}{2d} & \cdots & \sqrt{\frac{2}{d}}\cos\frac{\pi(2(d-1)+1)(d-1)}{2d} \end{bmatrix} \begin{bmatrix} x_0 \\ x_1 \\ \vdots \\ x_{d-1} \end{bmatrix}. \tag{5}$$

Notice that $C$ is a real orthogonal matrix whose rows consists of periodic cosine bases with increasing frequencies. Therefore, the absolute value of $\tilde{x}$ at a particular dimension indicates the magnitudes of the corresponding basis function, and a higher dimension in $\tilde{x}$ means the basis function is of higher frequency. Another important property of DCT is its invertibility. That is, signals in the frequency domain can be converted back to the spatial-temporal domain via the inverse DCT (iDCT): $x = C^{-1}\tilde{x} = C^\top \tilde{x}$. In the example above, we discussed one-dimensional DCT which is applied to vectors and is used in the linear analysis in Sec. 4. Transformations on images require two-dimensional DCT and can be done using $\tilde{x} = CxC^\top$, where $x, \tilde{x} \in \mathbb{R}^{d \times d}$, and $C$ is defined in (4); and the inverse two-dimensional DCT is $x = C^\top \tilde{x} C$. For more details on two-dimensional DCT, we refer the reader to Pennebaker & Mitchell (1992).

Previous work analyzes the sensitivity of neural network classifiers by examining the frequency characteristics of various types of perturbations, with an emphasis on understanding how data augmentation affects the

---

[2]Indices range from 0 to $d - 1$, as zero-frequency is commonly used to refer to a signal with a constant everywhere.

robustness of the model (Yin et al., 2019). In our work, the frequency interpretation of signals is an integral part of understanding the robustness difference between models trained by SGD and adaptive gradient methods. This perspective allows us to study the structure of complex signals using well-understood periodic basis functions such as cosines and understand the energy distribution of signals by examining the amplitude of the basis function. In particular, the energy of a discrete signal $x$ is defined as $E(x) = \sum_{i=0}^{d-1} |x_i|^2$, and by Parseval's theorem, is equivalent to the sum of squared amplitudes across all the bases, i.e., $E(x) = E(\tilde{x}) = \sum_{i=0}^{d-1} |\tilde{x}_i|^2$. Natural images are primarily made of low-frequency signals[3]: a high concentration of energy in the low-frequency harmonics renders the amplitude of the higher-frequency harmonics almost negligible (Tolhurst et al., 1992; Schaaf & Hateren, 1996), as shown in Figure 8 in Appendix G. Moreover, we show in Sec. 3.1 that there exist frequencies in natural datasets, which if removed from the training data, do not affect the standard generalization performance of the model. Based on this observation, in Sec. 4, we construct a synthetic dataset that mimics the characteristics of natural signals, and it allows us to study the learning dynamics of various optimization algorithms in a controlled setting.

## 3 A Claim on How Models Use Irrelevant Frequencies

Why do models trained by different optimization algorithms behave similarly in the standard setting where the training and the test inputs are i.i.d., while they perform drastically differently when faced with noisy or corrupted data? To answer this question, we first observe that there is irrelevant information in the natural dataset (Observation I), and attenuating them from the training input has negligible effects on the standard generalization of the model. This leads to our claim:

**Claim 3.1.** *To optimize the standard training objective, models only need to learn how to correctly use relevant information in the data. Their use of irrelevant information in the data, however, is under-constrained and can lead to solutions sensitive to perturbations.*

Because of this, by targeting the perturbations toward the subset of the signal that contains irrelevant information, we notice that models trained by different algorithms exhibit very different performance changes (Observation II).

### 3.1 Observation I: Irrelevant Frequencies in Natural Signals

Previous work demonstrated that the magnitude of the frequency components in natural images decreases as the frequency increases, and this decrease follows a $\frac{1}{f^2}$ relationship (Ruderman, 1994; Wainwright & Simoncelli, 1999). In Figure 8 of Appendix G, we make the observation on several common vision datasets that the distribution of spectral energy heavily concentrates at the low end of the frequency spectrum and decays quickly towards higher frequencies. The spectral sensitivity of the human eyes is limited (Gross, 2005), so patterns with low magnitudes and high frequencies are not important from the perspective of human observers, as they appear to us as nearly invisible and unintuitive information in the scene (Schwartz & Simoncelli, 2001; Schwartz, 2004). For machines, image-processing methods have long exploited the fact that most of the content-defining information in natural images is represented in low frequencies, and the high-frequency signal is redundant, irrelevant, and is often associated with noise (Wallace, 1991; Guo et al., 2020; Sharma et al., 2019).

Similarly, the notion of irrelevant frequencies also exists when training a neural network classifier. One way to illustrate this is by taking a supervised learning task, removing the irrelevant information from the training input, and then assessing the model's performance using the original test data. We observe that when modifying the training dataset by removing subsets of the signal with low spectral energy (Figure 2a) or high frequencies (Figure 2b), there is a negligible effect on models' classification accuracy on the original test data. In Appendix D, we explain how images are modified in detail, and visualizations of the modified images are included in Appendix G. In both settings after reducing more than half of the DCT basis vectors to zeros in the training data, the model's generalization ability remains strong. This observation suggests there is a considerable amount of irrelevant information in naturally occurring data from the perspective of

---

[3]We will always use the term "high" or "low" frequency on a relative scale.

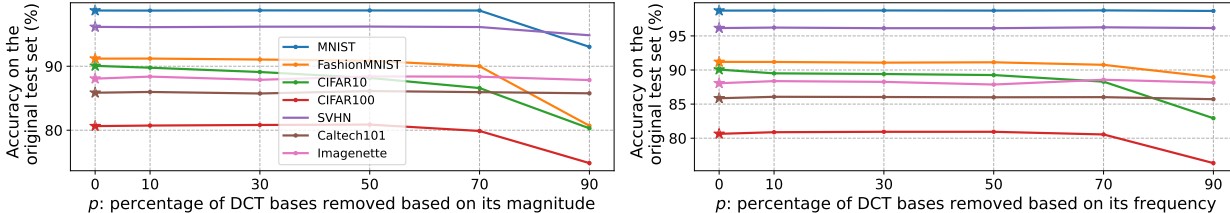

a. Parts of the signal with low spectral energy is irrelevant.  b. Parts of the signal with high-frequency basis is irrelevant.

Figure 2: **Irrelevant frequencies exist in the natural data.** Accuracy on the original test set remains high when the training inputs are modified by removing parts of the signal with a) low spectrum energy and b) high frequencies. Stars represent test accuracy on models trained using the original training input. In setting a), training images are filtered based on the magnitude of the DCT basis. Specifically, parts of the image with DCT bases that have a magnitude in the bottom $\frac{p}{100}$-th percentile are removed, so a large $p$ means more information is discarded. In setting b), training images are low-pass filtered, and $p$ denotes the percentage of the high-frequency components that are discarded in the training data. We explain the formulation of the two settings in Appendix D. Examples of the modified inputs are included in Appendix G.

a neural network classifier, and such information is often featured with low spectrum energy or lives at the high end of the frequency spectrum.

This observation leads to the first part of Claim 3.1. That is, models only need to learn how to correctly use the crucial class-defining information from the training data to optimize the training objective. On the other hand, the extent to which they utilize irrelevant information in the data is not well-regulated. This can be problematic and lead to solutions sensitive to perturbations. In Sec. 4, we validate Claim 3.1 using a linear regression analysis with a synthetic dataset that contains irrelevant information. We demonstrate there exist multiple optima of the training objective and those solutions can all correctly use the relevant information in the data, but the way they exclude irrelevant information from computing the output is different. Specifically, a robust model disregards irrelevant information by assigning a weight of zero to it, but a non-robust model has certain non-zero weights which, when combined with the irrelevant information in the input, yield a net-zero effect in the output. In this case, although the two models are indistinguishable under the original training objective, the non-robust model will experience a reduction in model performance should this irrelevant information become corrupted at test time.

### 3.2 Observation II: Model Robustness along Irrelevant Frequencies

Let us now focus on the second part of the claim. If models' responses to perturbations along the irrelevant frequencies explain their robustness difference, then we should expect a similar accuracy drop between models when perturbations are along relevant frequencies, but a much larger accuracy drop on less robust models when test inputs are perturbed along irrelevant frequencies. Consider the robustness of the models when the test data are corrupted with Gaussian noise: the perturbation along each spatial dimension is i.i.d and drawn from a zero-mean Gaussian distribution with finite variance. This type of noise is commonly referred to as the additive white Gaussian noise, where white refers to the property that the noise has uniform power across the frequency spectrum (Diebold, 1998). Nevertheless, the previous discussion suggests that noise along different frequencies does not have an equal impact on the models' output. To verify this, we assess the impact on model accuracy by perturbing only specific frequency ranges of the test inputs with band-limited Gaussian noise.

To construct the band-limited Gaussian noise, we first follow the previous work (Wang et al., 2020) to group DCT basis vectors based on their distance to the 0-frequency DC term and divide the entire DCT spectrum into ten bands where each band occupies the same number of DCT bases. This is to ensure an identical $\ell_2$ norm among all the perturbations. Denote the binary mask of the $i$-th band by using $M^{(i)} \in \{0, 1\}^{d \times d}$, its

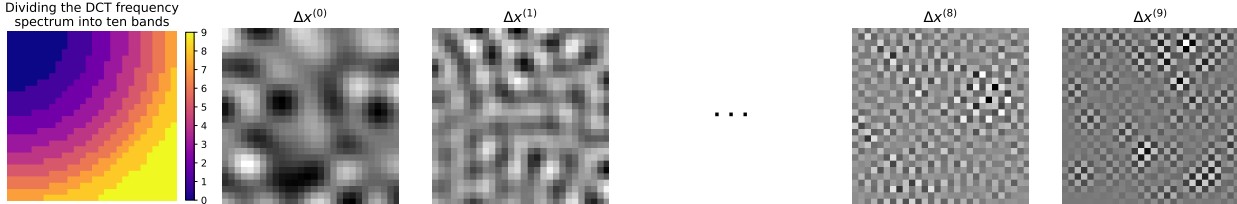

Figure 3: **Visualization of the band-limited Gaussian perturbations.** The DCT spectrum is divided into ten equally sized bands to generate band-limited Gaussian perturbations. Denote them by using $\Delta x^{(i)}$, where $i \in \{0, 1, ..., 9\}$. The frequency represented in the spectrum plot increases from the top-left (lowest frequency) to the bottom-right corner (highest frequency). Therefore, as the band moves towards higher frequencies, perturbations exhibit more high-frequency checkerboard patterns.

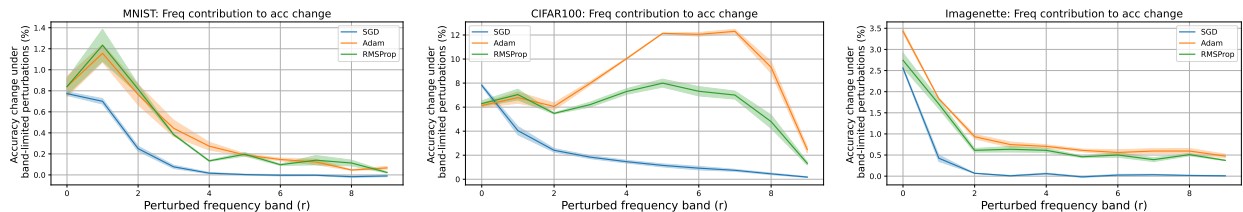

Figure 4: **The effect of band-limited Gaussian perturbations on the model.** Perturbations from the lowest band, i.e., $\Delta x^{(0)}$, have a similar effect on all the models, despite being trained by different algorithms and exhibiting different robustness properties. On the other hand, models' responses vary significantly when the perturbation focuses on higher frequency bands. The results are averaged over three independently initialized and trained models, and the shaded area indicates the standard error among the three models.

corresponding band-limited Gaussian noise is $\Delta x^{(i)} = C^\top (M^{(i)} \odot \delta)C$, where $\delta \sim \mathcal{N}(0, \sigma^2 I_{d \times d})$ and $C$ is the DCT transformation matrix defined in (4). Figure 3 illustrates how the frequency bases are grouped into ten equally sized bands and examples of the band-limited Gaussian noise. Denote the perturbations by using $\Delta x^{(i)}$, with $\Delta x^{(0)}$ and $\Delta x^{(9)}$ representing the lowest and the highest band, respectively. To investigate the effect of the perturbation $\Delta x^{(i)}$ on the models, we measure the change in classification accuracy when the test inputs are perturbed by $\Delta x^{(i)}$:

$$\frac{1}{N} \sum_{n=1}^{N} \mathbb{I}\left\{F(X_n) = Y_n\right\} - \frac{1}{NK} \sum_{n=1}^{N} \sum_{k=1}^{K} \mathbb{I}\left\{F(X_n + \Delta x_k^{(i)}) = Y_n\right\}, \tag{6}$$

where $F$ is a neural network classifier, $\{(X_n, Y_n)\}_{n=1}^{N}$ represents the test dataset, each test input is perturbed by i.i.d sampled $\Delta x_k^{(i)}$ and the subscript $k$ is used to differentiate between $K$ instances of the randomly sampled noise; and we use $K = 10$ in our experiments. It is important to realize in (6) that the additive noise $\Delta x$ is applied to the spatial signal $X$, although we are limiting the frequency band of the noise.

Figure 4 demonstrates how the classification accuracy degrades under different band-limited Gaussian noises on MNIST, CIFAR100, and Imagenette; and results on the other datasets are included in Appendix G. First, notice that the perturbation from the lowest band $\Delta x^{(0)}$ has a similar impact on all the models regardless of the algorithm they are trained by. There is however a noticeable difference in how models trained by SGD and adaptive gradient methods respond to perturbations from higher frequency bands. On models trained by SGD, the flattened curve implies that the effect of high-frequency perturbations on the generalization performance quickly diminishes to zero, suggesting that models are not sensitive to changes along the dimensions of irrelevant frequencies. Contrarily on models trained by the two adaptive gradient methods, we observe a difference in the way models respond to perturbations of higher frequency bands. On CIFAR100, for example, the two models are highly vulnerable to Gaussian perturbations from bands 5 to

7. This observation shows that when models, during their training phase, do not have mechanisms in place to limit their use of irrelevant frequencies, their performance can be compromised if data along irrelevant frequencies become corrupted at test time.

One can also observe that models' responses to high-frequency Gaussian perturbations varies among datasets. This can be attributed to the fact that (ir-)relevant frequencies are most likely going to be a unique characteristic for a particular dataset. We do not expect a dataset that solely consists of hand-written digits to share the same (ir-)relevant frequencies as one that consists of real-world objects. Moreover, the dimension (image resolution) of inputs for a given dataset matters, as a higher dimension potentially can allow more irrelevant frequencies. Therefore, we emphasize that the goal of our work is not to identify the exact (ir-)relevant frequencies among datasets. Rather, the analysis is built on the **presence** of irrelevant frequencies in the dataset, especially towards the higher end of the frequency spectrum, and how models differ in their robustness when trained by different algorithms. In the next section, we investigate the reason for such a robustness difference by studying how the irrelevant frequencies affect the learning dynamics of GD and signGD under a synthetic linear regression task.

## 4 Linear Regression Analysis with an Over-parameterized Model

In this section, we study the learning dynamics of GD and signGD on least squares regression with linear models to understand why models trained using the two algorithms have the same standard generalization performance but exhibit different robustness against perturbations. On a synthetic dataset that emulates the energy distribution of natural datasets in the frequency domain, we design a learning task that has multiple optima for the standard population risk, each with a different adversarial risk. We analyze the weight adaptation under GD and signGD in both spatial and frequency domains and show that training with signGD can result in larger weights associated with irrelevant frequencies, resulting in models with a higher adversarial risk. Our result verifies claim 3.1. We report the main results here and defer the full derivations to Appendix E.

### 4.1 Problem Setup

Consider a linear model $f(x, w) = \langle\, w\,,\, x\,\rangle$ with $x, w \in \mathbb{R}^d$, where $w$ and $x$ are the weight and the signal represented in the spatial domain, respectively. Since the DCT transformation matrix $C$ is an orthogonal matrix whose rows and columns are unit vectors, an alternative way to represent this model is:

$$f(x, w) = \langle\, w\,,\, x\,\rangle = w^\top x = w^\top C^\top C x = \langle\, \tilde{w}\,,\, \tilde{x}\,\rangle = f(\tilde{x}, \tilde{w}),$$

where $\tilde{w}$ and $\tilde{x}$ are the exact same weight and signal but are now represented in the frequency domain. This means that for linear models, computing the output of the model can be carried out in either domain as long as we use the matching representation of the signal and the weight. The goal of the linear analysis is to study the learning dynamics of different algorithms in a synthetic and controlled environment where one can clearly define the frequency-domain signal-target (ir)relevance to help understand the behavior of models in more complex settings. For this reason, let $\tilde{w}^*$ denote the frequency-domain representation of the true model that is used to interact with the input $\tilde{x}$ and generate the target: $y = \tilde{x}^\top \tilde{w}^*$, where $\tilde{w}^* = (\tilde{w}_0^*, \tilde{w}_2^*, \ldots, \tilde{w}_{d-1}^*)^\top$. We consider the squared error pointwise loss, which can be equally formulated in both domains:

$$\ell(x, y; w) = \frac{1}{2} |f(x, w) - y|^2 \qquad \text{and} \qquad \ell(\tilde{x}, y; \tilde{w}) = \frac{1}{2} |f(\tilde{x}, \tilde{w}) - y|^2$$
$$= \frac{1}{2} |\langle\, x\,,\, w\,\rangle - \langle\, x\,,\, w^*\,\rangle|^2 \qquad\qquad\qquad = \frac{1}{2} |\langle\, \tilde{x}\,,\, \tilde{w}\,\rangle - \langle\, \tilde{x}\,,\, \tilde{w}^*\,\rangle|^2 .$$

Denote the error between the learned weight and the true weight at iteration $t$ by $e(t) = w(t) - w^*$, and the standard risk by $\mathcal{R}_s(w) = \mathbb{E}\left[\ell(X, Y; w)\right]$. In a similar way, those terms can be represented in the frequency domain as $\tilde{e}(t) = \tilde{w}(t) - \tilde{w}^*$ and $\mathcal{R}_s(\tilde{w}) = \mathbb{E}\left[\ell(\tilde{X}, Y; \tilde{w})\right]$. Now we are ready to explain the design philosophy behind the synthetic dataset, the structure of the true model $\tilde{w}^*$, and particularly, with regard to robustness, the ideal model that minimizes the effect of perturbations.

Suppose that $\tilde{X}$ follows a Gaussian distribution $\mathcal{N}(\tilde{\mu}, \tilde{\Sigma})$. For analytical tractability, we consider $\tilde{\mu} = 0$ and a diagonal structure of $\tilde{\Sigma}$, i.e., $\tilde{\Sigma} = \text{diag}(\tilde{\sigma}_0^2, ..., \tilde{\sigma}_{d-1}^2)$. This implies that in the spatial domain, $X$ follows a Gaussian distribution $\mathcal{N}(0, \Sigma)$ where $\Sigma = C^\top \tilde{\Sigma} C$. In Appendix E.1, we provide examples of the spatial-domain structure of the data, when we define the distribution directly in the frequency domain. In Sec. 3, we demonstrate that natural datasets exhibit a particular energy profile where signals contain irrelevant information represented by high-frequency and low-amplitude waves. To emulate this setting with a synthetic dataset, we define frequencies that are (ir)relevant in generating the target. Let $\mathbb{I}_{\text{irrel}} \subseteq \{1, 2, ..., d-1\}$ and $\mathbb{I}_{\text{rel}} = \{0, 1, 2, ..., d-1\} - \mathbb{I}_{\text{irrel}}$ denote the set of irrelevant and relevant frequencies, respectively. Recall that the goal is to make high-frequency components of the dataset irrelevant, so we exclude the DC term ($0 \notin \mathbb{I}_{\text{irrel}}$) when considering irrelevant frequencies, as it is the lowest frequency possible. Next, we specify the energy distribution of the synthetic dataset. The expected energy of a random signal following such a distribution is

$$\mathbb{E}\left[E(\tilde{X})\right] = \mathbb{E}\left[\sum_{i=0}^{d-1} |\tilde{X}_i|^2\right] = \sum_{i=0}^{d-1} \mathbb{E}\left[\tilde{X}_i^2\right] = \sum_{i=0}^{d-1} \tilde{\sigma}_i^2. \tag{7}$$

We assume that $\tilde{\sigma}_i^2 = 0$ if $i \in \mathbb{I}_{\text{irrel}}$, meaning the irrelevant frequencies of the data from the synthetic dataset have zero energy contributions. The purpose of this is to imitate the behavior of real-world datasets, where the high-frequency components have a negligible impact on the overall energy of the signal.

To see how having irrelevant frequencies affect the structure of the true model, notice that the definition of the synthetic dataset implies $\tilde{X}_i = 0$ for all $i \in \mathbb{I}_{\text{irrel}}$. This means that the true target value does not depend on those irrelevant frequencies. Clearly, this linear model is over-paramaterized because one only needs to specify $\tilde{w}_i^*$ for all $i \in \mathbb{I}_{\text{rel}}$ to establish the signal-target relationship.

The objective of the standard risk with such a synthetic dataset is not strictly convex, i.e., there are multiple minimizers with zero standard risk, as the value of $\tilde{w}_i^*$ for all $i \in \mathbb{I}_{\text{irrel}}$ has no impact on the model output. For clarity, let us define $\tilde{\mathcal{W}}^* = \left\{ \tilde{w}^* : \mathcal{R}_s(\tilde{w}^*) = 0 \right\}$ as the set that includes all standard risk minimizers. Having multiple standard risk minimizers is the result of over-parametrization; however, there is a unique solution that achieves zero standard risk and makes the model immune to any perturbations parallel to the directions of the irrelevant frequencies, and it corresponds to having zero weight at irrelevant frequencies: $\tilde{w}_i^* = 0$ for all $i \in \mathbb{I}_{\text{irrel}}$: Define such a robust standard risk minimizer as $\tilde{w}^{\text{R}} \in \tilde{\mathcal{W}}^*$, we have

$$\tilde{w}_i^{\text{R}} \triangleq \begin{cases} \tilde{w}_i^* & \text{for all} \quad i \in \mathbb{I}_{\text{rel}} \\ 0 & \text{otherwise.} \end{cases} \tag{8}$$

Note that we use $\tilde{w}^*$ to denote any arbitrary standard minimizers in $\tilde{\mathcal{W}}^*$. To see why $\tilde{w}^{\text{R}}$ is the most robust standard minimizer, we introduce the adversarial risk to capture the worst-case performance of the model under an $\ell_2$-constrained perturbation. Similar to the squared error loss, the adversarial risk can also be equally formulated in both domains:

$$\mathcal{R}_{\text{a}}(w) \triangleq \mathbb{E}_{(X,Y)}\left[\max_{||\Delta x||_2 \leq \epsilon} \ell(X + \Delta x, Y; w)\right] \qquad \text{and} \qquad \mathcal{R}_{\text{a}}(\tilde{w}) \triangleq \mathbb{E}_{(\tilde{X},Y)}\left[\max_{||\Delta \tilde{x}||_2 \leq \epsilon} \ell(\tilde{X} + \Delta \tilde{x}, Y; \tilde{w})\right],$$

where the $\ell_2$-constraint with a size of $\epsilon$ has an equivalent effect in both domains. To understand the adversarial risk from a frequency-domain perspective, let us focus on $\mathcal{R}_{\text{a}}(\tilde{w})$:

$$\mathcal{R}_{\text{a}}(\tilde{w}) = \mathbb{E}_{\tilde{X}}\left[\max_{||\Delta \tilde{x}||_2 \leq \epsilon} \frac{1}{2} \left|\left\langle \tilde{X}, \tilde{w} - \tilde{w}^* \right\rangle + \left\langle \Delta \tilde{x}, \tilde{w} \right\rangle\right|^2\right], \tag{9}$$

where we focus on the expectation over $\tilde{X}$, as $Y$ is replaced with $\left\langle \tilde{X}, \tilde{w}^* \right\rangle$. Notice that the maximization is inside the expectation. This means that we are finding a separate perturbation for each input. Therefore, the maximizer, $\Delta \tilde{x}^*$, of a given $\tilde{X}$ within the expectation in (9) is

$$\Delta \tilde{x}^* \triangleq \arg\max_{||\Delta \tilde{x}||_2 \leq \epsilon} \frac{1}{2} \left|\left\langle \tilde{X}, \tilde{w} - \tilde{w}^* \right\rangle + \left\langle \Delta \tilde{x}, \tilde{w} \right\rangle\right|^2 = \epsilon \, \text{sign}[\left\langle \tilde{X}, \tilde{w} - \tilde{w}^* \right\rangle] \frac{\tilde{w}}{||\tilde{w}||_2}. \tag{10}$$

Now knowing the worst-case perturbation to any $\tilde{X}$, we can continue the derivation in (9) with

$$
\begin{aligned}
\mathcal{R}_{\mathrm{a}}(\tilde{w}) &= \frac{1}{2}\mathbb{E}_{\tilde{X}}\left[\left|\left\langle\,\tilde{X}\,,\,\tilde{w}-\tilde{w}^*\,\right\rangle + \epsilon\operatorname{sign}[\left\langle\,\tilde{X}\,,\,\tilde{w}-\tilde{w}^*\,\right\rangle]\,\|\tilde{w}\|_2\right|^2\right] \\
&= \frac{1}{2}\sum_{i\in\mathbb{I}_{\mathrm{rel}}}\tilde{\sigma}_i^2(\tilde{w}_i-\tilde{w}_i^*)^2 + \epsilon\sqrt{\frac{2}{\pi}\sum_{i\in\mathbb{I}_{\mathrm{rel}}}\tilde{\sigma}_i^2(\tilde{w}_i-\tilde{w}_i^*)^2}\,\|\tilde{w}\|_2 + \frac{\epsilon^2}{2}\,\|\tilde{w}\|_2^2.
\end{aligned}
\tag{11}
$$

Finding the exact minimizer to (11) is more involved. Without doing that however, it is obvious that for an arbitrary standard risk minimizer $\tilde{w}^*$ from $\tilde{\mathcal{W}}^*$, we can evaluate (11) at $\tilde{w}^*$ and obtain its the adversarial risk as

$$
\mathcal{R}_{\mathrm{a}}(\tilde{w}^*) = \frac{\epsilon^2}{2}\|\tilde{w}^*\|_2^2,
\tag{12}
$$

where the first two terms in (11) become zero at any fixed standard risk minimizer. This shows that the robustness of the standard risk minimizers against $\ell_2$-bounded perturbations is inversely proportional to the norm of the linear model. That is, a smaller norm implies better robustness. Recall that when evaluating the standard risk of the model, the weights associated with irrelevant frequencies do not matter, since they are never used in computing the output of the model. On the contrary, the $\|\tilde{w}^*\|_2^2$ term in (12) implies that those weights matter when considering the robustness of the model under perturbations. It is not difficult to see that the minimum adversarial risk can be achieved on a unique standard risk minimizer $\tilde{w}^{\mathrm{R}}$ (8).

Therefore, in the over-parameterized linear regression setting, a standard risk minimizer with a minimum norm is preferred when considering the robustness of the model, and a model with zero weight at irrelevant frequencies is the most robust solution among the standard risk minimizers. With this example, we verify Claim 3.1. While standard risk minimizers can correctly use the relevant information of the data, their use of irrelevant information is under-constrained. This can result in significant weight assigned to irrelevant frequencies, making models more susceptible to perturbations.

Next, we study the learning dynamics of GD and signGD and demonstrate that the solutions found by GD and signGD differ in the weight of the irrelevant frequencies. This causes the solutions found by the two algorithms to have a similar standard population risk, but behave very differently under perturbations.

## 4.2 Analysis on the Learning Dynamics of GD and signGD

We now analyze the weight adaptation of a linear model under GD and signGD, and experimentally verify our results. Our analysis shows that for the over-parameterized linear model, GD finds solutions with a standard risk of exactly zero, and signGD finds solutions with a standard risk close to zero. However, they have different robustness properties. In the presence of irrelevant frequencies, GD is more likely to converge to a solution that is less sensitive to perturbations along the direction of irrelevant frequencies, whereas signGD is more likely to converge to solutions that are more prone to such perturbations.

### 4.2.1 GD Dynamics

Let us start with GD in the spatial domain. Suppose that we initialize the weights in the spatial domain as $w(0) = W \sim N(0, \Sigma_W)$ where $\Sigma_W \in \mathbb{R}^{d\times d}$. Similar to how both $\tilde{X}$ and $X$ follow a Gaussian distribution, the frequency representation of the initialized weight also follows a Gaussian distribution: $\tilde{w}(0) = \tilde{W} \sim \mathcal{N}(0, \tilde{\Sigma}_W)$ where $\tilde{\Sigma}_W = C\Sigma_W C^\top$. To train the model, we use GD on the population risk:

$$
w(t+1) \leftarrow w(t) - \eta\nabla_w\mathcal{R}_{\mathrm{s}}(w(t)).
\tag{13}
$$

The gradient computed using the population risk is $\nabla_w\mathcal{R}_{\mathrm{s}}(w(t)) = \mathbb{E}\left[XX^\top\right]e(t) = \Sigma e(t)$, and the learning dynamics of GD in the spatial domain can be captured using:

$$
e(t+1) = w(t+1) - w^* = w(t) - w^* - \eta\Sigma e(t) = (I - \eta\Sigma)^{t+1}e(0).
\tag{14}
$$

This shows that the learned weight converges to the optimal weight $w^*$ at a rate depending on $\Sigma$. To see the GD dynamics in the frequency domain, we can simply perform DCT on both sides of (14):

$$
\tilde{e}(t+1) = C(I - \eta\Sigma)^{t+1}e(0) = (I - \eta\tilde{\Sigma})^{t+1}\tilde{e}(0),
\tag{15}
$$

where $\tilde{\Sigma}$ is the covariance of $\tilde{x}$. It is easy to see that no weight adaptation happens for the irrelevant frequencies because $\tilde{\sigma}_i^2 = 0$ for all $i \in \mathbb{I}_{\text{irrel}}$. As $\tilde{\Sigma}$ is diagonal, choosing the learning rate $\eta$ such that $\eta \max_{i \in \{0,\ldots,d-1\}} \tilde{\Sigma}_{ii} < 1$, we get that the asymptotic solution is

$$\boldsymbol{\tilde{w}}_i^{\text{GD}} \triangleq \lim_{t \to \infty} \tilde{w}_i(t) = \begin{cases} \tilde{w}_i^* & \text{for all} \quad i \in \mathbb{I}_{\text{rel}} \\ \tilde{w}_i(0) & \text{otherwise.} \end{cases} \tag{16}$$

That is, the initial random weights at the irrelevant frequencies do not change. Using (12), we have

$$\mathcal{R}_{\text{a}}(\boldsymbol{\tilde{w}}^{\text{GD}}) = \frac{\epsilon^2}{2}||\boldsymbol{\tilde{w}}^{\text{GD}}||_2^2 = \frac{\epsilon^2}{2}\left\{\sum_{i \in \mathbb{I}_{\text{rel}}} \tilde{w}_i^{*2} + \sum_{j \in \mathbb{I}_{\text{irrel}}} \tilde{w}_j(0)^2\right\}. \tag{17}$$

Comparing the standard risk minimizer found by GD with the robust standard risk minimizer in (8), we notice that the GD solution is not the most robust among all standard risk minimizers, as it is sensitive to perturbations along irrelevant frequencies. Suppose that the initialized weight in the frequency domain is randomly sampled from $\mathcal{N}(0, \sigma^2 I_{d \times d})$, and the signal-target relationship is determined by a handful of relevant frequencies. Taking the expectation of (17) over the randomly initialized weight, we have $\mathbb{E}_{\tilde{w}(0)}\left[\mathcal{R}_{\text{a}}(\boldsymbol{\tilde{w}}^{\text{GD}})\right] \approx O(\epsilon^2 d\sigma^2)$, so the adversarial risk can be quite significant if there is a large number of irrelevant frequencies, i.e., $|\mathbb{I}_{\text{rel}}| \ll d$ and $|\mathbb{I}_{\text{irrel}}| \approx d$.

This example shows that the GD solution is sensitive to initialization. Because there is no mechanism in place to actively ensure that the weights associated with these irrelevant frequencies become zero, GD is not forcing the initial weights to go to zero at those frequencies. One solution is to include the weight norm as a penalty term along with the original optimization objective, but this can result in learning a biased solution. Another simple fix is to initialize the weight at exactly 0. This robustifies the GD solution by initializing those irrelevant weights at the most robust state.

### 4.2.2 SignGD Dynamics

Adaptive gradient algorithms like Adam and RMSProp utilize historical gradient information as a momentum mechanism for updating model parameters, thereby expediting the learning process. However, it is important to note that their acceleration is not solely attributable to this feature, nor is their adaptiveness limited to it. In (3), we have demonstrated how signGD, a memory-free version of Adam and RMSProp, can adaptively adjust the update using a coordinate-wise learning rate. Although signGD is not a suitable choice for training deep neural networks (Riedmiller & Braun, 1993; Ma et al., 2022b), examining its behavior can provide insights into the learning dynamics of other adaptive gradient methods (Karimi et al., 2016; Balles & Hennig, 2018; Moulay et al., 2019). Additionally, in Sec. 4.2.3, we empirically justify the use of signGD as a suitable alternative in understanding the learning dynamics of Adam and RMSProp.

Again, let us start with signGD in the spatial domain. The update rule using the population risk takes the sign of the gradient computed using the population risk

$$w(t+1) \leftarrow w(t) - \eta \operatorname{sign}[\nabla_w \mathcal{R}_{\text{s}}(w)], \tag{18}$$

and its learning dynamics in the spatial domain is

$$e(t+1) = w(t+1) - w^* = e(t) - \eta \operatorname{sign}[\Sigma e(t)]. \tag{19}$$

Unlike the GD dynamics in (14), (19) shows that the behavior of signGD depends on the sign of $\Sigma e(t)$, and this means that when $|[\Sigma e(t)]_i| \ll 1$, training using signGD can accelerate the learning along the $i$-th dimension. Although we can obtain $\Sigma$ from $\Sigma = C^\top \tilde{\Sigma} C$, the structure of $\Sigma$ is subject to variation based on $\tilde{\Sigma}$, so it is difficult to find an analytical solution for the dynamics of the model trained under signGD, such as (14) where we have a closed form for $e(t)$ as a function of $e(0)$ for models trained under GD. This means that analyzing the signGD dynamics is limited to studying its step-by-step update based on the sign of the entries in $\Sigma e(t)$.

The signGD learning dynamics in the frequency domain can be obtained by taking the DCT transformation on both sides of (19):

$$\tilde{e}(t+1) = \tilde{e}(t) - \eta C \operatorname{sign}[\Sigma e(t)] = \tilde{e}(t) - \eta C \operatorname{sign}[C^\top \tilde{\Sigma} \tilde{e}(t)], \tag{20}$$

where the error and the covariance terms inside of the sign are also transformed into their frequency-domain representations. Equation 20 shows that analyzing the behavior of signGD in the frequency domain requires knowing the sign of the entries in $C^\top \tilde{\Sigma} \tilde{e}(t)$. This term can be understood as an inverse DCT transformation of $\tilde{\Sigma} \tilde{e}(t)$, and with a diagonal structure of $\tilde{\Sigma}$, we know that $\tilde{\Sigma} \tilde{e}(t) = \left[ \tilde{\sigma}_0^2, ..., \tilde{\sigma}_{d-1}^2 \right]^\top \odot \tilde{e}(t)$. However, similar to the situation in (19), the sign of the entries in $C^\top \tilde{\Sigma} \tilde{e}(t)$ is dependent on $\tilde{e}(t)$ at different steps, so obtaining an analytical solution for the frequency-domain dynamics is also challenging.

In both (19) and (20), we see that understanding the signGD dynamics for any general $\tilde{\Sigma}$ can be complicated. Thus, we focus on a structure of $\tilde{\Sigma}$ that simplifies the analysis but still allows us to understand why training with signGD results in vulnerable models. In particular, we consider a data distribution where $\tilde{X} \sim \mathcal{N}(\tilde{\mu} = 0, \tilde{\Sigma} = \operatorname{diag}\left\{ \tilde{\sigma}_0^2, \tilde{\sigma}_1^2, 0 \right\})$. This definition implies that the data distribution contains irrelevant information at the highest frequency basis and we have $\tilde{X}_2 = 0$ for all datapoints.

Now, we continue with signGD learning dynamics in the frequency domain from (20). Let us denote $A(t) = \frac{\sqrt{3}}{3} \tilde{\sigma}_0^2 \tilde{e}_0(t)$ and $B(t) = \frac{\sqrt{2}}{2} \tilde{\sigma}_1^2 \tilde{e}_1(t)$, and $C = C^{(3)}$ follows the DCT transformation matrix defined in (4). With some algebraic manipulation, we have

$$\tilde{e}(t+1) = \tilde{e}(t) - \eta \begin{bmatrix} \frac{\sqrt{3}}{3}(\operatorname{sign}[A(t)+B(t)] + \operatorname{sign}[A(t)] + \operatorname{sign}[A(t)-B(t)]) \\ \frac{\sqrt{2}}{2}(\operatorname{sign}[A(t)+B(t)] - \operatorname{sign}[A(t)-B(t)]) \\ \frac{\sqrt{6}}{6}\operatorname{sign}[A(t)+B(t)] - \frac{\sqrt{6}}{3}\operatorname{sign}[A(t)] + \frac{\sqrt{6}}{6}\operatorname{sign}[A(t)-B(t)] \end{bmatrix}, \tag{21}$$

and we include its complete derivation in Appendix E.7. With this particular choice of $\tilde{\Sigma}$, (21) shows that weight adaptation depends on the sign of three terms: $A(t)$, $A(t) + B(t)$ and $A(t) - B(t)$. In Table 9 of Appendix E.8, we study the learning dynamics of signGD by analyzing all 27 sign combinations and their corresponding updates. We report the main results here and defer the detailed analysis to Appendix E.8.

With a constant learning rate of $\eta$, the asymptotic signGD solution converges to an $O(\eta)$ neighborhood of the standard risk minimizer:

$$\limsup_{t \to \infty} |\tilde{w}_i(t) - \tilde{w}_i^*| = O(\eta), \tag{22}$$

where $i \in \{0, 1\}$. In particular, we demonstrate that $\tilde{w}_0$ oscillates in an $O(\eta)$ neighborhood of $\tilde{w}_0^*$. Consider $T$ as the first iteration after which $\tilde{w}_0$ starts oscillating, and define $\Delta \tilde{w}_2$ as the sum of all the updates in $\tilde{w}_2$ up to the $T$-th iteration. The limiting behavior of $\tilde{w}_2$ under signGD update is

$$\limsup_{t \to \infty} |\tilde{w}_2(t)| = |\tilde{w}_2(T) + O(\eta)| = |\tilde{w}_2(0) + \Delta \tilde{w}_2 + O(\eta)|, \tag{23}$$

where $\tilde{w}_2(0)$ is the weight at initialization. This means that after $T$ iterations, for all $t' > T$, $\tilde{w}_2(t')$ stays in an $O(\eta)$ neighborhood of $\tilde{w}_2(T)$. As such, we have the asymptotic solution found by signGD:

$$\boldsymbol{\tilde{w}}^{\mathrm{signGD}} = [\tilde{w}_0^*, \ \tilde{w}_1^*, \ \tilde{w}_2(0) + \Delta \tilde{w}_2]^\top + O(\eta). \tag{24}$$

From the perspective of training under the standard risk, the signGD solution is close to the optimum. Specifically, its standard risk is

$$\mathcal{R}_{\mathrm{s}}(\boldsymbol{\tilde{w}}^{\mathrm{signGD}}) = \mathbb{E}\left[ \ell(\tilde{X}, Y; \boldsymbol{\tilde{w}}^{\mathrm{signGD}}) \right] = \frac{1}{2}\mathbb{E}\left[ \left\langle \tilde{X}, \boldsymbol{\tilde{w}}^{\mathrm{signGD}} - \tilde{w}^* \right\rangle^2 \right] = O((\tilde{\sigma}_0^2 + \tilde{\sigma}_1^2)\eta^2). \tag{25}$$

Note that the standard risk of the GD solution is exactly zero; and by choosing a sufficiently small learning rate $\eta$, the standard risk of the signGD solution can also be close to zero as well. However, their adversarial risks are very different. Specifically, the adversarial risk of the asymptotic signGD solution is

$$\mathcal{R}_{\mathrm{a}}(\boldsymbol{\tilde{w}}^{\mathrm{signGD}}) = \frac{\epsilon^2}{2}||\boldsymbol{\tilde{w}}^{\mathrm{signGD}}||_2^2 = \frac{\epsilon^2}{2}\left\{ \tilde{w}_0^{*2} + \tilde{w}_1^{*2} + (\tilde{w}_2(0) + \Delta \tilde{w}_2)^2 \right\}. \tag{26}$$

We can compare it with the adversarial risk of the asymptotic solution found by GD under the same setup:

$$\mathcal{R}_a(\tilde{\boldsymbol{w}}^{\text{GD}}) = \frac{\epsilon^2}{2} \left\{ \tilde{w}_0^{*2} + \tilde{w}_1^{*2} + \tilde{w}_2(0)^2 \right\}. \tag{27}$$

It can be observed that the main difference between the two adversarial risks in (26) and (27) arises from the difference in weights learned at the irrelevant frequency. Since their use of irrelevant frequency in the data is under-constrained, neither algorithm can reduce $\tilde{w}_2$ to zero, thereby neither solution is the most robust standard risk minimizer. As discussed in Sec. 4.2.1, the GD solution is sensitive to weight initialization. Before understanding the $\Delta \tilde{w}_2$ term in the signGD solution, we first introduce two assumptions on the synthetic dataset that serve to better emulate the distribution found in the natural dataset. Consider a dataset with a strong task-relevant correlation between the relevant frequency component of the data and the target, a realistic scenario as we discussed in Sec. 3.2. In this case, $|\tilde{w}_0^*|$ and $|\tilde{w}_1^*|$ can be large. Additionally, with a weight initialization around zero, such as in methods by He et al. (2015) and Glorot & Bengio (2010), the initial error $|\tilde{e}_0(0)|$ and $|\tilde{e}_1(0)|$ can be large and close to $|\tilde{w}_0^*|$ and $|\tilde{w}_1^*|$ when $|\tilde{w}_0^*| \gg |\tilde{w}_0(0)|$ and $|\tilde{w}_1^*| \gg |\tilde{w}_1(0)|$. Moreover, it is discussed in Sec. 3.1 and later supported empirically in Figure 8 of Appendix G that the distribution of spectral energy heavily concentrates at the low end of the frequency spectrum and decays quickly towards higher frequencies. Since $\tilde{\sigma}_i^2$ is interpreted as the expected energy of a random variable at the $i$-th frequency, it is reasonable to expect that $\frac{\tilde{\sigma}_1^2}{\tilde{\sigma}_0^2} < \frac{1}{3}$.

With the two assumptions, we demonstrate that $\Delta \tilde{w}_2$ is proportional to $|\tilde{w}_0^*|$ or $|\tilde{w}_1^*|$ depending on the initialization of $|A(0)|$ and $|B(0)|$. In particular, we have

$$|\Delta \tilde{w}_2| \approx \begin{cases} \sqrt{3}C\,|\tilde{w}_0^*| & \text{if} \quad |A(0)| < |B(0)| \\ \frac{3\sqrt{2}\tilde{\sigma}_1^2}{2\tilde{\sigma}_0^2}C\,|\tilde{w}_1^*| & \text{if} \quad |A(0)| > |B(0)|, \end{cases} \tag{28}$$

where $C \in [\frac{\sqrt{6}}{6}, \frac{\sqrt{6}}{3}]$. To quantitatively understand the robustness difference between solutions found by the two algorithms, we consider the ratio between the adversarial risk of the standard risk minimizers found by GD (27) and signGD (26) with a three-dimensional input space. We observe that the solution found by signGD is more sensitive to perturbations compared to the GD solution:

$$\frac{\mathcal{R}_a(\tilde{\boldsymbol{w}}^{\text{signGD}})}{\mathcal{R}_a(\tilde{\boldsymbol{w}}^{\text{GD}})} \approx \begin{cases} 1 + C_3 \frac{\tilde{w}_0^{*2}}{\tilde{w}_0^{*2} + \tilde{w}_1^{*2}} & \text{if} \quad |A(0)| < |B(0)| \\ 1 + C_4 \frac{\tilde{w}_1^{*2}}{\tilde{w}_0^{*2} + \tilde{w}_1^{*2}} & \text{if} \quad |A(0)| > |B(0)|, \end{cases} \tag{29}$$

where $\frac{1}{2} \leq C_3 \leq 2$ and $\frac{3}{4}\frac{\tilde{\sigma}_1^4}{\tilde{\sigma}_0^4} \leq C_4 \leq 3\frac{\tilde{\sigma}_1^4}{\tilde{\sigma}_0^4}$. Given that this ratio is always greater than 1, the linear model obtained through GD is always more robust against $\ell_2$-bounded perturbations in comparison to the model obtained from signGD.

### 4.2.3 Empirical Validation

To validate our analysis, in Figure 5 we create a three-dimensional dataset using $(\tilde{\sigma}_0^2, \tilde{\sigma}_1^2, \tilde{\sigma}_2^2) = (0.01, 0.0025, 0)$, and $(\tilde{w}_0^*, \tilde{w}_1^*, \tilde{w}_0^*) = (5, 10, 0)$, and compare the dynamics of the frequency-domain weight error on models trained by GD, Adam, RMSProp, and signGD. All models are initialized with the same weight and are trained using a fixed learning rate of 0.01. At each training iteration, we sample 1000 data points and compute the gradient based on the sampled data. We want to clarify that even though we analyze the weight update dynamics in both frequency and spatial domains, the actual training still takes place in the spatial domain.

In (15), we show that the GD solution $\tilde{\boldsymbol{w}}_i^{\text{GD}}(t)$ converges to $\tilde{w}_i^*$ with a rate of $1 - \eta\tilde{\sigma}_i^2$. Therefore, when $\tilde{\sigma}_i^2$ is small, learning can be particularly slow for weights associated with the $i$-th frequency, as shown in Figure 5a. On the other hand, notice in Table 9 that $|\tilde{e}_0|$ gets reduced by at least $\frac{\sqrt{3}}{3}$ regardless of the magnitude of $\tilde{\sigma}_0^2$ for signGD. This means that the magnitude of $\tilde{\sigma}_i^2$ does not directly affect the convergence speed. Instead, the relative magnitude between $A(t)$ and $B(t)$ determines the frequency which receives priority during the learning process. As a result, we observe an acceleration for models trained by signGD.

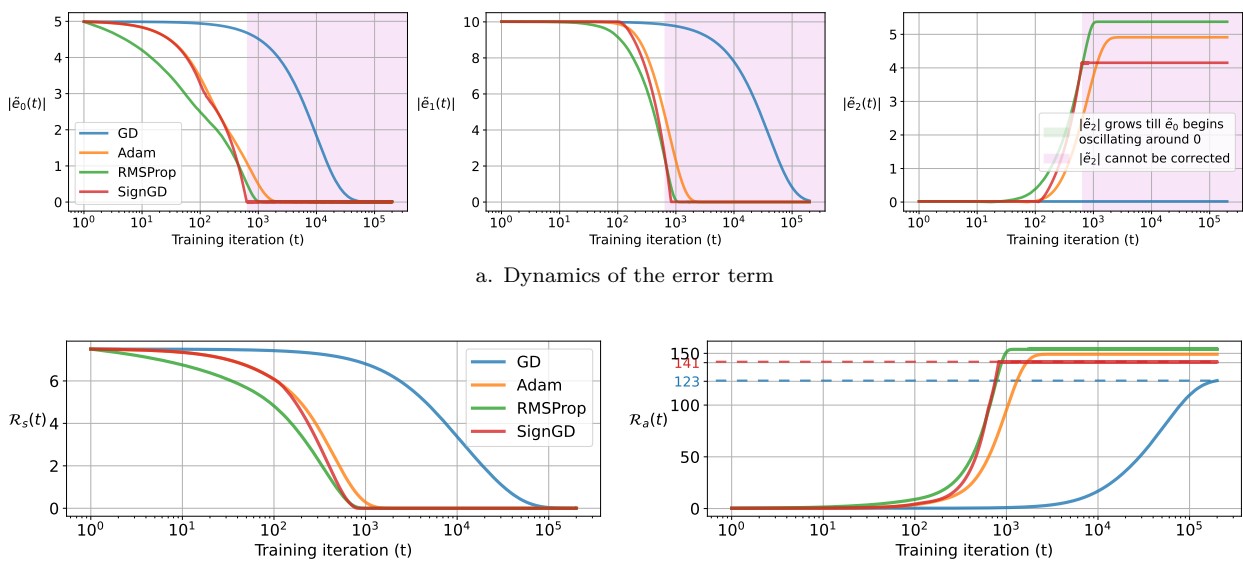

a. Dynamics of the error term

b. The standard population risk $\mathcal{R}_s$ and the adversarial population risk $\mathcal{R}_a$

Figure 5: **Comparing (a) the learning dynamics, (b) the standard and adversarial population risk of linear models trained by GD, Adam, RMSProp, and signGD.** We create a three-dimensional dataset created using $(\tilde{\sigma}_0^2, \tilde{\sigma}_1^2, \tilde{\sigma}_2^2) = (0.01, 0.0025, 0)$, and $(\tilde{w}_0^*, \tilde{w}_1^*, \tilde{w}_2^*) = (5, 10, 0)$. All models are initialized with the same weight $(\tilde{w}_0(0), \tilde{w}_1(0), \tilde{w}_2(0)) = (0.01, -0.01, 0.02)$ and trained using a fixed learning rate of 0.01. **(a) Dynamics of the error term.** During the signGD training process, the error along the irrelevant frequency grows until $\tilde{e}_0$ starts to oscillate around 0. In our example, the green highlighted areas in the figure correspond to the iterations before $\tilde{e}_0$ starts to oscillate, and the red areas show that the error along the irrelevant frequency cannot be corrected. **(b) The standard population risk and the adversarial population risk ($\epsilon = \sqrt{2}$).** We notice that despite all models can reach zero standard population risk, their adversarial population risks are different. The adversarial population risk of models trained by adaptive gradient methods is higher than the one from the model trained by GD, indicating lower robustness.

Next, we observe that the error trajectory for the model trained by signGD closely resembles the one from the model trained by Adam for $\tilde{e}_0$ and $\tilde{e}_1$. In the analysis of signGD, we show that $|\tilde{e}_2|$ increases till $|\tilde{e}_0|$ starts oscillating in $O(\eta)$. Figure 5a shows that this pattern can be observed in models trained by Adam as well. This shows that signGD is a suitable alternative to understanding the learning dynamics of models under the proposed linear regression task. For models trained by GD, since there is no update on the weight associated with the irrelevant frequency, $\tilde{e}_2$ remains at the initialized value throughout training. To demonstrate the weight adaptation under signGD, we divide the training into two phases, as highlighted by two background colors. The green area indicates that $|\tilde{e}_0|$ decreases and $|\tilde{e}_2|$ increases in the meanwhile. Once oscillation begins for $|\tilde{e}_0|$, $|\tilde{e}_2|$ can no longer be corrected. This behavior corresponds to the purple area in Figure 5a.

In Figure 5b, we compare the standard population risk and the adversarial population risk of different models. We notice that despite all models reaching near zero standard population risk, their adversarial population risk is different. In particular, the adversarial population risk of models trained by adaptive gradient methods is higher than the one from the model trained by GD, indicating lower robustness. Choosing $\epsilon = \sqrt{2}$ in (12), the adversarial risk of those standard risk minimizers is exactly the squared $\ell_2$ norm of the weight. With our choice of initialization, the resulting $|A(0)|$ and $|B(0)|$ are 0.0289 and 0.0177 respectively. This means that the ratio between the two adversarial risks is $\frac{\mathcal{R}_a(\tilde{\boldsymbol{w}}^{\text{signGD}})}{\mathcal{R}_a(\tilde{\boldsymbol{w}}^{\text{GD}})} \in [1.04, 1.15]$ according to (29), and this aligns with the ratio of 1.146 obtained empirically from the experiments.

This simple problem illustrates how the optimization algorithms and an over-parameterized model might interact, and learning with signGD can lead to a solution that is more prone to perturbations. In this section, we focus on analyzing the robustness of the solution from a frequency domain perspective, that is,

Table 1: **Comparing the upper bound on the Lipschitz constant and the averaged robust accuracy of neural networks trained by SGD, Adam, and RMSProp.** We follow (Gouk et al., 2021) to compute the Lipschitz constants of each layer in isolation and multiply them together to establish an upper bound on the constant of the entire network. Notice that across all selected datasets, models trained by SGD have a considerably smaller upper bound compared to models trained by Adam and RMSProp. In Figure 1, we demonstrate the robustness of the neural networks under Gaussian noise, $\ell_2$ and $\ell_\infty$ bounded adversarial perturbations (Croce & Hein, 2020). Here, we average the accuracy across the perturbations and get a single score quantifying the model's robustness. All results are averaged over three independently initialized and trained models.

| | Dataset | MNIST | Fashion | CIFAR10 | CIFAR100 | SVHN | Caltech101 | Imagenette |
|---|---|---|---|---|---|---|---|---|
| | SGD | **3.80** | **3.83** | **26.81** | **40.41** | **22.65** | **18.53** | **23.99** |
| $\prod_{i=1}^{l} L(\phi_i)$ | Adam | 5.75 | 8.12 | 28.70 | 41.87 | 30.45 | 26.20 | 28.55 |
| | RMSProp | 6.21 | 5.11 | 37.75 | 41.71 | 28.31 | 45.84 | 27.11 |
| | SGD | **77.97%** | **77.95%** | **63.21%** | **55.65%** | **69.08%** | **71.42%** | **67.59%** |
| Averaged Robust Acc. | Adam | 65.64% | 67.60% | 57.71% | 45.25% | 65.60% | 55.03% | 58.86% |
| | RMSProp | 63.54% | 71.34% | 56.47% | 47.55% | 65.37% | 53.16% | 57.98% |

the behavior of $\tilde{w}$ with an input perturbation of $\Delta\tilde{x}$. In Appendix E.9, we present a spatial interpretation of the result and demonstrate how signals with irrelevant frequencies contain spatially redundant dimensions.

## 5 Connecting the Norm of Linear Models to the Lipschitzness of Neural Networks

The takeaway from the over-parameterized linear regression analysis is that among all standard risk minimizers, the minimum norm solution is the most robust one. That is, a smaller weight norm implies better robustness. This suggests a connection between the weight norm and model robustness. Nonetheless, the major limitation of the analysis is that it is designed for a linear model. In this section, we generalize such a connection to the deep learning setting and verify it using the robustness of neural networks trained by different algorithms.

One major obstacle is that the notion of weight norm as defined for linear models is not generally applicable to neural networks. However, we can still relate the weight of a network to its sensitivity with respect to changes in the input space. Consider a single-layer ReLU-activated feedforward network with $x \in \mathbb{R}^d$ and $W \in \mathbb{R}^{D \times d}$. With a perturbation of $\Delta x \in \mathbb{R}^d$ constrained by the vector $\ell_p$-norm, the maximum change in the model output as measured by the same norm can be bounded using

$$\|\text{ReLU}(W(x + \Delta x)) - \text{ReLU}(Wx)\|_p \leq \|W\Delta x\|_p \leq \|W\|_p \|\Delta x\|_p, \tag{30}$$

where $\|W\|_p$ denotes the vector $\ell_p$-norm induced matrix norm of the weight $W$ and is referred to as the Lipschitz constant of this single-layer model.

Consider the $\ell_p$ vector norm, for all $x_1, x_2 \in \mathbb{R}$, a function $f$ is said to be Lipschitz continuous if $\|f(x_1) - f(x_2)\|_p \leq L\|x_1 - x_2\|_p$, for some real-valued Lipschitz constant $L \geq 0$.[4] Indeed, the Lipschitz constant of a function with respect to inputs captures how sensitive the model is in relation to changes in the input space.

In the single-layer model example, its Lipschitz constant is exactly the matrix norm of the weight. More generally, consider the feed-forward neural network as a series of function compositions:

$$f(x) = (\phi_l \circ \phi_{l-1} \circ \ldots \circ \phi_1)(x),$$

---

[4]Any value of $L$ satisfying the Lipschitz condition is considered a valid Lipschitz constant. For the sake of clarity, we will refer to the smallest (optimal) Lipschitz constant as $L$.

where each $\phi_i$ is a linear operation, an activation function, or a pooling operation. A particularly useful property of the Lipschitz function is that the composition of Lipschitz functions with Lipschitz constant $L_1$, $L_2$, ..., $L_N$ w.r.t. the same norm is also Lipschitz with an upper-bound on the Lipschitz constant $L \leq L_1 L_2 ... L_N$. Denoting the Lipschitz constant of function $f$ as $L(f)$, we can establish an upper bound on the Lipschitz constant for the entire feed-forward neural network using

$$L(f) \leq \prod_{i=1}^{l} L(\phi_i). \tag{31}$$

As such, for a multi-layer neural network that comprises repeated layers of linear operation followed by non-linear activation, we can upper bound the change in model output with respect to the change in the input space by multiplying the operator norms of the weights. It is important to realize that (31) is not a tight upper bound, and in fact, computing the exact Lipschitz constant of the neural network is NP-hard (Virmaux & Scaman, 2018). Nonetheless, this approach allows us to draw connections between the weight and the robustness of the model in the context of neural networks.

Results in Sec. 4 indicate that linear models trained by signGD have larger weight norms, indicating less robustness. Therefore, we expect in the deep learning setting that neural networks trained by SGD are more robust, as they have a smaller Lipschitz upper bound, as shown in Figure 1. To verify this, we follow the techniques in Gouk et al. (2021) and compute an upper bound on the Lipschitz constant of the same neural networks trained by SGD, Adam, and RMSProp in Figure 1. Results are shown in Table 1. The result shows that across all datasets and architectures, models trained by SGD have a smaller upper bound on the Lipschitz constant compared to models trained by the two adaptive gradient methods. In Figure 1, we demonstrate the robustness of the neural networks under Gaussian noise, $\ell_2$ and $\ell_\infty$ bounded adversarial perturbations (Croce & Hein, 2020). In Table 1, we average the accuracy across the perturbations and get a single score quantifying the model's robustness. We observe that a smaller upper bound on the Lipschitz constant of a neural network implies better robustness against perturbations.

## 6 Conclusions

In this paper, we highlighted the robustness difference between models trained by SGD and adaptive gradient methods, particularly Adam and RMSProp. To understand this phenomenon, we leveraged a frequency-domain analysis, and demonstrated that natural datasets contain frequencies that are irrelevant to minimizing the standard training loss. Empirically, through a band-limited perturbation analysis on neural networks trained on common vision datasets, we showed that models trained by the adaptive gradient method utilize the statistics in the irrelevant frequency, and thus they experience a huge drop in performance when the same statistics become corrupted. Analytically, on a synthetic linear regression task where the dataset was designed to contain target-irrelevant frequencies, we showed that while both GD and signGD can find the solution with standard risks close to zero, the adversarial risk of the asymptotic solution found by signGD can be larger than that of GD. Such results from the linear analysis explained the observation in Figure 1 and suggested that a smaller model parameters' weight norms may indicate a larger model robustness. Finally, in the deep learning setting, we showed that models trained by SGD have a noticeably smaller Lipschitz constant than those trained by Adam and RMSProp.

Our work has some limitations. First, when conducting a theoretical analysis of various optimizers, we opted for signGD as a simpler alternative to Adam and RMSProp. Second, our focus was primarily on linear models. However, it is crucial to acknowledge that deep neural networks inherently possess non-linear characteristics, which limit the depth of insights derived from linear models. Therefore, one promising future direction is to incorporate tools such as neural tangent kernels (Jacot et al., 2018), which provide a deeper understanding of network dynamics. Third, our analysis focuses on optimization algorithms along with the standard objective function. We can also study the effect of optimizer with alternative objectives that are designed to improve the robustness of the model (Simon-Gabriel et al., 2019; Wen et al., 2020; Ma et al., 2020; Foret et al., 2021). For instance, the effect of adversarial training using perturbations similar to the Fast Gradient Sign Method (FGSM) under the linear regression setup has been studied by Ma et al. (2020). In

linear classification, Wei et al. (2023) showed that minimizing the sharpness-aware loss (SAM) (Foret et al., 2021) can lead to robust models. Further discussions on this study can be found in Appendix F. Another promising direction for future research is to analyze model robustness by coupling various optimization algorithms with different optimization objectives.

### Acknowledgments

Avery Ma acknowledges the funding from the Natural Sciences and Engineering Research Council (NSERC) through the Canada Graduate Scholarships – Doctoral (CGS D) program. Amir-massoud Farahmand acknowledges the funding from the CIFAR AI Chairs program, as well as the support of the NSERC through the Discovery Grant program (2021-03701). Yangchen Pan acknowledges the support from the Turing AI World Leading Fellow. Resources used in preparing this research were provided, in part, by the Province of Ontario, the Government of Canada through CIFAR, and companies sponsoring the Vector Institute. We would like to also thank the members of the Adaptive Agents Lab who provided feedback on a draft of this paper.

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

## A  Summary of the Supplementary Material

The supplementary material is organized as follows. In Appendix B, we first describe the data augmentation, the exact optimization schedule, and the model architectures used to train the models. In Appendix C, we describe the complete generalization and robustness results in Table 3 and how they are used to generate Figure 1. In Appendix D, we discuss how the training inputs are modified when making the observations in Sec. 3. In Appendix E, we provide additional detail on the linear analysis in Sec. 4. In Appendix F, we discuss how our analysis relates to the sharpness-aware minimization (SAM). Finally, in Appendix G, we provide additional figures including visualization of the perturbed images, the modified images used in Sec. 3.1, and the frequency sensitivity comparison in Sec. 3.2.

## B  Training Details

**Data augmentation:** In our paper, we study how the presence of irrelevant information in the dataset affects the robustness of the model when trained by different algorithms. We approach this problem from a frequency-domain perspective. Specifically, we leverage the structure and energy profile of the dataset in the frequency domain. While data augmentation methods are widely used in training machine learning models to improve generalization and reduce overfitting, understanding how those methods affect the datasets in the frequency domain requires additional analysis tailored for each augmentation method. Therefore, on FashionMNIST, CIFAR10, CIFAR100, Caltech101, and Imagenette, training inputs are augmented with random horizontal flipping, a method that does not change the frequency profile of the image.

**Optimization schedule:** For all models, we use the following default PyTorch (v1.12.1) optimization settings. For SGD, we disable all of the following mechanism: dampening, weight decay, and Nesterov. For Adam, we use the default values of $\beta_1 = 0.9$ $\beta_2 = 0.999$, $\epsilon = 10^{-8}$ and disable weight decay and disable AMSgrad (Reddi et al., 2018). For RMSProp, we use default values of $\alpha = 0.99$, $\epsilon = 10^{-8}$, and disable momentum and disable centered RMSProp which normalizes the gradient by an estimation of its variance. All models are trained for 200 epochs. In Table 2, we list the initial learning rate. The learning rate decreases by a factor of 0.1 at epoch 100 and 150.

Table 2: **Experiment setup:** the initial learning rate and the definition of neural networks in this paper.

| Dataset | Optimization | Initial Learning Rate |
|---|---|---|
| MNIST | SGD | 0.1 |
| | Adam | 0.0005 |
| | RMSProp | 0.0005 |
| FashionMNIST | SGD | 0.1 |
| | Adam | 0.0005 |
| | RMSProp | 0.0005 |
| CIFAR10 | SGD | 0.2 |
| | Adam | 0.0002 |
| | RMSProp | 0.0005 |
| CIFAR100 | SGD | 0.3 |
| | Adam | 0.0005 |
| | RMSProp | 0.0005 |
| SVHN | SGD | 0.2 |
| | Adam | 0.0002 |
| | RMSProp | 0.0002 |
| Caltech101 | SGD | 0.05 |
| | Adam | 0.0002 |
| | RMSProp | 0.001 |
| Imagenette | SGD | 0.1 |
| | Adam | 0.0002 |
| | RMSProp | 0.0002 |
| Speech Commmands | SGD | 0.1 |
| | Adam | 0.1 |
| | RMSProp | 0.1 |

| Dataset | Structure |
|---|---|
| MNIST FashionMnist | Conv(1, 16, 4) - ReLU - Conv(16, 32, 4) - ReLU - FN(21632, 100) - FN(100, 10) - SM(10) |
| CIFAR10 CIFAR100 SVHN Caltech101 Imagenette | PreActResNet18 (He et al., 2016) ViT-B/16 (Dosovitskiy et al., 2021) |
| Speech Commands | M5 (Dai et al., 2017) |

**Model architecture:** For MNIST and FashionMNIST, we use a ReLU-activated, two-layer convolutional neural network ending with two fully-connected layers. For CIFAR10, CIFAR100, SVHN, Caltech101, and Imagenette, we use PreActResNet18 (He et al., 2016) and Vision Transformers (ViT-B/16) (Dosovitskiy et al., 2021). For the Speech Commands dataset (Warden, 2018), we use the M5 network architecture

defined by Dai et al. (2017). See Table 2 for details of all architectures used in this paper. We denote Conv($i$, $o$, $k$) as a convolution layer having $i$ input channels, $o$ output channels with $k$ by $k$ filters, FN($i$, $o$) as a fully-connect layer with $i$ input channels and $o$ output channels, and SM($o$) as the soft-max layer with $o$ output. The stride for all convolution layers is 1. The main experiments in our work are centered around models based on convolutional neural networks, within the computer vision domain. Additional results from using ViT-B/16 and on the Speech Commands dataset can be found in Table 5 and Table 6, respectively.

**Batch normalization:** We concentrate on a particular aspect of the training process: the selection of optimizers. Our aim is to shed light on how this critical component influences the robustness of trained models. It has been recently shown that the use of batch normalization (BN) can also affect the robustness of the model (Benz et al., 2021b;a; Wang et al., 2022). Consequently, to maintain focus on the impact of optimizers, we have omitted BN in the training phase for the experiments leading to the results in Figure 1 and the analysis in Sec. 3. However, to show that our conclusions remain valid for models with BN, we have included additional results that incorporate BN in Table 4.

## C    Results on Standard Generalization and Model Robustness

**Main results:** Table 3 summarizes the result on the standard generalization ability and robustness properties of the models trained by SGD, Adam, and RMSProp on seven vision datasets. All results are averaged over three independently initialized and trained models. To evaluate standard generalization, we measure the classification accuracy of the models on the testing data. To capture model robustness, we measure the classification accuracy of the models on the testing data perturbed using Gaussian perturbations, $\ell_2$ and $\ell_\infty$-bounded perturbations (Croce & Hein, 2020). Perturbations with varying degrees of severity are included in the evaluation to ensure the observation of the robustness difference is not limited to perturbations with any particular parameters. The degree of severity is determined by the variance of the Gaussian perturbation and an $\ell_2$ and $\ell_\infty$ norm for the attacks. We select those parameters so the range of the accuracy differences between models is similar across different datasets. Particularly, the highlighted results in Table 3 are in a similar range, so we use them to plot Figure 1. We also ensure that the original image semantics remains in the perturbed images, and we provide a visualization of the perturbed images in Figure 17 to 19.

Finally, for CIFAR100 and Caltech101, because of the large number of classes in the dataset, we use the top-5 classification accuracy to plot Figure 1 as the results are within a range similar to other datasets with 10 classes. The observation of the similar standard generalization and different robustness holds on both top-1 and top-5 accuracy.

### C.1    Results on Models with Batch Normalization Enabled

When BN layers are activated in PreActResNet18, we observe that the models exhibit similar standard generalization performance, yet the robustness difference between SGD and adaptive gradient methods remains evident. This observation is in line with the results presented in Table 3, where BN layers are disabled. Notably, the accuracy of models with BN enabled is significantly lower compared to their BN-disabled counterparts under almost all types of perturbations, particularly under stronger perturbations. This finding aligns with the results from the previous work (Benz et al., 2021b;a; Wang et al., 2022).

### C.2    Results on Vision Transformers

In addition to the network designs considered in Table 2, we extend our work to ViT in order to verify whether similar observations can be drawn on other neural network architectures.

It is important to note that the dataset utilized in our paper is significantly smaller in size compared to larger datasets such as Imagenet and JFT-300M. Recent research, such as the work by Zhu et al. (2023), has shown that ViT tends to generalize poorly on small datasets when trained from scratch. In particular, Zhu et al. (2023) empirically demonstrated that ViT and ResNet learn distinct representations on small datasets while converging to similar representations on larger datasets.

Table 3: **Results on standard generalization and robustness of models trained by SGD, Adam, and RMSProp.** We evaluate the model robustness on the testing data perturbed using Gaussian perturbations, $\ell_2$ and $\ell_\infty$-bounded perturbations (Croce & Hein, 2020). We include various severity of perturbations to better capture the model robustness. Models trained by SGD are the most robust against the three types of perturbations across all datasets. The highlighted results are used in Figure 1, as they are in relatively similar ranges. Results are averaged over three independently initialized and trained models.

| Dataset | Optimization | Test | Gaussian perturbations | | | $\ell_2$-bounded attack | | | $\ell_\infty$-bounded attack | | |
|---|---|---|---|---|---|---|---|---|---|---|---|
| | | | $\sigma^2=0.01$ | $\sigma^2=0.05$ | $\sigma^2=0.1$ | $\epsilon=0.5$ | $\epsilon=0.7$ | $\epsilon=1.0$ | $\epsilon=0.05$ | $\epsilon=0.07$ | $\epsilon=0.1$ |
| MNIST | SGD | 98.72 | 98.59 | 97.94 | **95.64** | 93.00 | 87.33 | **66.33** | 87.53 | **71.93** | 31.50 |
| | Adam | 99.05 | 98.86 | 96.76 | **89.33** | 92.33 | 86.00 | **54.67** | 85.40 | **52.93** | 8.77 |
| | RMSProp | 98.90 | 98.70 | 97.02 | **90.63** | 91.67 | 83.00 | **50.00** | 82.73 | **50.00** | 7.33 |
| | | | $\sigma^2=0.001$ | $\sigma^2=0.005$ | $\sigma^2=0.01$ | $\epsilon=0.1$ | $\epsilon=0.5$ | $\epsilon=0.7$ | $\epsilon=0.01$ | $\epsilon=0.03$ | $\epsilon=0.05$ |
| FashionMNIST | SGD | 91.20 | 90.49 | 87.78 | **84.30** | **82.33** | 24.33 | 12.33 | **67.23** | 22.30 | 4.17 |
| | Adam | 90.98 | 89.91 | 85.03 | **78.91** | **70.33** | 6.00 | 0.33 | **53.57** | 5.33 | 0.00 |
| | RMSProp | 91.15 | 90.09 | 85.23 | **78.69** | **72.67** | 15.93 | 4.67 | **62.67** | 16.33 | 1.67 |
| | | | $\sigma^2=0.001$ | $\sigma^2=0.005$ | $\sigma^2=0.007$ | $\epsilon=0.1$ | $\epsilon=0.2$ | $\epsilon=0.3$ | $\epsilon=\frac{1}{255}$ | $\epsilon=\frac{2}{255}$ | $\epsilon=\frac{4}{255}$ |
| CIFAR10 | SGD | 90.16 | 87.35 | 74.13 | **68.66** | **67.40** | 37.57 | 16.30 | **53.93** | 21.27 | 0.93 |
| | Adam | 90.73 | 86.93 | 67.50 | **58.54** | **64.03** | 29.60 | 11.10 | **50.57** | 13.93 | 0.20 |
| | RMSProp | 90.46 | 86.25 | 70.03 | **61.52** | **60.10** | 25.47 | 8.87 | **47.87** | 14.03 | 0.33 |
| | | | $\sigma^2=0.001$ | $\sigma^2=0.005$ | $\sigma^2=0.007$ | $\epsilon=0.1$ | $\epsilon=0.2$ | $\epsilon=0.3$ | $\epsilon=\frac{1}{255}$ | $\epsilon=\frac{2}{255}$ | $\epsilon=\frac{4}{255}$ |
| CIFAR100(top1) | SGD | 59.76 | 56.88 | 46.26 | 41.28 | 28.47 | 12.93 | 5.90 | 18.80 | 5.57 | 1.17 |
| | Adam | 61.10 | 55.30 | 31.54 | 24.92 | 24.30 | 7.13 | 1.87 | 14.43 | 2.47 | 0.33 |
| | RMSProp | 60.36 | 56.46 | 36.42 | 29.50 | 28.90 | 10.47 | 2.83 | 17.90 | 3.47 | 0.20 |
| CIFAR100(top5) | SGD | 84.67 | 81.77 | 72.84 | **67.53** | 80.30 | 70.20 | **59.90** | 75.30 | 61.70 | **39.53** |
| | Adam | 85.41 | 81.37 | 58.11 | **49.54** | 80.53 | 66.73 | **52.77** | 74.70 | 53.70 | **33.43** |
| | RMSProp | 85.18 | 81.66 | 62.71 | **54.44** | 80.93 | 68.57 | **54.10** | 74.60 | 59.10 | **34.10** |
| | | | $\sigma^2=0.001$ | $\sigma^2=0.003$ | $\sigma^2=0.005$ | $\epsilon=0.1$ | $\epsilon=0.2$ | $\epsilon=0.3$ | $\epsilon=\frac{1}{255}$ | $\epsilon=\frac{2}{255}$ | $\epsilon=\frac{4}{255}$ |
| SVHN | SGD | 96.11 | 95.68 | 94.47 | **93.81** | 85.53 | **63.60** | 39.63 | 80.67 | **49.83** | 17.03 |
| | Adam | 96.48 | 96.03 | 94.04 | **91.46** | 80.77 | **57.83** | 35.93 | 78.93 | **47.50** | 12.43 |
| | RMSProp | 96.42 | 95.91 | 94.07 | **91.87** | 81.13 | **57.90** | 34.10 | 76.93 | **46.33** | 11.30 |
| | | | $\sigma^2=0.01$ | $\sigma^2=0.05$ | $\sigma^2=0.1$ | $\epsilon=0.5$ | $\epsilon=1.0$ | $\epsilon=1.5$ | $\epsilon=\frac{1}{255}$ | $\epsilon=\frac{2}{255}$ | $\epsilon=\frac{4}{255}$ |
| Caltech101(top1) | SGD | 70.80 | 68.13 | 57.67 | 43.46 | 58.77 | 47.03 | 35.17 | 48.27 | 27.87 | 4.70 |
| | Adam | 72.32 | 58.34 | 19.88 | 8.55 | 57.70 | 44.47 | 29.60 | 45.47 | 22.03 | 2.63 |
| | RMSProp | 73.82 | 69.38 | 51.34 | 33.17 | 37.80 | 11.30 | 2.80 | 14.77 | 1.93 | 0.03 |
| Caltech101(top5) | SGD | 85.96 | 84.84 | **77.57** | 67.16 | 85.30 | **84.43** | 79.00 | 83.97 | 75.47 | **52.27** |
| | Adam | 88.08 | 79.48 | **38.55** | 19.67 | 82.03 | **80.67** | 77.63 | 83.97 | 72.07 | **45.87** |
| | RMSProp | 88.37 | 85.90 | **72.19** | 52.74 | 80.20 | **63.40** | 45.93 | 63.97 | 41.33 | **23.90** |
| | | | $\sigma^2=0.01$ | $\sigma^2=0.05$ | $\sigma^2=0.1$ | $\epsilon=0.5$ | $\epsilon=1.0$ | $\epsilon=1.5$ | $\epsilon=\frac{1}{255}$ | $\epsilon=\frac{2}{255}$ | $\epsilon=\frac{4}{255}$ |
| Imagenette | SGD | 89.44 | **84.83** | 67.23 | 50.21 | **70.70** | 44.13 | 21.33 | **47.23** | 14.33 | 0.37 |
| | Adam | 89.75 | **71.84** | 29.05 | 17.72 | **65.30** | 31.53 | 11.83 | **39.43** | 6.23 | 0.07 |
| | RMSProp | 89.77 | **73.25** | 28.28 | 17.16 | **62.30** | 30.27 | 11.27 | **38.40** | 6.20 | 0.03 |

Therefore, we perform fine-tuning on a pre-trained ViT-B/16. Among the datasets we considered, Imagenette is a 10-class subset of the Imagenet-1k dataset, making it especially suitable for the fine-tuning task, since the publicly available ViT checkpoint was pre-trained on Imagenet-1k. Also, it is important to note that the pretrained models were originally trained using Adam. In our fine-tuning process, we treat ViT as a feature extractor (i.e., no weight update on the transformer encoder), with a focus on fine-tuning the Multi-Layer Perceptrons (MLP) head. Our approach follows prior work (Steiner et al., 2022) and incorporates the three different optimizers, each fine-tuned for 10 epochs. We initiated the fine-tuning process with an initial learning rate of 0.01, followed by a cosine decay learning rate schedule and a linear warmup. Throughout this process, we maintained a fixed batch size of 512.

To evaluate the robustness of the fine-tuned models, we maintained the exact same perturbation strengths, including the variance of Gaussian noise and $\epsilon$ for adversarial perturbations, as used in Table 3. The results can be found in Table 5. We draw three observations.

Table 4: **Results on standard generalization and robustness of models trained with BN enabled.** We follow the exact optimization configuration as the ones used in generating Table 3. The only modification is that **BN is enabled**.

| Dataset | Optimization | Test | Gaussian perturbations | | | $\ell_2$-bounded attack | | | $\ell_\infty$-bounded attack | | |
|---|---|---|---|---|---|---|---|---|---|---|---|
| | | | $\sigma^2 = 0.001$ | $\sigma^2 = 0.005$ | $\sigma^2 = 0.007$ | $\epsilon = 0.1$ | $\epsilon = 0.2$ | $\epsilon = 0.3$ | $\epsilon = \frac{1}{255}$ | $\epsilon = \frac{2}{255}$ | $\epsilon = \frac{4}{255}$ |
| CIFAR10 | SGD | 92.24 | 86.91 | 55.23 | 43.29 | 64.84 | 27.86 | 6.96 | 48.64 | 9.14 | 0.04 |
| | Adam | 93.38 | 85.67 | 50.41 | 39.11 | 56.5 | 15.96 | 2.63 | 37.7 | 3.8 | 0 |
| | RMSProp | 93.57 | 86.97 | 52.09 | 39.86 | 55.93 | 15.16 | 2.23 | 37.7 | 3.76 | 0 |
| | | | $\sigma^2 = 0.001$ | $\sigma^2 = 0.005$ | $\sigma^2 = 0.007$ | $\epsilon = 0.1$ | $\epsilon = 0.2$ | $\epsilon = 0.3$ | $\epsilon = \frac{1}{255}$ | $\epsilon = \frac{2}{255}$ | $\epsilon = \frac{4}{255}$ |
| CIFAR100 | SGD | 72.24 | 57.99 | 26.08 | 19.23 | 34.36 | 10.2 | 3.63 | 21.3 | 4.8 | 0.2 |
| | Adam | 71.36 | 55.85 | 24.55 | 18.23 | 27.56 | 6.66 | 1.73 | 15.26 | 2.9 | 0.23 |
| | RMSProp | 70.99 | 56.13 | 24.56 | 18.03 | 23.8 | 5.1 | 1.1 | 13.7 | 2 | 0.1 |
| | | | $\sigma^2 = 0.001$ | $\sigma^2 = 0.003$ | $\sigma^2 = 0.005$ | $\epsilon = 0.1$ | $\epsilon = 0.2$ | $\epsilon = 0.3$ | $\epsilon = \frac{1}{255}$ | $\epsilon = \frac{2}{255}$ | $\epsilon = \frac{4}{255}$ |
| SVHN | SGD | 94.16 | 95.81 | 94.96 | 92.90 | 81.73 | 60.56 | 41.96 | 76.13 | 49.26 | 15.3 |
| | Adam | 96.62 | 96.09 | 93.96 | 91.19 | 80.13 | 47.76 | 21.86 | 72.36 | 32.76 | 4.36 |
| | RMSProp | 96.44 | 94.86 | 93.75 | 91.02 | 79.86 | 48.43 | 22.16 | 72.36 | 33.7 | 3.96 |
| | | | $\sigma^2 = 0.01$ | $\sigma^2 = 0.05$ | $\sigma^2 = 0.1$ | $\epsilon = 0.5$ | $\epsilon = 1.0$ | $\epsilon = 1.5$ | $\epsilon = \frac{1}{255}$ | $\epsilon = \frac{2}{255}$ | $\epsilon = \frac{4}{255}$ |
| Caltech101 | SGD | 78.61 | 72.69 | 45.38 | 25.87 | 61.23 | 44.03 | 23.03 | 46.80 | 13.13 | 0.83 |
| | Adam | 79.58 | 62.21 | 21.08 | 10.35 | 56.76 | 34.46 | 13.06 | 37.3 | 5.66 | 0.23 |
| | RMSProp | 75.56 | 69.89 | 45.38 | 23.13 | 58.6 | 44.6 | 20.5 | 42.6 | 11.3 | 0.6 |
| | | | $\sigma^2 = 0.01$ | $\sigma^2 = 0.05$ | $\sigma^2 = 0.1$ | $\epsilon = 0.5$ | $\epsilon = 1.0$ | $\epsilon = 1.5$ | $\epsilon = \frac{1}{255}$ | $\epsilon = \frac{2}{255}$ | $\epsilon = \frac{4}{255}$ |
| Imagenette | SGD | 89.35 | 76.67 | 46.18 | 28.27 | 73.43 | 42.96 | 17.56 | 48.93 | 14.33 | 0.03 |
| | Adam | 91.88 | 67.29 | 24.30 | 14.92 | 67.66 | 24.06 | 3.06 | 31.4 | 6.23 | 0 |
| | RMSProp | 91.93 | 67.02 | 23.79 | 15.43 | 68.76 | 26.1 | 4.1 | 33.66 | 6.20 | 0 |

Table 5: **Results on standard generalization and robustness of ViT-B/16 fined-tuned on the Imagenette dataset.**

| Model | Optimization | Test | Gaussian perturbations | | | $\ell_2$-bounded attack | | | $\ell_\infty$-bounded attack | | |
|---|---|---|---|---|---|---|---|---|---|---|---|
| | | | $\sigma^2 = 0.01$ | $\sigma^2 = 0.05$ | $\sigma^2 = 0.1$ | $\epsilon = 0.5$ | $\epsilon = 1.0$ | $\epsilon = 1.5$ | $\epsilon = \frac{1}{255}$ | $\epsilon = \frac{2}{255}$ | $\epsilon = \frac{4}{255}$ |
| ViT-b/16 | SGD | 99.18 | 99.05 | 96.16 | 91.25 | 6.2 | 0 | 0 | 1.3 | 0 | 0 |
| | Adam | 99.93 | 99.51 | 94.43 | 88.59 | 5.1 | 0 | 0 | 0.7 | 0 | 0 |
| | RMSProp | 99.92 | 99.54 | 95.38 | 89.91 | 5 | 0 | 0 | 0.8 | 0 | 0 |

First, all models fine-tuned with the three different optimizers achieve near 100% test accuracy, a substantial improvement from the 89% accuracy when training from scratch using PreActResNet18. This significant boost in standard generalization highlights the effectiveness of fine-tuning with ViT. Second, we observe that the fine-tuned models exhibit a notable increase in robustness to Gaussian noise. However, they are highly vulnerable to adversarial perturbations. This observation is consistent with the results from existing literature (Zhang et al., 2019), where a trade-off is often present between standard accuracy and adversarial robustness. Finally, we make a similar observation on the robustness difference between models fine-tuned with the three optimizers, where models fine-tuned with SGD exhibited greater robustness to both Gaussian noise and adversarial perturbations when compared to models fine-tuned using Adam and RMSProp.

## C.3 Results with an Audio Dataset

Besides the vision domain, we extend our work to the audio domain since audio signals offer a frequency-based interpretation as well. We include additional results in Table 6, which compare the standard generalization and robustness properties of an audio classifier trained on the Speech Commands dataset (Warden, 2018). We focus on the PreActResNet18 architectures and all models are trained for 200 epochs, with an initial learning rate of 0.1 and learning rate decay by a factor of 0.1 at epoch 100 and 150. We consider the accuracy of

Table 6: **Results on standard generalization and robustness of models on an audio classification task on the Speech Commands dataset.**

| Dataset | Optimization | Test | Gaussian perturbations | | | $\ell_2$-bounded attack | | | $\ell_\infty$-bounded attack | | |
|---|---|---|---|---|---|---|---|---|---|---|---|
| | | | $\sigma^2 = 0.001$ | $\sigma^2 = 0.003$ | $\sigma^2 = 0.005$ | $\epsilon = 0.01$ | $\epsilon = 0.05$ | $\epsilon = 0.1$ | $\epsilon = 0.0001$ | $\epsilon = 0.0005$ | $\epsilon = 0.001$ |
| Speech Commands | SGD | 85.14 | 55.76 | 39.88 | 33.42 | 70.01 | 20.31 | 9.81 | 75.6 | 36.71 | 13.47 |
| | Adam | 85.47 | 54.73 | 38.87 | 31.95 | 60.74 | 17.87 | 8.49 | 71.97 | 29.2 | 10.15 |
| | RMSProp | 84.67 | 52.37 | 36.59 | 27.94 | 59.57 | 19.04 | 8.88 | 70.50 | 30.95 | 11.01 |

Table 7: **Results on standard generalization and robustness of models trained by SGD without and with momentum (0.9).**

| Dataset | Optimization | Test | Gaussian perturbations | | | $\ell_2$-bounded attack | | | $\ell_\infty$-bounded attack | | |
|---|---|---|---|---|---|---|---|---|---|---|---|
| | | | $\sigma^2 = 0.001$ | $\sigma^2 = 0.005$ | $\sigma^2 = 0.007$ | $\epsilon = 0.1$ | $\epsilon = 0.2$ | $\epsilon = 0.3$ | $\epsilon = \frac{1}{255}$ | $\epsilon = \frac{2}{255}$ | $\epsilon = \frac{4}{255}$ |
| CIFAR10 | SGD | 90.16 | 87.35 | 74.13 | 68.66 | 67.40 | 37.57 | 16.30 | 53.93 | 21.27 | 0.93 |
| | SGD-m | 89.79 | 87.28 | 73.207 | 66.783 | 67.1 | 38.067 | 15.9 | 55.667 | 20.233 | 0.6333 |
| | | | $\sigma^2 = 0.001$ | $\sigma^2 = 0.005$ | $\sigma^2 = 0.007$ | $\epsilon = 0.1$ | $\epsilon = 0.2$ | $\epsilon = 0.3$ | $\epsilon = \frac{1}{255}$ | $\epsilon = \frac{2}{255}$ | $\epsilon = \frac{4}{255}$ |
| CIFAR100 | SGD | 59.76 | 56.88 | 46.26 | 41.28 | 28.47 | 12.93 | 5.90 | 18.80 | 5.57 | 1.17 |
| | SGD-m | 56.08 | 55.03 | 44.97 | 40.03 | 29.2 | 12.4 | 5.7 | 19.7 | 6.5 | 0.8 |
| | | | $\sigma^2 = 0.001$ | $\sigma^2 = 0.003$ | $\sigma^2 = 0.005$ | $\epsilon = 0.1$ | $\epsilon = 0.2$ | $\epsilon = 0.3$ | $\epsilon = \frac{1}{255}$ | $\epsilon = \frac{2}{255}$ | $\epsilon = \frac{4}{255}$ |
| SVHN | SGD | 96.11 | 95.68 | 94.47 | 93.81 | 85.53 | 63.60 | 39.63 | 80.67 | 49.83 | 17.03 |
| | SGD-m | 96.14 | 95.71 | 94.44 | 92.80 | 82.7 | 60.53 | 39.03 | 77.33 | 49.26 | 15.96 |
| | | | $\sigma^2 = 0.01$ | $\sigma^2 = 0.05$ | $\sigma^2 = 0.1$ | $\epsilon = 0.5$ | $\epsilon = 1.0$ | $\epsilon = 1.5$ | $\epsilon = \frac{1}{255}$ | $\epsilon = \frac{2}{255}$ | $\epsilon = \frac{4}{255}$ |
| Caltech101 | SGD | 70.80 | 68.13 | 57.67 | 43.46 | 58.77 | 47.03 | 35.17 | 48.27 | 27.87 | 4.70 |
| | SGD-m | 69.89 | 67.77 | 55.77 | 41.34 | 54.86 | 43.36 | 31.06 | 44.16 | 23.9 | 3.133 |
| | | | $\sigma^2 = 0.01$ | $\sigma^2 = 0.05$ | $\sigma^2 = 0.1$ | $\epsilon = 0.5$ | $\epsilon = 1.0$ | $\epsilon = 1.5$ | $\epsilon = \frac{1}{255}$ | $\epsilon = \frac{2}{255}$ | $\epsilon = \frac{4}{255}$ |
| Imagenette | SGD | 89.44 | 84.83 | 67.23 | 50.21 | 70.70 | 44.13 | 21.33 | 47.23 | 14.33 | 0.37 |
| | SGD-m | 88.69 | 85.05 | 68.58 | 51.15 | 75.53 | 49.2 | 24.53 | 56.43 | 17.76 | 0.2 |

models under Gaussian- and adversarially-perturbed test sets. Manual verification was conducted to ensure that the noisy audio phrase could still be recognizable.

Results demonstrate that despite similar test accuracy, the models trained using SGD exhibit greater robustness when compared to the other two optimization methods. These insights provide valuable context to the generalizability of our initial observations, offering a more comprehensive understanding of how optimizers perform in the context of different data modalities.

### C.4 Results with Momentum-enabled SGD

Additional results with momentum-enabled SGD (SGD-m) are included in Table 7. We maintain the exact same optimization configuration as that is used for generating the SGD results presented in Table 3, and the only variation is an additional momentum term with a coefficient of $\beta = 0.9$. The result shows that models optimized by both vanilla SGD and SGD-m exhibit similar trends in terms of generalization and robustness.

## D Modifying the Training Inputs in Sec.3.1

We demonstrate irrelevant frequencies in two settings: i) DCT basis vectors with a low magnitude are irrelevant and ii) high-frequency DCT bases are irrelevant. In Figure 9 to 15, we visualize the original images and the modified images used in Sec. 3.1.

To understand how the modified training images are generated, we use $\Phi_{\mathrm{nrg}}(x, p)$ with $0 < p < 100$ to denote the operation that modifies the input image $x$ by removing the DCT basis vectors whose magnitudes are in the bottom $\frac{p}{100}$-th percentile. We use $M_{\mathrm{nrg}}(x, p)$ to denote the binary mask used in the process. Consider an image $x \in \mathbb{R}^{d \times d}$, the entire process can be formulated as

$$\Phi_{\mathrm{nrg}}(x, p) = C(\tilde{x} \odot M_{\mathrm{nrg}}(x, p))C^\top,$$

Table 8: **Examples of the synthetic data distribution in the frequency and the spatial domain.**

| $\tilde{\Sigma}$ | Frequency-domain Representation | Spatial-domain Representation |
|---|---|---|
| $\mathrm{diag}\left\{\tilde{\sigma}_0^2, 0, 0\right\}$ | $(\tilde{X}_0, 0, 0)$ | $(\sqrt{\frac{1}{3}}\tilde{X}_0, \sqrt{\frac{1}{3}}\tilde{X}_0, \sqrt{\frac{1}{3}}\tilde{X}_0)$ |
| $\mathrm{diag}\left\{0, \tilde{\sigma}_1^2, 0\right\}$ | $(0, \tilde{X}_1, 0)$ | $(\sqrt{\frac{1}{2}}\tilde{X}_1, 0, -\sqrt{\frac{1}{2}}\tilde{X}_1)$ |
| $\mathrm{diag}\left\{0, 0, \tilde{\sigma}_2^2\right\}$ | $(0, 0, \tilde{X}_2)$ | $(\sqrt{\frac{1}{6}}\tilde{X}_2, -\sqrt{\frac{2}{3}}\tilde{X}_2, +\sqrt{\frac{1}{6}}\tilde{X}_2)$ |
| $\mathrm{diag}\left\{0, \tilde{\sigma}_1^2, \tilde{\sigma}_2^2\right\}$ | $(0, \tilde{X}_1, \tilde{X}_2)$ | $(\sqrt{\frac{1}{2}}\tilde{X}_1 + \sqrt{\frac{1}{6}}\tilde{X}_2, -\sqrt{\frac{2}{3}}\tilde{X}_2, -\sqrt{\frac{1}{2}}\tilde{X}_1 + \sqrt{\frac{1}{6}}\tilde{X}_2)$ |
| $\mathrm{diag}\left\{\tilde{\sigma}_0^2, 0, \tilde{\sigma}_2^2\right\}$ | $(\tilde{X}_0, 0, \tilde{X}_2)$ | $(\sqrt{\frac{1}{3}}\tilde{X}_0 + \sqrt{\frac{1}{6}}\tilde{X}_2, \sqrt{\frac{1}{3}}\tilde{X}_0 - \sqrt{\frac{2}{3}}\tilde{X}_2, \sqrt{\frac{1}{3}}\tilde{X}_0 + \sqrt{\frac{1}{6}}\tilde{X}_2)$ |
| $\mathrm{diag}\left\{\tilde{\sigma}_0^2, \tilde{\sigma}_0^2, 0\right\}$ | $(\tilde{X}_0, \tilde{X}_1, 0)$ | $(\sqrt{\frac{1}{3}}\tilde{X}_0 + \sqrt{\frac{1}{2}}\tilde{X}_1, \sqrt{\frac{1}{3}}\tilde{X}_0, \sqrt{\frac{1}{3}}\tilde{X}_0 - \sqrt{\frac{1}{2}}\tilde{X}_1)$ |
| $\mathrm{diag}\left\{\tilde{\sigma}_0^2, \tilde{\sigma}_1^2, \tilde{\sigma}_2^2\right\}$ | $(\tilde{X}_0, \tilde{X}_1, \tilde{X}_2)$ | $(\sqrt{\frac{1}{3}}\tilde{X}_0 + \sqrt{\frac{1}{2}}\tilde{X}_1 + \sqrt{\frac{1}{6}}\tilde{X}_2, \sqrt{\frac{1}{3}}\tilde{X}_0 - \sqrt{\frac{2}{3}}\tilde{X}_2, \sqrt{\frac{1}{3}}\tilde{X}_0 - \sqrt{\frac{1}{2}}\tilde{X}_1 + \sqrt{\frac{1}{6}}\tilde{X}_2)$ |

where $\odot$ is the element-wise product and $C$ is the DCT transformation matrix. The binary mask $M_{\mathrm{nrg}} \in \{0, 1\}^{d \times d}$ is defined as

$$M_{\mathrm{nrg}}(x, p) = \begin{cases} 1 & \text{if} \quad |\tilde{x}_{i,j}| > \phi(\tilde{x}, p) \\ 0 & \text{otherwise,} \end{cases}$$

where $\phi(\tilde{x}, p) \in \mathbb{R}$ computes the $\frac{p}{100}$-th percentile in $|\tilde{x}|$. Therefore, DCT basis vectors with a magnitude smaller than the threshold are first discarded in $\tilde{x}$, and then this filtered $\tilde{x}$ is converted back to the spatial domain.

Similarly, we use $\Phi_{\mathrm{freq}}(x, p)$ to denote the operation that modifies the input image $x$ by removing the DCT basis vectors whose frequency are in the highest $\frac{p}{100}$-th percentile. We use $M_{\mathrm{freq}}(p)$ to denote the binary mask used in the process. This operation can be formulated as

$$\Phi_{\mathrm{freq}}(x, p) = C(\tilde{x} \odot M_{\mathrm{freq}}(p))C^\top,$$

where $M_{\mathrm{freq}} \in \{0, 1\}^{d \times d}$ is defined as:

$$M_{\mathrm{freq}}(p) = \begin{cases} 1 & \text{if} \quad i^2 + j^2 > \frac{p}{100}\sqrt{2}d \\ 0 & \text{otherwise,} \end{cases}$$

and $i, j$ are frequency bases. Notice that $M_{\mathrm{freq}}$ only depends on the size of the image, whereas $M_{\mathrm{nrg}}$ depends on the input $x$ since we identify the threshold value in $|\tilde{x}|$. Examples of the modified images and the modification process are shown in Appendix G.

# E Linear Regression Analysis

## E.1 Understanding the Synthetic Dataset

The goal of the linear analysis is to study the learning dynamics of different algorithms on a synthetic dataset where one can clearly define the frequency-domain signal-target (ir)relevance. This motivates us to directly define the distribution of the input signal in the frequency domain. In Sec. 4, we consider $\tilde{X}$ follows a Gaussian distribution $\mathcal{N}(\tilde{\mu}, \tilde{\Sigma})$, and for analytical tractability, we consider $\tilde{\mu} = 0$ and a diagonal structure of $\tilde{\Sigma}$, i.e., $\tilde{\Sigma} = \mathrm{diag}(\tilde{\sigma}_0^2, ..., \tilde{\sigma}_{d-1}^2)$. Admittedly, it is quite unconventional to define the data distribution directly in the frequency domain, so we provide a few examples in Table 8 to illustrate the structure of the input data in both representations.

Similar to Sec. 4.2.2, we focus on a low dimensional setting with $d = 3$. The first six rows in Table 8 represent the scenario when there are zero variances in $\tilde{\Sigma}$. Notice the notion of irrelevant information in the data is different in the two representations. In the frequency domain, an irrelevant frequency indicates that the data has a value of zero at the particular frequency. In the spatial domain, having irrelevant frequency means that there are redundant dimensions in the spatial representation of the data because the value of data at some dimensions can be fully predictable by knowing the values of data at some other dimensions.

### E.2  Derivation of Equation 9

The adversarial risk under an $\ell_2$-norm bounded perturbation with a size of $\epsilon$ is

$$
\begin{aligned}
\mathcal{R}_{\mathrm{a}}(\tilde{w}) &\triangleq \mathbb{E}_{(\tilde{X},Y)}\left[\max_{||\Delta\tilde{x}||_2\leq\epsilon} \ell(\tilde{X}+\Delta\tilde{x},Y;\tilde{w})\right] \\
&= \mathbb{E}_{(\tilde{X},Y)}\left[\max_{||\Delta\tilde{x}||_2\leq\epsilon} \frac{1}{2}\left|f(\tilde{X}+\Delta\tilde{x},\tilde{w})-Y\right|^2\right] \\
&= \mathbb{E}_{\tilde{X}}\left[\max_{||\Delta\tilde{x}||_2\leq\epsilon} \frac{1}{2}\left|\left\langle \tilde{X}+\Delta\tilde{x},\,\tilde{w}\right\rangle-\left\langle \tilde{X},\,\tilde{w}^*\right\rangle\right|^2\right] \\
&= \mathbb{E}_{\tilde{X}}\left[\max_{||\Delta\tilde{x}||_2\leq\epsilon} \frac{1}{2}\left|\left\langle \tilde{X},\,\tilde{w}-\tilde{w}^*\right\rangle+\left\langle \Delta\tilde{x},\,\tilde{w}\right\rangle\right|^2\right],
\end{aligned}
$$

where we focus on the expectation over $\tilde{X}$, as $Y$ is replaced with $\left\langle \tilde{X},\,\tilde{w}^*\right\rangle$.

### E.3  Derivation of Equation 10

Given a r.v. $\tilde{X}$, we define $\Delta\tilde{x}^*$ to be the maximizer of the term inside the expectation of (9):

$$
\Delta\tilde{x}^* \triangleq \arg\max_{||\Delta\tilde{x}||_2\leq\epsilon}\frac{1}{2}\left|\left\langle \tilde{X},\,\tilde{w}-\tilde{w}^*\right\rangle+\left\langle \Delta\tilde{x},\,\tilde{w}\right\rangle\right|^2.
$$

To maximize this term, we need the two inner product terms to have the same sign. This means

$$
\Delta\tilde{x}^* = \mathrm{sign}[\left\langle \tilde{X},\,\tilde{w}-\tilde{w}^*\right\rangle]\arg\max_{||\Delta\tilde{x}||_2\leq\epsilon}\left|\left\langle \Delta\tilde{x},\,\tilde{w}\right\rangle\right|^2.
$$

For the remaining argmax term, we can first use the Cauchy-Schwarz inequality to obtain

$$
\max_{||\Delta\tilde{x}||_2\leq\epsilon}\left|\left\langle \Delta\tilde{x},\,\tilde{w}\right\rangle\right|^2 \leq \max_{||\Delta\tilde{x}||_2\leq\epsilon}\|\Delta\tilde{x}\|_2^2\,\|\tilde{w}\|_2^2 = \epsilon^2\,\|\tilde{w}\|_2^2,
$$

which leads to

$$
\arg\max_{||\Delta\tilde{x}||_2\leq\epsilon}\left|\left\langle \Delta\tilde{x},\,\tilde{w}\right\rangle\right|^2 = \epsilon\frac{\tilde{w}}{\|\tilde{w}\|_2}.
$$

Finally, we have

$$
\Delta\tilde{x}^* = \epsilon\,\mathrm{sign}[\left\langle \tilde{X},\,\tilde{w}-\tilde{w}^*\right\rangle]\frac{\tilde{w}}{||\tilde{w}||_2}.
$$

### E.4  Derivation of Equation 11

The adversarial risk is

$$
\begin{aligned}
\mathcal{R}_{\mathrm{a}}(\tilde{w}) &= \frac{1}{2}\mathbb{E}_{\tilde{X}}\left[\left|\left\langle \tilde{X},\,\tilde{w}-\tilde{w}^*\right\rangle+\epsilon\,\mathrm{sign}[\left\langle \tilde{X},\,\tilde{w}-\tilde{w}^*\right\rangle]||\tilde{w}||_2\right|^2\right] \\
&= \frac{1}{2}\mathbb{E}_{\tilde{X}}\left[\left\langle \tilde{X},\,\tilde{w}-\tilde{w}^*\right\rangle^2+2\epsilon\left|\left\langle \tilde{X},\,\tilde{w}-\tilde{w}^*\right\rangle\right|||\tilde{w}||_2+\epsilon^2||\tilde{w}||_2^2\right] \\
&= \frac{1}{2}\sum_{i\in\mathbb{I}_{\mathrm{rel}}}\tilde{\sigma}_i^2(\tilde{w}_i-\tilde{w}_i^*)^2+\epsilon\mathbb{E}_{\tilde{X}}\left[\left|\left\langle \tilde{X},\,\tilde{w}-\tilde{w}^*\right\rangle\right|\right]||\tilde{w}||_2+\frac{\epsilon^2}{2}||\tilde{w}||_2^2.
\end{aligned}
$$

To compute the expectation, we first denote $Z = \sum_{i\in\mathbb{I}_{\mathrm{rel}}}\tilde{X}_i(\tilde{w}_i-\tilde{w}_i^*)$. Because $\tilde{\sigma}_i^2 = 0$ for all $i\in\mathbb{I}_{\mathrm{irrel}}$, this allows us to ignore those irrelevant frequencies in the summation in $Z$. This leads us to

$$
\mathbb{E}_{\tilde{X}}\left[\left|\left\langle \tilde{X},\,\tilde{w}-\tilde{w}^*\right\rangle\right|\right] = \mathbb{E}_Z\left[|Z|\right].
$$

Since $Z$ is a linear combination of zero-mean Gaussian r.v.'s, this makes it also a zero-mean Gaussian r.v, i.e., $\mathbb{E}[Z] = 0$. The variance of $Z$ is

$$
\begin{aligned}
\sigma_Z^2 &= \mathbb{E}[Z^2] - \mathbb{E}[Z]^2 \\
&= \mathbb{E}\left[ \sum_{i \in \mathbb{I}_{\text{rel}}} \sum_{j \in \mathbb{I}_{\text{rel}}, i \neq j} \left[ \tilde{X}_i \tilde{X}_j (\tilde{w}_i - \tilde{w}_i^*)(\tilde{w}_j - \tilde{w}_j^*) \right] + \sum_{i \in \mathbb{I}_{\text{rel}}} \left[ \tilde{X}_i^2 (\tilde{w}_i - \tilde{w}_i^*)^2 \right] \right] \\
&= \mathbb{E}\left[ \sum_{i \in \mathbb{I}_{\text{rel}}} \left[ \tilde{X}_i^2 (\tilde{w}_i - \tilde{w}_i^*)^2 \right] \right] \\
&= \sum_{i \in \mathbb{I}_{\text{rel}}} \tilde{\sigma}_i^2 (\tilde{w}_i - \tilde{w}_i^*)^2,
\end{aligned}
$$

where the expectation on the cross-multiplication term is zero because $\tilde{X}_i$ and $\tilde{X}_j$ are independent r.v.'s. This means $Z \sim \mathcal{N}(0, \sigma_Z^2)$ with $\sigma_Z^2 = \sum_{i \in \mathbb{I}_{\text{rel}}} \tilde{\sigma}_i^2 (\tilde{w}_i - \tilde{w}_i^*)^2$. Therefore, $\mathbb{E}_Z\left[|Z|\right]$ is the expectation of a folded normal distribution:

$$
\mathbb{E}_Z\left[|Z|\right] = \sigma_Z \sqrt{\frac{2}{\pi}} = \sqrt{\frac{2}{\pi} \sum_{i \in \mathbb{I}_{\text{rel}}} \tilde{\sigma}_i^2 (\tilde{w}_i - \tilde{w}_i^*)^2}.
$$

### E.5 Derivations of the GD Dynamics: Equation 14 and Equation 15

The gradient computed using the population risk is $\nabla_w \mathcal{R}_s(w(t)) = \mathbb{E}\left[XX^\top\right] e(t) = \Sigma e(t)$, and the learning dynamics of GD in the spatial domain can be captured using:

$$
\begin{aligned}
e(t+1) &= w(t+1) - w^* \\
&= w(t) - \eta \nabla_w \mathcal{R}_s(w(t)) - w^* \\
&= w(t) - w^* - \eta \Sigma e(t) \\
&= e(t) - \eta \Sigma e(t) \\
&= (I - \eta \Sigma) e(t) \\
&= (I - \eta \Sigma)^{t+1} e(0).
\end{aligned}
$$

This shows that the learned weight converges to the optimal weight $w^*$ at a rate depending on $\Sigma$. To see the GD dynamics in the frequency domain, we can simply perform DCT on both sides of (14):

$$
\begin{aligned}
\tilde{e}(t+1) &= C(I - \eta \Sigma)^{t+1} e(0) \\
&= C(I - \eta \Sigma)^{t+1} C^\top \tilde{e}(0) \\
&= C(I - \eta \Sigma)^{t} C^\top C(I - \eta \Sigma) C^\top \tilde{e}(0) \\
&= C(I - \eta \Sigma)^{t-1} C^\top C(I - \eta \Sigma) C^\top C(I - \eta \Sigma) C^\top \tilde{e}(0) \\
&= \left[ C(I - \eta \Sigma) C^\top \right]^{t+1} \tilde{e}(0) \\
&= (I - \eta C \Sigma C^\top)^{t+1} \tilde{e}(0) \\
&= (I - \eta \tilde{\Sigma})^{t+1} \tilde{e}(0),
\end{aligned}
$$

where $\tilde{\Sigma}$ is the covariance of $\tilde{x}$.

### E.6 Derivation of the signGD Dynamics for any $\tilde{\Sigma}$: Equation 19 and Equation 20

The signGD learning dynamics in the spatial domain is

$$
\begin{aligned}
e(t+1) &= w(t+1) - w^* \\
&= w(t) - \eta \operatorname{sign}[\nabla_w \mathcal{R}_s(w)] - w^* \\
&= e(t) - \eta \operatorname{sign}[\Sigma e(t)].
\end{aligned}
$$

The signGD learning dynamics in the frequency domain are obtained by taking the DCT transformation on both sides of (19):

$$
\begin{aligned}
\tilde{e}(t+1) &= \tilde{e}(t) - \eta C \operatorname{sign}[\Sigma e(t)] \\
&= \tilde{e}(t) - \eta C \operatorname{sign}[C^\top \tilde{\Sigma} C C^\top \tilde{e}(t)] \\
&= \tilde{e}(t) - \eta C \operatorname{sign}[C^\top \tilde{\Sigma} \tilde{e}(t)].
\end{aligned}
$$

## E.7 Derivation of the signGD Dynamics for $\tilde{\Sigma} = \operatorname{diag}\left\{\tilde{\sigma}_0^2, \tilde{\sigma}_1^2, 0\right\}$: Equation 21

To understand $\tilde{e}(t+1)$ with our specific choice of $\Sigma = C^\top \tilde{\Sigma} C$ and $\tilde{\Sigma} = \operatorname{diag}\left\{\tilde{\sigma}_0^2, \tilde{\sigma}_1^2, 0\right\}$, first notice that the DCT transformation matrix $C = C^{(3)}$ follows the definition in (4):

$$
C = \begin{bmatrix}
\sqrt{\frac{1}{3}} & \sqrt{\frac{1}{3}} & \sqrt{\frac{1}{3}} \\
\sqrt{\frac{2}{3}}\cos\frac{\pi}{6} & \sqrt{\frac{2}{3}}\cos\frac{\pi}{2} & \sqrt{\frac{2}{3}}\cos\frac{5\pi}{6} \\
\sqrt{\frac{2}{3}}\cos\frac{\pi}{3} & \sqrt{\frac{2}{3}}\cos\pi & \sqrt{\frac{2}{3}}\cos\frac{5\pi}{3}
\end{bmatrix}
= \begin{bmatrix}
\sqrt{\frac{1}{3}} & \sqrt{\frac{1}{3}} & \sqrt{\frac{1}{3}} \\
\sqrt{\frac{1}{2}} & 0 & -\sqrt{\frac{1}{2}} \\
\sqrt{\frac{1}{6}} & -\sqrt{\frac{2}{3}} & \sqrt{\frac{1}{6}}
\end{bmatrix}.
$$

Denote $\sqrt{\frac{1}{3}}\tilde{\sigma}_0^2 \tilde{e}_0(t)$ and $\sqrt{\frac{1}{2}}\tilde{\sigma}_1^2 \tilde{e}_1(t)$ by using $A(t)$ and $B(t)$, respectively. Putting it all together, we have:

$$
\begin{aligned}
\tilde{e}(t+1) &= \tilde{e}(t) - \eta C \operatorname{sign}[C^\top \tilde{\Sigma} C e(t)] \\
&= \tilde{e}(t) - \eta C \operatorname{sign}[C^\top \tilde{\Sigma} \tilde{e}(t)] \\
&= \tilde{e}(t) - \eta \begin{bmatrix}
\sqrt{\frac{1}{3}} & \sqrt{\frac{1}{3}} & \sqrt{\frac{1}{3}} \\
\sqrt{\frac{1}{2}} & 0 & -\sqrt{\frac{1}{2}} \\
\sqrt{\frac{1}{6}} & -\sqrt{\frac{2}{3}} & \sqrt{\frac{1}{6}}
\end{bmatrix} \operatorname{sign}\left[\begin{bmatrix}
\sqrt{\frac{1}{3}} & \sqrt{\frac{1}{3}} & \sqrt{\frac{1}{3}} \\
\sqrt{\frac{1}{2}} & 0 & -\sqrt{\frac{1}{2}} \\
\sqrt{\frac{1}{6}} & -\sqrt{\frac{2}{3}} & \sqrt{\frac{1}{6}}
\end{bmatrix}^\top \begin{bmatrix} \tilde{\sigma}_0^2 \tilde{e}_0(t) \\ \tilde{\sigma}_1^2 \tilde{e}_1(t) \\ 0 \end{bmatrix}\right] \\
&= \tilde{e}(t) - \eta \begin{bmatrix}
\sqrt{\frac{1}{3}} & \sqrt{\frac{1}{3}} & \sqrt{\frac{1}{3}} \\
\sqrt{\frac{1}{2}} & 0 & -\sqrt{\frac{1}{2}} \\
\sqrt{\frac{1}{6}} & -\sqrt{\frac{2}{3}} & \sqrt{\frac{1}{6}}
\end{bmatrix} \begin{bmatrix}
\operatorname{sign}\left[\sqrt{\frac{1}{3}}\tilde{\sigma}_0^2 \tilde{e}_0(t) + \sqrt{\frac{1}{2}}\tilde{\sigma}_1^2 \tilde{e}_1(t)\right] \\
\operatorname{sign}\left[\sqrt{\frac{1}{3}}\tilde{\sigma}_0^2 \tilde{e}_0(t)\right] \\
\operatorname{sign}\left[\sqrt{\frac{1}{3}}\tilde{\sigma}_0^2 \tilde{e}_0(t) - \sqrt{\frac{1}{2}}\tilde{\sigma}_1^2 \tilde{e}_1(t)\right]
\end{bmatrix} \\
&= \tilde{e}(t) - \eta \begin{bmatrix}
\sqrt{\frac{1}{3}} & \sqrt{\frac{1}{3}} & \sqrt{\frac{1}{3}} \\
\sqrt{\frac{1}{2}} & 0 & -\sqrt{\frac{1}{2}} \\
\sqrt{\frac{1}{6}} & -\sqrt{\frac{2}{3}} & \sqrt{\frac{1}{6}}
\end{bmatrix} \begin{bmatrix}
\operatorname{sign}[A(t) + B(t)] \\
\operatorname{sign}[A(t)] \\
\operatorname{sign}[A(t) - B(t)]
\end{bmatrix} \\
&= \tilde{e}(t) - \eta \begin{bmatrix}
\frac{\sqrt{3}}{3}\left(\operatorname{sign}[A(t) + B(t)] + \operatorname{sign}[A(t)] + \operatorname{sign}[A(t) - B(t)]\right) \\
\frac{\sqrt{2}}{2}\left(\operatorname{sign}[A(t) + B(t)] - \operatorname{sign}[A(t) - B(t)]\right) \\
\frac{\sqrt{6}}{6}\operatorname{sign}[A(t) + B(t)] - \frac{\sqrt{6}}{3}\operatorname{sign}[A(t)] + \frac{\sqrt{6}}{6}\operatorname{sign}[A(t) - B(t)]
\end{bmatrix}.
\end{aligned}
$$

## E.8 Understanding the Dynamics of signGD with $\tilde{\Sigma} = \operatorname{diag}\left\{\tilde{\sigma}_0^2, \tilde{\sigma}_1^2, 0\right\}$

Previous work has shown that for strictly convex problems with a unique minimum, the signGD solution converges to the minimum under a sequence of decaying learning rate: $\lim_{t\to\infty} \eta(t) = 0$ (Moulay et al., 2019). In this section, we follow Sec. 4.2.1 where the GD dynamics is studied under a constant learning rate and investigate the behavior of signGD under a fixed $\eta$. Compared to the asymptotic GD solution that converges exactly to the standard risk minimizer, we demonstrate that the asymptotic signGD solution converges to an $O(\eta)$ neighborhood of the standard risk minimizer.

For the rest of this section, we first perform a partition-based analysis to study the learning dynamics of $\tilde{e}_0$ and $\tilde{e}_1$ in Appx. E.8.1. Proposition E.4 summarizes how the value of weight changes in different partitions,

Table 9: **Learning dynamics of signGD.** The dynamics of the error term in the frequency domain can be written in a tabular format and the exact update depends on the initialized weight $\tilde{w}(0)$ and the true model $\tilde{w}^*$. We use $A(t)$ and $B(t)$ to denote $\frac{\sqrt{3}}{3}\tilde{\sigma}_0^2\tilde{e}_0(t)$ and $\frac{\sqrt{2}}{2}\tilde{\sigma}_1^2\tilde{e}_1(t)$, respectively. Invalid sign combinations are denoted by using n/a.

| No. | sign[$A(t)$] | sign[$A(t) + B(t)$] | sign[$A(t) - B(t)$] | $\tilde{e}(t+1)$ | $|A(t)|$ vs. $|B(t)|$ |
|---|---|---|---|---|---|
| 1 | 1 | 1 | 1 | $\tilde{e}(t) - \eta \left[ \sqrt{3}, 0, 0 \right]^\top$ | $|A(t)| > |B(t)|$ |
| 2 | 1 | 1 | $-1$ | $\tilde{e}(t) - \eta \left[ \frac{\sqrt{3}}{3}, \sqrt{2}, -\frac{\sqrt{6}}{3} \right]^\top$ | $|A(t)| < |B(t)|$ |
| 3 | 1 | 1 | 0 | $\tilde{e}(t) - \eta \left[ \frac{2\sqrt{3}}{3}, \frac{\sqrt{2}}{2}, -\frac{\sqrt{6}}{6} \right]^\top$ | $|A(t)| = |B(t)|$ |
| 4 | 1 | $-1$ | 1 | $\tilde{e}(t) - \eta \left[ \frac{\sqrt{3}}{3}, -\sqrt{2}, -\frac{\sqrt{6}}{3} \right]^\top$ | $|A(t)| < |B(t)|$ |
| n/a | 1 | $-1$ | $-1$ | n/a | n/a |
| n/a | 1 | $-1$ | 0 | n/a | n/a |
| 5 | 1 | 0 | 1 | $\tilde{e}(t) - \eta \left[ \frac{2\sqrt{3}}{3}, -\frac{\sqrt{2}}{2}, -\frac{\sqrt{6}}{6} \right]^\top$ | $|A(t)| = |B(t)|$ |
| n/a | 1 | 0 | $-1$ | n/a | n/a |
| n/a | 1 | 0 | 0 | n/a | n/a |
| n/a | $-1$ | 1 | 1 | n/a | n/a |
| 6 | $-1$ | 1 | $-1$ | $\tilde{e}(t) - \eta \left[ -\frac{\sqrt{3}}{3}, \sqrt{2}, \frac{\sqrt{6}}{3} \right]^\top$ | $|A(t)| < |B(t)|$ |
| n/a | $-1$ | 1 | 0 | n/a | n/a |
| 7 | $-1$ | $-1$ | 1 | $\tilde{e}(t) - \eta \left[ -\frac{\sqrt{3}}{3}, -\sqrt{2}, \frac{\sqrt{6}}{3} \right]^\top$ | $|A(t)| < |B(t)|$ |
| 8 | $-1$ | $-1$ | $-1$ | $\tilde{e}(t) - \eta \left[ -\sqrt{3}, 0, 0 \right]^\top$ | $|A(t)| > |B(t)|$ |
| 9 | $-1$ | $-1$ | 0 | $\tilde{e}(t) - \eta \left[ -\frac{2\sqrt{3}}{3}, -\frac{\sqrt{2}}{2}, \frac{\sqrt{6}}{6} \right]^\top$ | $|A(t)| = |B(t)|$ |
| n/a | $-1$ | 0 | 1 | n/a | n/a |
| 10 | $-1$ | 0 | $-1$ | $\tilde{e}(t) - \eta \left[ -\frac{2\sqrt{3}}{3}, \frac{\sqrt{2}}{2}, \frac{\sqrt{6}}{6} \right]^\top$ | $|A(t)| = |B(t)|$ |
| n/a | $-1$ | 0 | 0 | n/a | n/a |
| n/a | 0 | 1 | 1 | n/a | n/a |
| 11 | 0 | 1 | $-1$ | $\tilde{e}(t) - \eta \left[ 0, \sqrt{2}, 0 \right]^\top$ | $|A(t)| < |B(t)|$ |
| n/a | 0 | 1 | 0 | n/a | n/a |
| 12 | 0 | $-1$ | 1 | $\tilde{e}(t) - \eta \left[ 0, -\sqrt{2}, 0 \right]^\top$ | $|A(t)| < |B(t)|$ |
| n/a | 0 | $-1$ | $-1$ | n/a | n/a |
| n/a | 0 | $-1$ | 0 | n/a | n/a |
| n/a | 0 | 0 | 1 | n/a | n/a |
| n/a | 0 | 0 | $-1$ | n/a | n/a |
| 13 | 0 | 0 | 0 | Optimal | $|A(t)| = |B(t)| = 0$ |

and the weight adaptation of $\tilde{e}_0$ and $\tilde{e}_1$ is summarized in Corollary E.6. Based on the corollary, we analyze the dynamics of $\tilde{e}_2$ in Appx. E.8.2. Lastly, we focus on the differences between the adversarial risk of solutions found by GD and signGD in Appx. E.8.3.

### E.8.1 Dynamics of $\tilde{e}_0$ and $\tilde{e}_1$ under signGD

With our choice of $\tilde{\Sigma}$, (21) shows that weight adaptation depends on the sign of three terms: $A(t)$, $A(t)+B(t)$ and $A(t) - B(t)$. This allows us to study the learning dynamics of signGD by analyzing the three terms in Table 9. There are 27 sign combinations in total; however, not all of them are valid. For example, consider the combination with sign[$A(t)$] = 1, sign[$A(t) + B(t)$] = $-1$. Those two conditions imply that $B(t) < 0$ and $|A(t)| < |B(t)|$, and this means that sign[$A(t) - B(t)$] must be positive. This makes the entry with sign[$A(t) - B(t)$] = $-1$ invalid, as shown in the fifth row in Table 9. We denote those entries with invalid sign combinations as n/a.

Notice in (21) that the weight adaptation under signGD depends on the dynamics of $A(t)$ and $B(t)$, and they are functions of $\tilde{e}_0(t)$ and $\tilde{e}_1(t)$ respectively, so let us first focus on understanding the weight adaptation at the first two frequency bases.

There are 13 possible updates in Table 9. Notice that the non-zero updates are always in the direction to reduce $|\tilde{e}_0(t)|$ and $|\tilde{e}_1(t)|$, and the step size depends on the magnitude of $|A(t)|$ and $|B(t)|$. To simplify the analysis, let us focus on the updates on $A$ and $B$ instead. For example, in updates 1 and 8, decreasing $|\tilde{e}_0|$ by $\sqrt{3}\eta$ is equivalent to decreasing $|A|$ by $\tilde{\sigma}_0^2\eta$.

Now take note of the limited number of update magnitudes for $|A|$, specifically $\tilde{\sigma}_0^2\eta$, $\frac{2\tilde{\sigma}_0^2\eta}{3}$, and $\frac{\tilde{\sigma}_0^2\eta}{3}$, which correspond to updating $|\tilde{e}_0|$ by $\sqrt{3}\eta$, $\frac{2\sqrt{3}}{3}\eta$, and $\frac{\sqrt{3}}{3}\eta$, respectively. Similarly, $|B|$ has only two update magnitudes, namely $\tilde{\sigma}_1^2\eta$ and $\frac{\tilde{\sigma}_1^2\eta}{2}$, which correspond to updating $|\tilde{e}_1|$ with $\sqrt{2}\eta$ and $\frac{\sqrt{2}}{2}\eta$, respectively. This observation leads to the following proposition.

**Proposition E.1.** *Suppose the initial weight $w(0) \sim \mu$ are sampled from probability density $\mu$, then neither $A$ nor $B$ ($\tilde{e}_0$ nor $\tilde{e}_1$) can be reduced to exactly 0 almost surely.*

*Proof.* Due to the limited number of update magnitudes, reducing $|A|$ and $|B|$ to 0 requires their initial value to be exactly some integer multiplication of those updates. However, with the initial weight sampled from probability density $\mu$, the probability of the initial values of $A$ and $B$ being the exact integer multiple of the possible update is 0. $\square$

Next, we introduce the following lemma to understand the dynamics of $A(t)$.

**Lemma E.2.** *Consider the update rule $x(t + 1) = x(t) - \text{sign}[x(t)]\Delta(t)$, where $x \in \mathbb{R}$, $\Delta(t) \in \{\Delta_1, \Delta_2, \ldots, \Delta_{\max}\}$ and $0 < \Delta_1 < \Delta_2 < \cdots < \Delta_{\max}$. Then there exists $t$ such that $|x(t)| \leq \Delta_{\max}$. Moreover, whenever $|x(t)| \leq \Delta_{\max}$, the rest of the sequence stays $\Delta_{\max}$-bounded, i.e., $|x(t')| \leq \Delta_{\max}$ for all $t' \geq t$.*

*Proof.* In the following, we provide a proof for the case when $x(0) > 0$. A proof with $x(0) < 0$ can be done in a similar way. The proof can be divided into two parts.

1. Let us denote the sequence of $\{x(0), x(1), \ldots, x(t)\}$ by $\{x(t)\}$. First, we prove that there exists a $t$ such that $|x(t)| \leq \Delta_{\max}$.

If $x(t) > \Delta_{\max}$, then $x(t + 1) = x(t) - \Delta(t) \geq x(t) - \Delta_{\max} > 0$.

Consider $\{x(t)\}$ with $x(t') > \Delta_{\max}$ for all $t' \in \{0, 1, \ldots, t\}$, we have $x(t' + 1) = x(t') - \Delta(t') < x(t')$. This means that for any $x(t) > \Delta_{\max}$, the sequence $\{x(t)\}$ is decreasing.

We prove, by contradiction, that there exists a $t$ such that $|x(t)| < \Delta_{\max}$. Suppose that such a $t$ does not exist, then one of the two cases must happen.

1. $x(t) > \Delta_{\max}$ for all $t$.

2. $\exists k$ such that $x(0) > x(1) > \cdots > x(k) > \Delta_{\max}$, but $x(k + 1) < -\Delta_{\max}$.

For case 1, since $x(t) > \Delta_{\max}$, we know that $\{x(t)\}$ is decreasing and bounded from below, so we have $x(t) \to x^* \geq \Delta_{\max}$ as $t \to \infty$. This means that $\lim_{t\to\infty} x(t) = \lim_{t\to\infty} x(t + 1) = x^*$.

Using the update rule, we have

$$\lim_{t\to\infty} x(t + 1) = \lim_{t\to\infty} x(t) - \Delta(t)\,\text{sign}(x(t))$$
$$= \lim_{t\to\infty} x(t) - \Delta(t)$$
$$= x^* - \Delta(t),$$

or $x^* = x^* - \Delta(t)$, which is impossible because $\Delta(t) > 0$.

For case 2, by the assumption of the case, we have $x(k + 1) = x(k) - \Delta(k) < -\Delta_{\max}$, which is not possible because $x(k) > \Delta_{\max}$.

The same approach can be applied to prove the case when $x$ is initialized with a negative value, i.e., $x(0) < 0$.

The first part of the proof shows that there exists a $t$ such that $|x(t)| \leq \Delta_{\max}$.

2. Next, we show that for any $t$ such that $|x(t)| \leq \Delta_{\max}$, we have $|x(t+1)| \leq \Delta_{\max}$.

When $0 \leq x(t) \leq \Delta_{\max}$, we have

$$x(t+1) = x(t) - \Delta(t) \geq -\Delta_{\max} \qquad \text{and} \qquad x(t+1) = x(t) - \Delta(t) \leq x(t) \leq \Delta_{\max}.$$

When $-\Delta_{\max} \leq x(t) \leq 0$, we have

$$x(t+1) = x(t) + \Delta(t) \leq \Delta_{\max} \qquad \text{and} \qquad x(t+1) = x(t) + \Delta(t) \geq x(t) \geq -\Delta_{\max}.$$

This means that $-\Delta_{\max} \leq x(t+1) \leq \Delta_{\max}$, and this results holds for any $t$ such that $|x(t)| \leq \Delta_{\max}$.

To combine the two parts of the proof, consider the first of such $t$, i.e., $t = \min \{ t : |x(t)| \leq \Delta_{\max} \}$. We can prove, by mathematical induction, that $|x(t')| \leq \Delta_{\max}$ for all $t' \geq t$. $\square$

The following proposition describes the behavior of $A(t)$ under signGD.

**Proposition E.3.** *There exists $t$ such that $|A(t')| \leq \tilde{\sigma}_0^2 \eta$ for all $t' > t$.*

*Proof.* Table 9 shows that there is always a non-zero update in the direction to reduce $|A(t)|$, so we can define the dynamics of $A(t)$ as
$$A(t+1) = A(t) - \text{sign}[A(t)]\Delta(t),$$
where $\Delta(t) \in \left\{ \frac{\tilde{\sigma}_0^2 \eta}{3}, \frac{2\tilde{\sigma}_0^2 \eta}{3}, \tilde{\sigma}_0^2 \eta \right\}$. Lemma E.2 with $\Delta_{\max} = \tilde{\sigma}_0^2 \eta$ proves the proposition. $\square$

Proposition E.3 implies that once $|A|$ drops below $\tilde{\sigma}_0^2 \eta$, it remains below $\tilde{\sigma}_0^2 \eta$ for all future iterations. Combining Proposition E.3 with the update directions of $A$ in Table 9, we know that $A$ will begin oscillating around zero. However, there are some limitations of Proposition E.3. First, we do not know when exactly the oscillation starts: whether it starts immediately following the first iteration when $|A| \leq \tilde{\sigma}_0^2 \eta$ or from some iterations after it. Second, the characteristics of this oscillation (periodic or non-periodic) are unknown. Answers to these questions can improve our understanding of the behavior of $A$, and later become particularly useful in developing the asymptotic signGD solution of $\tilde{e}_2$, which is important because it leads to the adversarial risk of the signGD solution.

Because the update for $B(t)$ can be zero when $|A(t)| > |B(t)|$, Lemma E.2 is not suitable to understand the dynamics of $B$, as the lemma requires that all step sizes be greater than zero. Nevertheless, Proposition E.3 allows us to narrow down the range of $A$ and we can partition the set of all possible values of $A$ and $B$. By analyzing the dynamics of $A$ and $B$ in those partitions, we can develop the standard and adversarial population risk of the asymptotic signGD solution under a constant learning rate $\eta$.

Let us first divide the set of values of $(A, B)$ into partitions based on the value of $|A|$, and then divide those partitions into smaller subpartitions based on the relative magnitude of $|A|$ and $|B|$. Such a partitioning process is illustrated in Figure 6.

- $R_1 = \left\{ (A, B) : \frac{2\tilde{\sigma}_0^2 \eta}{3} < |A| < \tilde{\sigma}_0^2 \eta \text{ and } B \in (-\infty, \infty) \right\}$,

  - $R_{11} = \{ (A, B) : (A, B) \in R_1 \text{ and } |A| < |B| \}$,
  - $R_{12} = \{ (A, B) : (A, B) \in R_1 \text{ and } |A| > |B| \}$,

- $R_2 = \left\{ (A, B) : \frac{\tilde{\sigma}_0^2 \eta}{3} < |A| < \frac{2\tilde{\sigma}_0^2 \eta}{3} \text{ and } B \in (-\infty, \infty) \right\}$,

  - $R_{21} = \{ (A, B) : (A, B) \in R_2 \text{ and } |A| < |B| \}$,
  - $R_{22} = \left\{ (A, B) : (A, B) \in R_2 \text{ and } |A| > |B| \text{ and } \left| A + \tilde{\sigma}_0^2 \eta \right| > |B| \text{ and } \left| A - \tilde{\sigma}_0^2 \eta \right| > |B| \right\}$,
  - $R_{23} = \left\{ (A, B) : (A, B) \in R_2 \text{ and } |A| > |B| \text{ and } \left( \left| A + \tilde{\sigma}_0^2 \eta \right| < |B| \text{ or } \left| A - \tilde{\sigma}_0^2 \eta \right| < |B| \right) \right\}$,

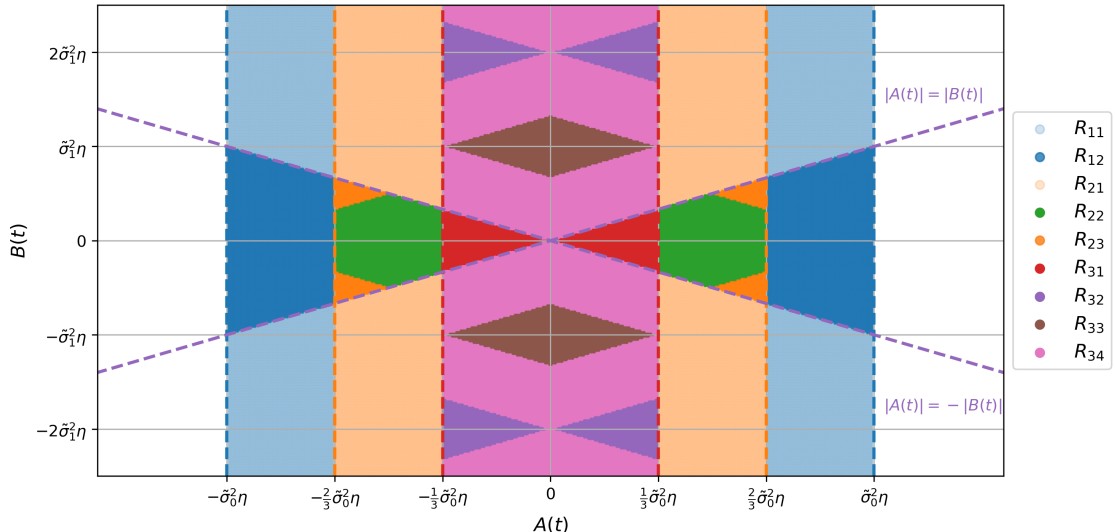

Figure 6: **Analyzing the dynamics of $A$ and $B$ by partitioning the set of values of $(A, B)$ in** $[-\tilde{\sigma}_0^2\eta, \tilde{\sigma}_0^2\eta] \times \mathbb{R}$. Such a set is first divided into partitions $R_1$, $R_2$ and $R_3$ based on the value of $|A|$. Then, we consider $R_s = \{R_{22}, R_{31}, R_{32}, R_{33}, R_{34}\}$ as the stationary subpartitions, because once $(A(t), B(t)) \in R_s$, the sequence remains in the stationary subpartitions. Also, we consider $R_t = \{R_{11}, R_{12}, R_{21}, R_{23}\}$ as the transient subpartitions, because any $(A(t), B(t)) \in R_t$ will soon enter one of the stationary subpartitions, that is, there exists $t' \geq t$ such that $(A(t'), B(t')) \in R_s$. We consider $\tilde{\sigma}_0^2 = \tilde{\sigma}_1^2 = 1$ and $\eta = 1$ in this figure.

- $R_3 = \left\{ (A, B) \, : \, |A| < \frac{\tilde{\sigma}_0^2\eta}{3} \text{ and } B \in (-\infty, \infty) \right\}$,

  - $R_{31} = \left\{ (A, B) \, : \, (A, B) \in R_3 \text{ and } |A| > |B| \right\}$,
  - $R_{32} = \left\{ (A, B) \, : \, \bigcup_{k \in \mathbb{Z}_{\text{even}} - \{0\}} \left\{ (A, B) \in R_3 \text{ and } |A| > \left| B + k\tilde{\sigma}_1^2\eta \right| \right\} \right\}$,
  - $R_{33} = \left\{ (A, B) \, : \, \bigcup_{k \in \mathbb{Z}_{\text{odd}}} \left\{ (A, B) \in R_3 \text{ and } |A| + \left| B + k\tilde{\sigma}_1^2\eta \right| < \frac{\tilde{\sigma}_0^2\eta}{3} \right\} \right\}$,
  - $R_{34} = R_3 - (R_{31} \cup R_{32} \cup R_{33})$,

where $\mathbb{Z}_{\text{odd}}$ and $\mathbb{Z}_{\text{even}}$ are the set of odd and even integers, respectively.

There are nine non-overlapping subpartitions. We call $R_{22}$, $R_{31}$, $R_{32}$, $R_{33}$ and $R_{34}$ the *stationary* subpartitions and denote $R_s = \{R_{22}, R_{31}, R_{32}, R_{33}, R_{34}\}$. They are called stationary subpartitions because once $(A(t), B(t)) \in R_s$, the sequence remains in the stationary subpartition. On the other hand, we call $R_{11}$, $R_{12}$, $R_{21}$, $R_{23}$ the *transient* subpartitions and denote $R_t = \{R_{11}, R_{12}, R_{21}, R_{23}\}$. They are called the transient subpartitions because any $(A(t), B(t)) \in R_t$ will soon enter one of the stationary subpartitions, that is, there exists $t' \geq t$ such that $(A(t'), B(t')) \in R_s$. The dynamics of $A$ and $B$ can be summarized in the following proposition.

**Proposition E.4.**
***Transient subpartitions:*** *For each of $R \in R_t$, consider $t$ such that $(A(t), B(t)) \in R$, then there exists $t' > t$ such that $(A(t'), B(t')) \in R_s$. The transition of $(A(t), B(t))$ from $R_t$ to $R_s$ happens at most 3 iterations after $t$; specifically, it corresponds to the scenario of $(A(t), B(t)) \in R_{11}$, $(A(t+1), B(t+1)) \in R_{23}$, $(A(t+2), B(t+2)) \in R_{21}$, and finally $(A(t+3), B(t+3)) \in R_3$.*

***Stationary subpartitions:*** *For each of $R \in R_s$, consider $t$ such that $(A(t), B(t)) \in R$. For any $t' \geq t$, $|A(t')| \leq \frac{2\tilde{\sigma}_0^2\eta}{3}$ and $A(t')$ shows 2-periodic behavior switching between positive and negative signs, that is, for any $i \in \mathbb{Z}_{\geq 0}$, we have $A(t+2i) = A(t)$ and $\text{sign}(A(t+2i)) = \text{sign}(A(t)) = -\text{sign}(A(t+2i+1))$. To be more specific about each stationary subpartition, we have*

1. *For each of $R \in \{R_{22}, R_{31}\}$, consider $t$ such that $(A(t), B(t)) \in R$. For any $t' \geq t$, $|B(t')| \leq \tilde{\sigma}_1^2 \eta$ and $B(t')$ remains constant, that is, $B(t') = B(t)$.*

2. *For each of $R \in \{R_{32}, R_{33}\}$, consider $t$ such that $(A(t), B(t)) \in R$.*

   (a) *There exists $\bar{t} > t$ such that $(A(\bar{t}), B(\bar{t})) \in R_{31}$. Denote the smallest $\bar{t}$ as $\bar{t}^*$.*

   (b) *For any $\bar{t}^* > t' \geq t$, we have $|B(t'+1)| = |B(t')| - \text{sign}[B(t')]$.*

   (c) *For any $t' \geq \bar{t}^*$, $|B(t')| \leq \tilde{\sigma}_1^2 \eta$ and $B(t')$ remains constant, that is, $B(t') = B(t)$.*

3. *Consider $t$ such that $(A(t), B(t)) \in R_{34}$.*

   (a) *For any $t' \geq t$, $(A(t'), B(t'))$ remains in $R_{34}$.*

   (b) *There exists $\bar{t} > t$ such that for any $\bar{t} > t' \geq t$, the sign of $B(t')$ remains constant.*

   (c) *For any $t' \geq \bar{t}$, $|B(t')| \leq \tilde{\sigma}_1^2 \eta$ and $B(t')$ shows 2-periodic behavior switching between positive and negative signs, that is, for any $i \in \mathbb{Z}_{\geq 0}$, we have $B(\bar{t} + 2i) = B(\bar{t})$ and $\text{sign}(B(\bar{t} + 2i)) = \text{sign}(B(\bar{t})) = -\text{sign}(B(\bar{t} + 2i + 1))$.*

*Proof.* From Proposition E.3, we know that from an arbitrary $(A(0), B(0))$, $|A|$ will drop below $\tilde{\sigma}_0^2 \eta$ under the signGD update, which means that $(A, B)$ must enter one of the subpartitions. This allows us to continue analyzing the behavior of $(A(t), B(t))$ by assuming it enters one of the subpartitions at iteration $t$.

**Analysis of $R_{11}$:** For any $(A(t), B(t))$ in $R_{11}$, we know that $A(t+1) = A(t) - \text{sign}[A(t)]\frac{\tilde{\sigma}_0^2 \eta}{3}$. This means that $\frac{\tilde{\sigma}_0^2 \eta}{3} < |A(t+1)| < \frac{2\tilde{\sigma}_0^2 \eta}{3}$, so $(A(t+1), B(t+1))$ is in $R_2$ and we can study its dynamics using $R_2$.

**Analysis of $R_{12}$:** For any $(A(t), B(t))$ in $R_{12}$, we know that $A(t+1) = A(t) - \text{sign}[A(t)]\tilde{\sigma}_0^2 \eta$. This means that $|A(t+1)| < \frac{\tilde{\sigma}_0^2 \eta}{3}$. Therefore, $(A(t+1), B(t+1))$ is in $R_3$ and we can analyze its dynamics using $R_3$.

**Analysis of $R_{21}$:** For any $(A(t), B(t))$ in $R_{21}$, we know that $A(t+1) = A(t) - \text{sign}[A(t)]\frac{\tilde{\sigma}_0^2 \eta}{3}$. This means that $|A(t+1)| < \frac{\tilde{\sigma}_0^2 \eta}{3}$. Therefore, $(A(t+1), B(t+1))$ is in $R_3$ and we can analyze its dynamics using $R_3$.

**Dynamics of $(A, B)$ in $R_{22}$ and $R_{23}$:** For any $(A(t), B(t))$ in $R_{22}$ and $R_{23}$, we have $A(t+1) = A(t) - \text{sign}[A(t)]\tilde{\sigma}_0^2 \eta$ and $B(t+1) = B(t)$, so we know that $\frac{\tilde{\sigma}_0^2 \eta}{3} < |A(t+1)| < \frac{2\tilde{\sigma}_0^2 \eta}{3}$.

**Analysis of $R_{22}$:** For any $(A(t), B(t))$ in $R_{22}$, we have $|A(t+1)| > |B(t+1)|$. This means that $(A(t+1), B(t+1))$ remains in $R_{22}$, and we have $A(t+2) = A(t+1) - \text{sign}[A(t+1)]\tilde{\sigma}_0^2 \eta$ and $B(t+2) = B(t+1)$, which means that $(A(t+2), B(t+2))$ returns to the starting position at $(A(t), B(t))$. In fact, for any $t' \geq t$, $A(t')$ shows 2-periodic behavior switching between positive and negative signs and $B(t')$ remains constant, that is, for any $i \in \mathbb{Z}_{\geq 0}$, we have $A(t+2i) = A(t)$, $\text{sign}(A(t+2i)) = \text{sign}(A(t)) = -\text{sign}(A(t+2i+1))$, and $B(t+2i+1) = B(t+2i) = B(t)$.

**Analysis of $R_{23}$:** For any $(A(t), B(t))$ in $R_{23}$, by the definition of subpartition, we have that $|A(t+1)| < |B(t+1)|$, so $(A(t+1), B(t+1))$ is in $R_{21}$. This means that $(A(t+2), B(t+2))$ is in $R_3$ and we can analyze its dynamics using partition $R_3$.

**Analysis of $R_{31}$:** For any $(A(t), B(t))$ in $R_{31}$, we know that $A(t+1) = A(t) - \text{sign}[A(t)]\tilde{\sigma}_0^2 \eta$ and $B(t+1) = B(t)$. This means that $\frac{2\tilde{\sigma}_0^2 \eta}{3} < |A(t+1)| < \tilde{\sigma}_0^2 \eta$. Since $|A(t+1)| > |B(t+1)| = |B(t)|$, we have that $B(t+2) = B(t+1)$ and $A(t+2) = A(t+1) - \text{sign}[A(t+1)]\tilde{\sigma}_0^2 \eta$, which means that $(A(t+2), B(t+2))$ returns to the starting position at $(A(t), B(t))$. Therefore, for any $t' \geq t$, $A(t')$ shows 2-periodic behavior switching between positive and negative signs and $B(t')$ remains constant, that is, for any $i \in \mathbb{Z}_{\geq 0}$, we have $A(t+2i) = A(t)$, $\text{sign}(A(t+2i)) = \text{sign}(A(t)) = -\text{sign}(A(t+2i+1))$, and $B(t+2i+1) = B(t+2i) = B(t)$.

**Dynamics of $A$ in $R_{32}$, $R_{33}$ and $R_{34}$:** The behavior of $A$ in $R_{32}$, $R_{33}$ and $R_{34}$ is the same. For any $(A(t), B(t))$ in $\{R_{32}, R_{33}, R_{34}\}$, we know that

$$A(t+1) = A(t) - \text{sign}[A(t)]\frac{\tilde{\sigma}_0^2 \eta}{3} \quad \text{and} \quad B(t+1) = B(t) - \text{sign}[B(t)]\tilde{\sigma}_1^2 \eta. \tag{32}$$

This means that $|A(t+1)| < \frac{\tilde{\sigma}_0^2 \eta}{3}$, so $(A(t+1), B(t+1))$ remains in $R_3$. For any $t' \geq t$, $A(t')$ shows 2-periodic behavior switching between positive and negative signs, that is, for any $i \in \mathbb{Z}_{\geq 0}$, we have

$$A(t+2i) = A(t) \quad \text{and} \quad \text{sign}(A(t+2i)) = \text{sign}(A(t)) = -\text{sign}(A(t+2i+1)). \tag{33}$$

The behavior of $B$ is different across the three subpartitions, so we analyze them separately.

**Analysis of $R_{32}$:** Because all subpartitions are non-overlapping, for any $(A(t), B(t))$ in $R_{32}$, there exists a unique $k \in \mathbb{Z}_{\text{even}} - \{0\}$ such that $A(t)$ and $B(t)$ satisfies $|A(t)| > |B(t) + k\tilde{\sigma}_1^2 \eta|$. Next, we show that starting from any $(A(t), B(t))$ in $R_{32}$, after $|k|$ iterations of signGD update, we have $|A(t + |k|)| > |B(t + |k|)|$, which means that $(A(t + |k|), B(t + |k|))$ is in $R_{31}$.

This can be proved by showing that $|A(t + |k|)| = |A(t)|$ and $|B(t + |k|)| = |B(t) + k\tilde{\sigma}_1^2 \eta|$. For any $t' \geq t$, $A(t')$ shows 2-periodic behavior, and because $k$ is an even number, we have $|A(t + |k|)| = |A(t)|$.

Since $|B(t) + k\tilde{\sigma}_1^2 \eta| > 0$ and $B(t+1) = B(t) - \text{sign}[B(t)]\tilde{\sigma}_1^2 \eta$, we know that the sign of $B$ remains the same for the next $|k| - 1$ updates. This means that

$$B(t + |k|) = B(t) - \sum_{i=0}^{|k|-1} \text{sign}[B(t+i)]\tilde{\sigma}_1^2 \eta = B(t) - |k|\,\text{sign}[B(t)]\tilde{\sigma}_1^2 \eta.$$

By the definition of the subpartition, we have $\frac{\tilde{\sigma}_0^2 \eta}{3} > |B(t) + k\tilde{\sigma}_1^2 \eta|$; and this is true if and only if $B(t)$ and $k$ have opposite signs because $k$ is a non-zero even integer. Therefore, we have

$$\begin{aligned}
|B(t + |k|)| &= |B(t) - |k|\,\text{sign}[B(t)]\tilde{\sigma}_1^2 \eta| \\
&= |B(t) - |k|\,(-\text{sign}[k])\tilde{\sigma}_1^2 \eta| \\
&= |B(t) + |k|\,\text{sign}[k]\tilde{\sigma}_1^2 \eta| \\
&= |B(t) + k\tilde{\sigma}_1^2 \eta|.
\end{aligned}$$

**Analysis of $R_{33}$:** Similarly, for any $(A(t), B(t))$ in $R_{33}$, there exists a unique $k \in \mathbb{Z}_{\text{odd}}$ such that $A(t)$ and $B(t)$ satisfies $|A(t)| + |B(t) + k\tilde{\sigma}_1^2 \eta| < \frac{\tilde{\sigma}_0^2 \eta}{3}$. Again, we show that starting from any $(A(t), B(t))$ in $R_{33}$, after $|k|$ iterations of signGD update, $(A(t + |k|), B(t + |k|))$ is in $R_{31}$. This can be proved by showing that $\frac{\tilde{\sigma}_0^2 \eta}{3} - |A(t)| \leq |A(t + |k|)|$ and $|B(t) + k\tilde{\sigma}_1^2 \eta| = |B(t + |k|)|$.

First, by using the same analysis of $|B(t) + k\tilde{\sigma}_1^2 \eta|$ in $R_{32}$, we have $|B(t) + k\tilde{\sigma}_1^2 \eta| = |B(t + |k|)|$. Next, the behavior of $A(t)$ follows (32), which means that for any $t' \geq t$, $A(t')$ shows 2-periodic behavior, and because $k$ is an odd number, we have $|A(t + |k|) - A(t)| = |A(t + 1) - A(t)| = \frac{\tilde{\sigma}_0^2 \eta}{3}$.

Also, we have $|A(t + |k|) - A(t)| \leq |A(t + |k|)| + |A(t)|$, which means that $\frac{\tilde{\sigma}_0^2 \eta}{3} \leq |A(t + |k|)| + |A(t)|$, or $\frac{\tilde{\sigma}_0^2 \eta}{3} - |A(t)| \leq |A(t + |k|)|$. Combining with the definition of the subpartition, we have

$$|B(t) + |k|| < \frac{\tilde{\sigma}_0^2 \eta}{3} - |A(t)| \leq |A(t + |k|)|.$$

Therefore, starting from any $(A(t), B(t))$ in $R_{33}$, after $|k|$ iterations of signGD updates, we have $|A(t + |k|)| > |B(t + |k|)|$, which together with the fact that $|A(t + |k|)| < \frac{\tilde{\sigma}_0^2 \eta}{3}$ as shown before, implies that $(A(t + |k|), B(t + |k|))$ is in $R_{31}$.

**Analysis of $R_{34}$:** Finally, we prove, by contradiction, that for any $(A(t), B(t))$ in $R_{34}$, there is no $t' > t$ such that $(A(t'), B(t'))$ in $R_{31}$. Suppose that $(A(t'), B(t'))$ enters $R_{31}$, then $k \triangleq t' - t$ must be either an odd number or an even number. By the definition of the subpartition, if $k$ is a non-zero even number, it means that $(A(t), B(t))$ must be in $R_{32}$; whereas if $k$ is an odd number, it means that $(A(t), B(t))$ must be in $R_{33}$. Neither is possible since all subpartitions are non-overlapping, so for any $(A(t), B(t))$ in $R_{34}$, $(A(t'), B(t'))$ remains in $R_{34}$ for all $t' \geq t$. This means that there will always be a non-zero update on $B$, and this allows us to apply Lemma E.2. For any $(A(t), B(t))$ in $R_{34}$, there exists $t$ such that $|B(t')| \leq \tilde{\sigma}_1^2 \eta$ for all $t' \geq t$. $\quad\square$

Combining Proposition E.1 and the dynamics of $(A, B)$ in the stationary subpartitions described in Proposition E.4, we have the following remark.

**Remark E.5.** *The asymptotic solution of $A$ oscillates in $[-\frac{2\tilde{\sigma}_0^2\eta}{3}, \frac{2\tilde{\sigma}_0^2\eta}{3}]$, which is a tighter bound compared to the one in Proposition E.3. The asymptotic solution of $B$ either remains constant in $[-\tilde{\sigma}_1^2\eta, \tilde{\sigma}_1^2\eta]$ (1 and 2c in Proposition E.4) or oscillates in $[-\tilde{\sigma}_1^2\eta, \tilde{\sigma}_1^2\eta]$ (3c in Proposition E.4). Since $A(t)$ and $B(t)$ denote $\frac{\sqrt{3}}{3}\tilde{\sigma}_0^2\tilde{e}_0(t)$ and $\frac{\sqrt{2}}{2}\tilde{\sigma}_1^2\tilde{e}_1(t)$, respectively, this means that $\limsup_{t\to\infty}|\tilde{e}_0(t)| = \frac{2\sqrt{3}}{3}\eta$, and $\limsup_{t\to\infty}|\tilde{e}_1(t)| = \sqrt{2}\eta$,*

From the dynamics of $(A, B)$ in the transient subpartitions described in Proposition E.4, we have the following corollary.

**Corollary E.6.** *Suppose that $\tilde{e}_0$ enters $[-\sqrt{3}\eta, \sqrt{3}\eta]$ at iteration $t$, then $A$ starts exhibiting a 2-periodic oscillation at most 3 iterations after $t$. This means that the maximum difference between the number of positive and negative $A$'s after iteration $t$ is 2: $\left|\sum_{i=t+1}^{\infty} \mathbb{I}\{\text{sign}[A(i)] = 1\} - \mathbb{I}\{\text{sign}[A(i)] = -1\}\right| \le 2$.*

### E.8.2 Dynamics of $\tilde{e}_2$ under signGD

We are now ready to analyze the dynamics of $\tilde{e}_2$ through the behavior of $\tilde{e}_0$ and $\tilde{e}_1$. Particularly, we demonstrate that the final value of $|\tilde{e}_2|$ is affected by the magnitude of $|\tilde{e}_0(0)|$ and $|\tilde{e}_1(0)|$. First, notice that the update direction along $\tilde{e}_2$ follows the opposite of the sign of $\tilde{e}_0$, and at every iteration when $|A(t)| \le |B(t)|$, there is a non-zero weight adaptation for $\tilde{e}_2$. Once the oscillation begins for $\tilde{e}_0$, the dynamics of $\tilde{e}_0$ and $\tilde{e}_2$ become similar. Consider $T$ as the first iteration when $|\tilde{e}_0|$ drops below $\sqrt{3}\eta$. We then have

$$
\begin{aligned}
\limsup_{t\to\infty}|\tilde{e}_2(t)| &= \left|\tilde{e}_2(T) + \eta\sum_{t=T+1}^{\infty}\left\{\mathbb{I}\{|A(t)| < |B(t)|\}\frac{-\sqrt{6}}{3} + \mathbb{I}\{|A(t)| = |B(t)|\}\frac{-\sqrt{6}}{6}\right\}\text{sign}[\tilde{e}_0(t)]\right| \\
&\le |\tilde{e}_2(T)| + \frac{\sqrt{6}}{3}\eta\left|\sum_{t=T+1}^{\infty}\text{sign}[\tilde{e}_0(t)]\right| \\
&\le |\tilde{e}_2(T)| + \frac{2\sqrt{6}}{3}\eta,
\end{aligned}
\tag{34}
$$

where we use Corollary E.6 to upper bound the absolute value of the summation of the sign of $\tilde{e}_0$ after the $T$-th iteration in the last inequality. This means that after $T$ iterations, $\tilde{e}_2$ stays in an $O(\eta)$ neighborhood of $\tilde{e}_2(T)$; in other words, $\tilde{w}_2$ stays in an $O(\eta)$ neighborhood of $\tilde{w}_2(T)$. Also, notice that (34) does not include $\mathbb{I}\{|A(t)| > |B(t)|\}$ since $\tilde{e}_2$ is updated only when $|A(t)| \le |B(t)|$, as shown in Table 9.

Define $\Delta\tilde{w}_2$ as the sum of all the updates in $\tilde{w}_2$ up to the $T$-th iteration:

$$
\Delta\tilde{w}_2 \triangleq \eta\sum_{t=0}^{T-1}\left\{\mathbb{I}\{|A(t)| < |B(t)|\}\frac{-\sqrt{6}}{3} + \mathbb{I}\{|A(t)| = |B(t)|\}\frac{-\sqrt{6}}{6}\right\}\text{sign}[\tilde{e}_0(t)],
\tag{35}
$$

which leads to

$$
\limsup_{t\to\infty}|\tilde{w}_2(t)| = |\tilde{w}_2(T) + O(\eta)| = |\tilde{w}_2(0) + \Delta\tilde{w}_2 + O(\eta)|,
\tag{36}
$$

where $\tilde{w}_2(0)$ is the weight at initialization.

Putting (36) together with Remark E.5, the asymptotic solution found by signGD is

$$
\tilde{\boldsymbol{w}}^{\text{signGD}} = \left[\tilde{w}_0^*, \tilde{w}_1^*, \tilde{w}_2(0) + \Delta\tilde{w}_2\right]^\top + O(\eta).
\tag{37}
$$

From the perspective of training under the standard risk, the signGD solution is close to the optimum. Specifically, its standard risk is

$$
\begin{aligned}
\mathcal{R}_{\mathrm{s}}(\tilde{\boldsymbol{w}}^{\mathrm{signGD}}) &= \mathbb{E}\left[\ell(\tilde{X}, Y; \tilde{\boldsymbol{w}}^{\mathrm{signGD}})\right] \\
&= \frac{1}{2}\mathbb{E}\left[\left\langle \tilde{X},\, \tilde{\boldsymbol{w}}^{\mathrm{signGD}} - \tilde{w}^* \right\rangle^2\right] \\
&= \frac{1}{2}\left(\mathbb{E}\left[\tilde{X}_0^2\right](\tilde{\boldsymbol{w}}_0^{\mathrm{signGD}} - \tilde{w}_0^*)^2 + \mathbb{E}\left[\tilde{X}_1^2\right](\tilde{\boldsymbol{w}}_1^{\mathrm{signGD}} - \tilde{w}_1^*)^2\right) \\
&= \frac{1}{2}\left(\tilde{\sigma}_0^2 O(\eta^2) + \tilde{\sigma}_1^2 O(\eta^2)\right) \\
&= O((\tilde{\sigma}_0^2 + \tilde{\sigma}_1^2)\eta^2),
\end{aligned}
\tag{38}
$$

where $\mathbb{E}\left[\tilde{X}_0 \tilde{X}_1\right] = 0$ in (38) due to the diagonality of $\tilde{\Sigma}$. Note that the standard risk of the GD solution is exactly zero; and by choosing a small learning rate $\eta$, the standard risk of the signGD solution can be close to zero as well. However, their adversarial risks are very different. Specifically, the adversarial risk of the asymptotic signGD solution is

$$
\mathcal{R}_{\mathrm{a}}(\tilde{\boldsymbol{w}}^{\mathrm{signGD}}) = \frac{\epsilon^2}{2}||\tilde{\boldsymbol{w}}^{\mathrm{signGD}}||_2^2 = \frac{\epsilon^2}{2}\left\{\tilde{w}_0^{*2} + \tilde{w}_1^{*2} + (\tilde{w}_2(0) + \Delta\tilde{w}_2)^2 + O(\eta^2)\right\}.
\tag{39}
$$

Consider a sufficiently small learning rate: $\eta \ll \min\{\tilde{w}_0^*, \tilde{w}_1^*, \tilde{w}_2(0) + \Delta\tilde{w}_2\}$. This means that the contribution from $O(\eta^2)$ in $\mathcal{R}_{\mathrm{a}}(\tilde{\boldsymbol{w}}^{\mathrm{signGD}})$ is negligible. Then the adversarial risk of the signGD solution becomes

$$
\mathcal{R}_{\mathrm{a}}(\tilde{\boldsymbol{w}}^{\mathrm{signGD}}) = \frac{\epsilon^2}{2}\left\{\tilde{w}_0^{*2} + \tilde{w}_1^{*2} + (\tilde{w}_2(0) + \Delta\tilde{w}_2)^2\right\}.
\tag{40}
$$

We can compare it with the adversarial risk of the asymptotic solution found by GD under the same setup:

$$
\mathcal{R}_{\mathrm{a}}(\tilde{\boldsymbol{w}}^{\mathrm{GD}}) = \frac{\epsilon^2}{2}\left\{\tilde{w}_0^{*2} + \tilde{w}_1^{*2} + \tilde{w}_2^2(0)\right\}.
\tag{41}
$$

The main difference between the two adversarial risks in (40) and (41) is the difference in weights learned at the irrelevant frequency. Since their use of irrelevant frequency in the data is under-constrained, neither algorithm can reduce $\tilde{w}_2$ to zero, thereby neither solution is the most robust standard risk minimizer. The GD solution is sensitive to weight initialization. To understand the $\Delta\tilde{w}_2$ term in the signGD solution, first recall that $T$ denotes the first iteration when $|\tilde{e}_0|$ drops below $\sqrt{3}\eta$ (or $|A|$ drops below $\tilde{\sigma}_0^2\eta$), and from Corollary E.6 we know that $\tilde{w}_0$ starts oscillation at most 3 iterations after $T$. Recall in (36) that $O(\eta)$ has been utilized to account for the maximum sign variations, this means that we can consider oscillations which begin immediately after the $T$-th update. Suppose that $\eta$ is small so the sign of $\tilde{e}_0$ would not change before the oscillation starts, then we have

$$
\begin{aligned}
|\Delta\tilde{w}_2| &= \left|\eta\sum_{t=0}^{T-1}\left\{\mathbb{I}\{|A(t)| < |B(t)|\}\frac{-\sqrt{6}}{3} + \mathbb{I}\{|A(t)| = |B(t)|\}\frac{-\sqrt{6}}{6}\right\}\mathrm{sign}[\tilde{e}_0(t)]\right| \\
&= \left|\eta\sum_{t=0}^{T-1}\left\{\mathbb{I}\{|A(t)| < |B(t)|\}\frac{-\sqrt{6}}{3} + \mathbb{I}\{|A(t)| = |B(t)|\}\frac{-\sqrt{6}}{6}\right\}\right|,
\end{aligned}
$$

which leads to

$$
|\Delta\tilde{w}_2| = C\eta\sum_{t=0}^{T-1}\mathbb{I}\{|A(t)| \le |B(t)|\},
\tag{42}
$$

where $C$ denotes some value between $\frac{\sqrt{6}}{6}$ and $\frac{\sqrt{6}}{3}$, which correspond to always using the smaller and the larger updates, respectively.

### E.8.3 Dynamics of $|\Delta\tilde{w}_2|$ under signGD

There are two factors that can affect the magnitude of $\sum_{t=0}^{T-1}\mathbb{I}\{|A(t)| \le |B(t)|\}$ in (42): 1) the relative magnitudes between $\tilde{\sigma}_0^2$ and $\tilde{\sigma}_1^2$, and 2) the initial values of $|\tilde{e}_0|$ and $|\tilde{e}_1|$, or equivalently, the initial values of $|A|$ and $|B|$. To analyze this, we divide the set of values of $(|A(t)|, |B(t)|)$ into several partitions: the set of $[0, \tilde{\sigma}_0^2\eta] \times \mathbb{R}$ and $\mathbb{R} \times [0, \tilde{\sigma}_0^2\eta]$ is partitioned into $P_1$ and $P_2$, and the set of $[\tilde{\sigma}_0^2\eta, \infty) \times [\tilde{\sigma}_0^2\eta, \infty)$ is partitioned differently based on the value of $\frac{\tilde{\sigma}_1^2}{\tilde{\sigma}_0^2}$. Consider a line that travels through the point of $(\tilde{\sigma}_0^2\eta, \tilde{\sigma}_0^2\eta)$ and has a slope of $3\frac{\tilde{\sigma}_1^2}{\tilde{\sigma}_0^2}$. The ratio between $\tilde{\sigma}_0^2$ and $\tilde{\sigma}_1^2$ is particularly useful in analyzing $|\Delta\tilde{w}_2|$ because understanding the position of $(|A(0)|, |B(0)|)$ relative to such a line can lead to the value of $|B(T-1)|$, that is, the value of $|B|$ before the oscillation of $|A|$ begins. Since $|B|$ is updated only when $|A| \le |B|$, we have $\sum_{t=0}^{T-1}\mathbb{I}\{|A(t)| \le |B(t)|\} = \frac{|B(0)|-|B(T-1)|}{\tilde{\sigma}_1^2\eta}$. The definitions of partitions are

- $P_1 = \big\{ (A, B) : |A| < \tilde{\sigma}_0^2\eta \big\}$,

- $P_2 = \big\{ (A, B) : |A| > \tilde{\sigma}_0^2\eta \text{ and } |B| < \tilde{\sigma}_0^2\eta \big\}$,

- When $\frac{\tilde{\sigma}_1^2}{\tilde{\sigma}_0^2} > \frac{1}{3}$,

  - $P_3 = \Big\{ (A, B) : \tilde{\sigma}_0^2\eta < |A| < \frac{\tilde{\sigma}_0^2}{3\tilde{\sigma}_1^2}(|B| + (3\tilde{\sigma}_1^2 - \tilde{\sigma}_0^2)\eta) \Big\}$,
    - $P_{31} = \big\{ (A, B) : (A, B) \in P_3 \text{ and } |A| + \tilde{\sigma}_0^2\eta > |B| \big\}$,
  - $P_4 = \Big\{ (A, B) : \tilde{\sigma}_0^2\eta < |B| < \frac{3\tilde{\sigma}_1^2}{\tilde{\sigma}_0^2}|A| - (3\tilde{\sigma}_1^2 - \tilde{\sigma}_0^2)\eta \Big\}$,
    - $P_{41} = \big\{ (A, B) : (A, B) \in P_4 \text{ and } |B| < |A| < 2\tilde{\sigma}_0^2\eta \big\}$,
    - $P_{42} = \left\{ (A, B) : (A, B) \in P_4 \text{ and } |A| > |B| \text{ and } 2\tilde{\sigma}_0^2\eta < |A| < \frac{|B|+(3\tilde{\sigma}_1^2-\tilde{\sigma}_0^2)\eta}{3\frac{\tilde{\sigma}_1^2}{\tilde{\sigma}_0^2}} + \tilde{\sigma}_0^2\eta \right\}$,

- When $\frac{\tilde{\sigma}_1^2}{\tilde{\sigma}_0^2} < \frac{1}{3}$,

  - $P_5 = \big\{ (A, B) : \tilde{\sigma}_0^2\eta < |A| < |B| \big\}$,
  - $P_6 = \big\{ (A, B) : \tilde{\sigma}_0^2\eta < |B| < |A| \big\}$.

An illustration of partitions is provided in Figure 7, where the two plots demonstrate the two different ways of dividing the set of $[\tilde{\sigma}_0^2\eta, \infty) \times [\tilde{\sigma}_0^2\eta, \infty)$ based on the value of $\frac{\tilde{\sigma}_1^2}{\tilde{\sigma}_0^2}$. The connection between the values of $(|A(0)|, |B(0)|)$ and the size of $\sum_{t=0}^{T-1}\mathbb{I}\{|A(t)| \le |B(t)|\}$ is summarized in the next proposition.

**Proposition E.7.** *Denote $T$ as the iteration when $|\tilde{e}_0|$ drops below $\sqrt{3}\eta$. The value of $\sum_{t=0}^{T-1}\mathbb{I}\{|A(t)| \le |B(t)|\}$ depends on the relative magnitude between $\tilde{\sigma}_0^2$ and $\tilde{\sigma}_1^2$, and the initial values of $|A|$ and $|B|$. Specifically, we have*

$$\sum_{t=0}^{T-1}\mathbb{I}\{|A(t)| \le |B(t)|\} = \begin{cases} 0 & \text{if} \quad (|A(0)|, |B(0)|) \in (P_1 \bigcup P_2) \\ T & \text{if} \quad \frac{\tilde{\sigma}_1^2}{\tilde{\sigma}_0^2} > \frac{1}{3} \quad \text{and} \quad (|A(0)|, |B(0)|) \in P_3 \\ \frac{|B(0)|}{\tilde{\sigma}_1^2\eta} + [\frac{-2\tilde{\sigma}_0^2}{\tilde{\sigma}_1^2}, \frac{-\tilde{\sigma}_0^2}{\tilde{\sigma}_1^2}] & \text{if} \quad \frac{\tilde{\sigma}_1^2}{\tilde{\sigma}_0^2} > \frac{1}{3} \quad \text{and} \quad (|A(0)|, |B(0)|) \in P_4 \\ T & \text{if} \quad \frac{\tilde{\sigma}_1^2}{\tilde{\sigma}_0^2} < \frac{1}{3} \quad \text{and} \quad (|A(0)|, |B(0)|) \in P_5 \\ \frac{|B(0)|-\tilde{\sigma}_0^2\eta}{\frac{1}{3}\tilde{\sigma}_0^2\eta} & \text{if} \quad \frac{\tilde{\sigma}_1^2}{\tilde{\sigma}_0^2} < \frac{1}{3} \quad \text{and} \quad (|A(0)|, |B(0)|) \in P_6. \end{cases}$$

*Proof.*
We divide the analysis into two main parts: when $\frac{\tilde{\sigma}_1^2}{\tilde{\sigma}_0^2} > \frac{1}{3}$ and $\frac{\tilde{\sigma}_1^2}{\tilde{\sigma}_0^2} < \frac{1}{3}$, corresponding to the left and right figures in Figure 7. For each case, we analyze the behavior of $(A, B)$ within the partition.

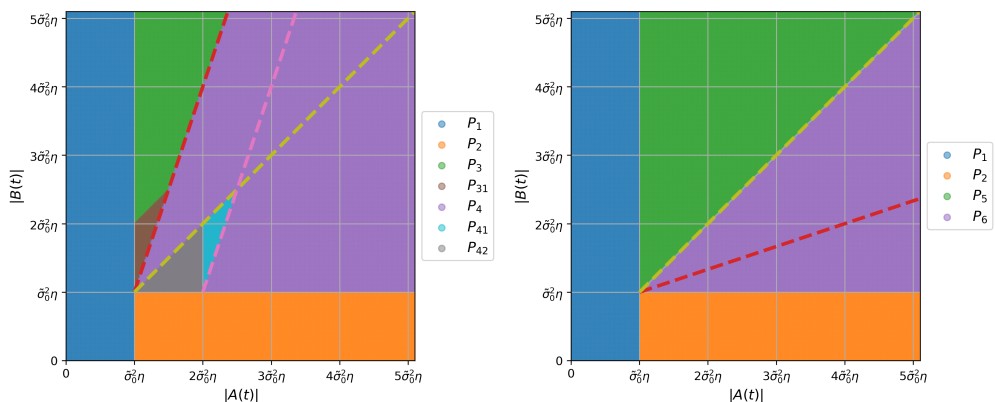

Figure 7: **Analyzing the value of $\sum_{t=0}^{T-1} \mathbb{I}\{|A(t)| \leq |B(t)|\}$ in (42) by partitioning the set of values of $(|A(t)|, |B(t)|)$, and the relative magnitude between $\tilde{\sigma}_0^2$ and $\tilde{\sigma}_1^2$ determines the partitions on which the analysis is based.** Specifically, the analysis is based on partitions $P_1$, $P_2$, $P_3$ and $P_4$, when $\frac{\tilde{\sigma}_1^2}{\tilde{\sigma}_0^2} > \frac{1}{3}$ (left), and on $P_1$, $P_2$, $P_5$ and $P_6$, when $\frac{\tilde{\sigma}_1^2}{\tilde{\sigma}_0^2} < \frac{1}{3}$ (right). The three smaller subpartitions are subsets of the main partition, i.e., $P_{31} \subset P_3$ and $P_{41}, P_{42} \subset P_4$, and they are used in the analysis of $P_4$. The value of $\sum_{t=0}^{T-1} \mathbb{I}\{|A(t)| \leq |B(t)|\}$ when $(|A(0)|, |B(0)|)$ is initialized in each partition is summarized in Proposition E.7. The two plots are created with $\tilde{\sigma}_0^2 = \tilde{\sigma}_1^2$ (left) and $\tilde{\sigma}_0^2 = 9\tilde{\sigma}_1^2$ (right), respectively. Note that those values are chosen for illustration purposes and do not affect the generality of the result. In both plots, the red dashed line corresponds to $|B(t)| = 3\frac{\tilde{\sigma}_1^2}{\tilde{\sigma}_0^2}|A(t)| - (3\tilde{\sigma}_1^2 - \tilde{\sigma}_0^2)\eta$ for $|A(t)| \in (\tilde{\sigma}_0^2\eta, \infty)$, and the yellow dashed line corresponds to $|B(t)| = |A(t)|$. The pink dashed line is parallel to the red dashed line with a horizontal gap of $\tilde{\sigma}_0^2\eta$.

**Analysis of $P_1$:** For any $(|A(0)|, |B(0)|)$ in $P_1$, since $|A(0)|$ is already below $\tilde{\sigma}_0^2\eta$, we have $T = 1$ because $A$ remains in $P_1$. This means that $\sum_{t=0}^{T-1} \mathbb{I}\{|A(t)| \leq |B(t)|\} = 0$.

**Analysis of $P_2$:** For any $(|A(0)|, |B(0)|)$ in $P_2$, $|A|$ decreases until it drops below $\tilde{\sigma}_0^2\eta$, while $|B|$ remains the same. This means that $|A|$ remains smaller than $|B|$, so we have $\mathbb{I}\{|A(t)| \leq |B(t)|\} = 0$ for all $t \in \{0, \ldots, T-1\}$. Therefore, we have $\sum_{t=0}^{T-1} \mathbb{I}\{|A(t)| \leq |B(t)|\} = 0$.

Next, the partitions of the set of $[\tilde{\sigma}_0^2\eta, \infty] \times [\tilde{\sigma}_0^2\eta, \infty]$ are defined differently based on the values of $\frac{\tilde{\sigma}_1^2}{\tilde{\sigma}_0^2}$ compared to $\frac{1}{3}$. This is because when $\frac{\tilde{\sigma}_1^2}{\tilde{\sigma}_0^2} > \frac{1}{3}$, it is possible for any $(|A(t)|, |B(t)|)$ satisfying $|A(t)| < |B(t)|$, there exists $t' > t$ such that $|A(t')| > |B(t')|$. In other words, $(|A|, |B|)$ can oscillate above and below the line defined by $|A| = |B|$, and this makes analyzing (42) difficult. However, when $\frac{\tilde{\sigma}_1^2}{\tilde{\sigma}_0^2} < \frac{1}{3}$, any $(|A(t)|, |B(t)|)$ that satisfies $|A(t)| < |B(t)|$ will stay above the line defined by $|A| = |B|$, and this means that $|A|$ will always get updated by $\frac{\tilde{\sigma}_0^2\eta}{3}$ and $|B|$ will always get updated by $\tilde{\sigma}_1^2\eta$. Because of this different behavior, we analyze these two cases separately by defining different partitions. This corresponds to the left and right figures in Figure 7. When $\frac{\tilde{\sigma}_1^2}{\tilde{\sigma}_0^2} > \frac{1}{3}$, the set of $[\tilde{\sigma}_0^2\eta, \infty] \times [\tilde{\sigma}_0^2\eta, \infty]$ is partitioned into $P_3$ and $P_4$.

**Analysis of $P_3$:** By definition, any $(|A(t)|, |B(t)|)$ in $P_3$ satisfies $3\frac{\tilde{\sigma}_1^2}{\tilde{\sigma}_0^2}|A(t)| < |B(t)| + (3\tilde{\sigma}_1^2 - \tilde{\sigma}_0^2)\eta$. Starting from any $(|A(0)|, |B(0)|)$ in $P_3$, the values of $|A|$ and $|B|$ decrease at a rate of $\frac{1}{3}\tilde{\sigma}_0^2\eta$ and $\tilde{\sigma}_1^2\eta$, respectively, and this means that two sides of the inequality decrease at the same rate. Hence, the sequence $(|A(t)|, |B(t)|)$ remains in $P_3$ for all $0 \leq t < T - 1$. This means that $\mathbb{I}\{|A(t)| \leq |B(t)|\} = 1$ for all $t \in \{0, \ldots, T-1\}$. Therefore, we have $\sum_{t=0}^{T-1} \mathbb{I}\{|A(t)| \leq |B(t)|\} = T$.

**Analysis of $P_4$:** Since $(|A(T-1)|, |B(T-1)|)$ must be in $P_1$, we can understand the value of $\sum_{t=0}^{T-1} \mathbb{I}\{|A(t)| \leq |B(t)|\}$ by considering how any $(|A(0)|, |B(0)|)$ in $P_4$ is transitioned to $(|A(T-1)|, |B(T-1)|)$ in $P_1$. Also, starting from any $(|A(t)|, |B(t)|)$ in $P_4$, we know that $|A(t)| - |A(t+1)| > 0$ and $|B(t)| - |B(t+1)| \geq 0$; hence, the transition from $P_4$ to $P_1$ must be described in

one of the following scenarios.

**Transition to $P_2$ then to $P_1$:** In this case, the value of $|B|$ must first drop below $\tilde{\sigma}_0^2 \eta$. Since $|B|$ decreases only when $|A| \leq |B|$, this means that, regardless of the initial value of $|A|$, the same number of updates is required to reduce $|B(0)|$ to $\tilde{\sigma}_0^2 \eta$, which is $\frac{|B(0)| - \tilde{\sigma}_0^2 \eta}{\tilde{\sigma}_1^2 \eta}$, and in each update, the condition $\mathbb{I}\{|A(t)| \leq |B(t)|\}$ is satisfied.

**Transition to $P_1$ directly:** For any $(|A(t)|, |B(t)|)$ in $P_4$ that satisfies $|B(t)| > |A(t)|$ (above the yellow dashed line in Figure 7), since the values of $|A|$ and $|B|$ decrease at a rate of $\frac{1}{3}\tilde{\sigma}_0^2 \eta$ and $\tilde{\sigma}_1^2 \eta$, respectively, $(|A(t+1)|, |B(t+1)|)$ cannot cross the red dashed line which has a slope of $3\frac{\tilde{\sigma}_1^2}{\tilde{\sigma}_0^2}$. Now let us consider any $(|A(t)|, |B(t)|)$ in $P_4$ that satisfies $|B(t)| < |A(t)|$ (below the yellow dashed line). In this case, $|A|$ decreases by $\tilde{\sigma}_0^2 \eta$, and the only scenario where $(|A(T-1)|, |B(T-1)|)$ ends up in $P_1$ is when $\tilde{\sigma}_0^2 \eta < |A(T-2)| < 2\tilde{\sigma}_0^2 \eta$. That is, $(|A(T-2)|, |B(T-2)|)$ is in $P_{42}$. When this happens, we have $\tilde{\sigma}_0^2 \eta < |B(T-2)| < 2\tilde{\sigma}_0^2 \eta$; and because there is no update in $|B(T-2)|$, we have $\tilde{\sigma}_0^2 \eta < |B(T-1)| < 2\tilde{\sigma}_0^2 \eta$. Therefore, we have $\sum_{t=0}^{T-1} \mathbb{I}\{|A(t)| \leq |B(t)|\} \in [\frac{|B(0)| - 2\tilde{\sigma}_0^2 \eta}{\tilde{\sigma}_1^2 \eta}, \frac{|B(0)| - \tilde{\sigma}_0^2 \eta}{\tilde{\sigma}_1^2 \eta}]$.

**Transition to $P_3$ then to $P_1$:** Let us first consider the transition from $P_4$ to $P_3$. Consider $t'$ such that $(|A(t)|, |B(t)|)$ is in $P_4$ for $0 \leq t < t'$ and $(|A(t')|, |B(t')|)$ is in $P_3$. Following the above analysis (direct transition to $P_1$), we know that $(|A(t'-1)|, |B(t'-1)|)$ must satisfy $|A(t'-1)| > |B(t'-1)|$, where $t'-1$ is the iteration before transitioning to $P_3$. Also, we know that $|A(t'-1)| > 2\tilde{\sigma}_0^2 \eta$ otherwise $(|A(t')|, |B(t')|)$ would be in $P_1$. The last condition for such a transition to happen is that the horizontal distance from $|B(t'-1)|$ to the line of $|B| = 3\frac{\tilde{\sigma}_1^2}{\tilde{\sigma}_0^2}|A| - (3\tilde{\sigma}_1^2 - \tilde{\sigma}_0^2)\eta$ (the red dashed line) must be smaller than $\tilde{\sigma}_0^2 \eta$. That is, $(|A(t'-1)|, |B(t'-1)|)$ is in $P_{41}$, and $(|A(t')|, |B(t')|)$ is in $P_{31}$. After the transition to $P_3$, the values of $|A|$ and $|B|$ decrease at a rate of $\frac{1}{3}\tilde{\sigma}_0^2 \eta$ and $\tilde{\sigma}_1^2 \eta$, respectively, and $|B(T-1)|$ has a range of $[\tilde{\sigma}_0 \eta, 2\tilde{\sigma}_0 \eta]$. Therefore, we have $\sum_{t=0}^{T-1} \mathbb{I}\{|A(t)| \leq |B(t)|\} \in [\frac{|B(0)| - 2\tilde{\sigma}_0^2 \eta}{\tilde{\sigma}_1^2 \eta}, \frac{|B(0)| - \tilde{\sigma}_0^2 \eta}{\tilde{\sigma}_1^2 \eta}]$.

When $\frac{\tilde{\sigma}_1^2}{\tilde{\sigma}_0^2} < \frac{1}{3}$, the set of $[\tilde{\sigma}_0^2 \eta, \infty] \times [\tilde{\sigma}_0^2 \eta, \infty]$ is partitioned into $P_5$ and $P_6$, as shown in the right figure of Figure 7.

**Analysis of $P_5$:** Starting from any $(|A(0)|, |B(0)|)$ in $P_5$, the values of $|A|$ and $|B|$ decrease at a rate of $\frac{1}{3}\tilde{\sigma}_0^2 \eta$ and $\tilde{\sigma}_1^2 \eta$, respectively. However, since $\frac{\tilde{\sigma}_1^2}{\tilde{\sigma}_0^2} < \frac{1}{3}$, there will not be any $0 \leq t \leq T - 1$ where $|A(t)| > |B(t)|$. This means that $\mathbb{I}\{|A(t)| \leq |B(t)|\} = 1$ for all $t \in \{0, \ldots, T-1\}$. Therefore, we have $\sum_{t=0}^{T-1} \mathbb{I}\{|A(t)| \leq |B(t)|\} = T$.

**Analysis of $P_6$:** Starting from any $(|A(0)|, |B(0)|)$ in $P_6$, the values of $|A|$ decreases until it becomes smaller than $|B(0)|$. Suppose that this happens at iteration $t'$, that is, $|A(t')| < |B(0)|$. Starting from $(|A(t')|, |B(t')|)$ in $P_5$, $|A|$ starts to decrease by $\frac{\tilde{\sigma}_0^2 \eta}{3}$ and $|B|$ starts to decrease by $\tilde{\sigma}_1^2 \eta$. Since $\frac{\tilde{\sigma}_1^2}{\tilde{\sigma}^2} < \frac{1}{3}$, this means that $(|A(t)|, |B(t)|)$ stays in $P_5$ for $t \in \{t', \ldots, T-2\}$, until it goes to $P_1$ when $|A(T-1)| < \tilde{\sigma}_0^2 \eta$. Therefore, we know that the total change in $|A|$ since $t'$-th iteration is $|A(t')| - \tilde{\sigma}_0^2 \eta = |B(0)| - \tilde{\sigma}_0^2 \eta$. Since $|A|$ can only be updated by the amount of $\frac{\tilde{\sigma}_0^2 \eta}{3}$ in $P_5$, we have $\sum_{t=0}^{T-1} \mathbb{I}\{|A(t)| \leq |B(t)|\} = \frac{|B(0)| - \tilde{\sigma}_0^2 \eta}{\frac{1}{3}\tilde{\sigma}_0^2 \eta}$. $\qquad \square$

We now use this analysis on the behavior of $\sum_{t=0}^{T-1} \mathbb{I}\{|A(t)| \leq |B(t)|\}$ to compute $|\Delta \tilde{w}_2|$, which plays a role in the adversarial risk of signGD, as shown in (40). For the initial values of $(|A|, |B|)$ to be in $P_1$ and $P_2$, the initial errors must be small. However, consider a dataset with a strong task-relevant correlation between the relevant frequency component of the data and the target, a realistic scenario as we discussed in Sec. 3.2. In this case, $|\tilde{w}_0^*|$ and $|\tilde{w}_1^*|$ can be large. Additionally, with a weight initialization around zero, such as in methods by He et al. (2015) and Glorot & Bengio (2010), the initial error $|\tilde{e}_0(0)|$ and $|\tilde{e}_1(0)|$ can be large and close to $|\tilde{w}_0^*|$ and $|\tilde{w}_1^*|$ when $|\tilde{w}_0^*| \gg |\tilde{w}_0(0)|$ and $|\tilde{w}_1^*| \gg |\tilde{w}_1(0)|$. Because of this, it is less likely for the initial values of $|A(0)|$ and $|B(0)|$ to be in the $P_1$ partition in Proposition E.7.

Moreover, it is discussed in Sec. 3.1 and later supported empirically in Figure 8 of Appendix G that the distribution of spectral energy heavily concentrates at the low end of the frequency spectrum and decays quickly towards higher frequencies. Since $\tilde{\sigma}_i^2$ is interpreted as the expected energy of a random variable at

the $i$-th frequency, it is reasonable to expect that $\frac{\tilde{\sigma}_1^2}{\tilde{\sigma}_0^2} < \frac{1}{3}$ and this allows us to further narrow down to initialization of $(|A|, |B|)$ in $P_5$ and $P_6$.

The proportional relationship between the size of (42) and the magnitude of $|\tilde{e}_0|$ and $|\tilde{e}_1|$ when $\frac{\tilde{\sigma}_1^2}{\tilde{\sigma}_0^2} < \frac{1}{3}$ and $(|A|, |B|)$ is initialized in $P_5$ or $P_6$ can be described in the following proposition.

**Proposition E.8.** *Suppose that the ratio between $\tilde{\sigma}_0^2$ and $\tilde{\sigma}_1^2$ satisfies $\frac{\tilde{\sigma}_1^2}{\tilde{\sigma}_0^2} < \frac{1}{3}$. The magnitude of $|\Delta\tilde{w}_2|$ depends on the initial values of $|\tilde{e}_0|$ and $|\tilde{e}_1|$, and the resulting $|A(0)|$ and $|B(0)|$. Specifically, we have*

$$|\Delta\tilde{w}_2| = \begin{cases} \sqrt{3}C |\tilde{e}_0(0)| & if \quad |A(0)| < |B(0)| \\ \frac{3\sqrt{2}\tilde{\sigma}_1^2}{2\tilde{\sigma}_0^2}C |\tilde{e}_1(0)| & if \quad |A(0)| > |B(0)|, \end{cases} \tag{43}$$

*where $C \in [\frac{\sqrt{6}}{6}, \frac{\sqrt{6}}{3}]$ and we neglect the contribution from $\eta$.*

*Proof.* From Proposition E.7, under the assumption that $\frac{\tilde{\sigma}_1^2}{\tilde{\sigma}_0^2} < \frac{1}{3}$, we have $\sum_{t=0}^{T-1} \mathbb{I}\{|A(t)| \le |B(t)|\} = T$ when $(|A(0)|, |B(0)|) \in P_5$, and this means that $|\Delta\tilde{w}_2| = C\eta T$ from (42). This also implies that for $t \in \{0, \ldots, T-1\}$, we have $|A(t)| < |B(t)|$ and $|A(t)| = |A(0)| - \frac{t}{3}\tilde{\sigma}_0^2\eta$.

Since $T$ is defined as the number of iteration required to reduce $|A(0)|$ to $\tilde{\sigma}_0^2\eta$, $T$ is $\frac{|A(0)| - \tilde{\sigma}_0^2\eta}{\frac{1}{3}\tilde{\sigma}_0^2\eta}$, and we have

$$|\Delta\tilde{w}_2| = C\eta T = C\eta \frac{|A(0)| - \tilde{\sigma}_0^2\eta}{\frac{1}{3}\tilde{\sigma}_0^2\eta} = 3C\frac{\frac{\sqrt{3}}{3}\tilde{\sigma}_0^2 |\tilde{e}_0(0)| - \tilde{\sigma}_0^2\eta}{\tilde{\sigma}_0^2} = C(\sqrt{3} |\tilde{e}_0(0)| - 3\eta).$$

From Proposition E.7, when $\frac{\tilde{\sigma}_1^2}{\tilde{\sigma}_0^2} < \frac{1}{3}$ and $(|A(0)|, |B(0)|)$ is in $P_6$, we have

$$|\Delta\tilde{w}_2| = C\eta \frac{|B(0)| - \tilde{\sigma}_0^2\eta}{\frac{1}{3}\tilde{\sigma}_0^2\eta} = 3C\frac{\frac{\sqrt{2}}{2}\tilde{\sigma}_1^2 |\tilde{e}_1(0)| - \tilde{\sigma}_0^2\eta}{\tilde{\sigma}_0^2} = C(\frac{3\sqrt{2}\tilde{\sigma}_1^2}{2\tilde{\sigma}_0^2} |\tilde{e}_1(0)| - 3\eta).$$

$\square$

Since the initial error $|\tilde{e}_0(0)|$ and $|\tilde{e}_1(0)|$ are close to $|\tilde{w}_0^*|$ and $|\tilde{w}_1^*|$, (43) can be written as

$$|\Delta\tilde{w}_2| \approx \begin{cases} \sqrt{3}C |\tilde{w}_0^*| & if \quad |A(0)| < |B(0)| \\ \frac{3\sqrt{2}\tilde{\sigma}_1^2}{2\tilde{\sigma}_0^2}C |\tilde{w}_1^*| & if \quad |A(0)| > |B(0)| \end{cases} \tag{44}$$

Now we can consider the ratio between the adversarial risk of the standard risk minimizers found by GD (41) and signGD (40) with a three-dimensional input space. We observe that the solution found by signGD is more sensitive to perturbations compared to the GD solution:

$$\frac{\mathcal{R}_a(\tilde{\boldsymbol{w}}^{\text{signGD}})}{\mathcal{R}_a(\tilde{\boldsymbol{w}}^{\text{GD}})} = \frac{\tilde{w}_0^{*2} + \tilde{w}_1^{*2} + (\tilde{w}_2(0) + \Delta\tilde{w}_2)^2}{\tilde{w}_0^{*2} + \tilde{w}_1^{*2} + \tilde{w}_2^2(0)} \approx 1 + \frac{\Delta\tilde{w}_2^2}{\tilde{w}_0^{*2} + \tilde{w}_1^{*2}},$$

where we neglect the contribution from $\tilde{w}_2(0)$ in the approximation since we have assumed that the values of $|\tilde{w}_0^*|$ and $|\tilde{w}_1^*|$ are large compared to the initialized weight $|\tilde{w}(0)_2|$. This leads to

$$\frac{\mathcal{R}_a(\tilde{\boldsymbol{w}}^{\text{signGD}})}{\mathcal{R}_a(\tilde{\boldsymbol{w}}^{\text{GD}})} \approx \begin{cases} 1 + C_3\frac{\tilde{w}_0^{*2}}{\tilde{w}_0^{*2} + \tilde{w}_1^{*2}} & if \quad |A(0)| < |B(0)| \\ 1 + C_4\frac{\tilde{w}_1^{*2}}{\tilde{w}_0^{*2} + \tilde{w}_1^{*2}} & if \quad |A(0)| > |B(0)|, \end{cases}$$

where $\frac{1}{2} \le C_3 \le 2$ and $\frac{3}{4}\frac{\tilde{\sigma}_1^4}{\tilde{\sigma}_0^4} \le C_4 \le 3\frac{\tilde{\sigma}_1^4}{\tilde{\sigma}_0^4}$.

### E.9 From Irrelevant Frequencies to Spatially Redundant Dimensions

We have demonstrated that when the use of irrelevant frequency is under-constrained, optimizing the standard training objective can lead to solutions with zero standard risk but are sensitive to perturbations. This section offers a spatial interpretation of the findings, where we illustrate that signals with irrelevant frequencies contain spatially redundant dimensions when transformed into the spatial domain. Both interpretations can be used to explain the vulnerability of the solutions.

To illustrate the concept of redundancy in the spatial domain, consider the synthetic dataset with the distribution defined in Sec. 4.2.2 and the data has a structure of $\left\{(\tilde{X}_0, \tilde{X}_1, 0)\right\}$ in the frequency domain. Taking the DCT transformation of $\tilde{X}$, we see that the spatial representation of the same dataset is

$$\left\{(\sqrt{\frac{1}{3}}\tilde{X}_0 + \sqrt{\frac{1}{2}}\tilde{X}_1, \sqrt{\frac{1}{3}}\tilde{X}_0, \sqrt{\frac{1}{3}}\tilde{X}_0 - \sqrt{\frac{1}{2}}\tilde{X}_1)\right\},$$

where $\tilde{X}_0$ and $\tilde{X}_1$ are random variables with frequency interpretations. In the spatial domain, redundancy refers to the existence of dimensions that are highly correlated with each other. The example mentioned above illustrates that the presence of a single irrelevant frequency in the data distribution corresponds to the existence of one redundant dimension in the spatial domain. Specifically, within this three-dimensional dataset, it is possible to express any dimension as a linear combination of the values at the other two dimensions.

This translation between spectral irrelevance and spatial redundancy can also be observed in the learned weight. Consider a standard risk minimizer $\tilde{w}^* = (\tilde{w}_0^*, \tilde{w}_1^*, 0)$, whose frequency-domain representation is

$$w^* = (\sqrt{\frac{1}{3}}w_0^* + \sqrt{\frac{1}{2}}w_1^*, \sqrt{\frac{1}{3}}w_0^*, \sqrt{\frac{1}{3}}w_0^* - \sqrt{\frac{1}{2}}w_1^*).$$

Because of the irrelevance from $\tilde{w}_2$, there are multiple other standard risk minimizers. In the spatial domain, this means $w^* + \tilde{w}_2\vec{w}_2$ with $\vec{w}_2 = (\sqrt{\frac{1}{6}}, -\sqrt{\frac{2}{3}}, \sqrt{\frac{1}{6}})$ and any choice of $\tilde{w}_2 \in \mathbb{R}$ is still a valid standard risk minimizer.[5] When the model trained by signGD has a large weight at $\tilde{w}_2$, this implies a large $\tilde{w}_2$ for the weight in the spatial domain. Because $\vec{w}_2$ and $w^*$ are orthogonal, we have $\|w^* + \tilde{w}_2\vec{w}_2\|_2 = \|w^*\|_2 + |\tilde{w}_2|$, therefore, the weight norm increases as $\tilde{w}_2$ gets large, and from (12), models are more vulnerable.

It is important to realize that having irrelevant frequencies is merely a sufficient condition for having spatially redundant features, but is not a necessary condition. For example, rearranging the dimensions of $x$ and $w^*$ in the above example still preserves the spatial redundancy in the dataset, and there are still infinitely many standard risk minimizers. However, it no longer guarantees zero entries in $\tilde{x}$ and $\tilde{w}^*$.

## F  Future Direction: Studying Model Robustness under Different Optimization Objectives

The Sharpness-Aware Minimization (SAM) objective, proposed by Foret et al. (2021), has demonstrated improvements in model robustness both in settings with noisy training labels and against adversarial perturbations (Wei et al., 2023).

Understanding the dynamics of the sharpness-aware loss, especially under different optimization algorithms, can be more involved. Without doing so, notice that the SAM objective in Foret et al. (2021) includes an $\ell_2$ regularization term on the weight norm. That is, training with the SAM objective penalizes models for having large weight norms. This is in line with our findings presented in Sec. 4, where we demonstrate that a minimum norm standard risk minimizer achieves the most robust standard risk minimizer.

Recent work by Wei et al. (2023) focused on linear models with classification and demonstrated that minimizing $\ell^{SAM}$ alone can lead to adversarially robust models. They designed a synthetic dataset based on the

---

[5]The $(\sqrt{\frac{1}{6}}, -\sqrt{\frac{2}{3}}, \sqrt{\frac{1}{6}})$ vector is the DCT basis for the $\tilde{w}_2$ term, i.e., $C^\top(0, 0, \tilde{w}_2) = \tilde{w}_2(\sqrt{\frac{1}{6}}, -\sqrt{\frac{2}{3}}, \sqrt{\frac{1}{6}})$.

hypothesis of the robust and non-robust features (Ilyas et al., 2019), and theoretically demonstrated on the linear classification that minimizing the sharpness-aware loss alone can result in models with larger weight on the robust features.

An important distinction to highlight between our analysis and that of Wei et al. (2023) is that, while both work theoretically analyze the adversarial robustness of linear models, our work focuses on models obtained via different optimization **algorithms**, while Wei et al. (2023) focuses on models under different optimization **objectives**. In our setting, under the same optimization objective, there exist multiple optimal solutions where their standard risks are identical, but their adversarial risks are different. On the other hand, in the setting of Wei et al. (2023), each objective has its own optimal solution. These solutions differ not just in adversarial robustness but also in their standard generalization performance. The two directions –optimization objectives and algorithms– are orthogonal, and the choice of an objective is independent of the choice of optimization algorithm. Understanding how models, trained under robustification objectives, behave when paired with various optimization algorithms is a promising avenue for future directions.

## G   Additional figures

In Figure 8, we visualize the energy distribution of various datasets containing natural images. Each dataset contains four plots. The $(i, j)$ coordinate in the first plot represents $\frac{1}{N} \sum_{n=1}^{N} |\tilde{x}_{n;(i,j)}|$, where $N$ is the number of training images, $\tilde{x}_n$ is the DCT transformation of $x_n$, and $\tilde{x}_{n;(i,j)}$ denotes the amplitude of the $(i, j)$-th basis in the $n$-th sample. In the second plot, we visualize the diagonal values from the first plot: $\left\{ \frac{1}{N} \sum_{n=1}^{N} |\tilde{x}_{n;(i,i)}| \right\}_{i=0,\ldots,d-1}$. We observe across all datasets that there is a high concentration of energy in the low-frequency harmonics and the amplitude of the higher-frequency harmonics becomes almost negligible. Therefore, we repeat the first two plots in the natural log scale ($\log_e$). The $(i, j)$ coordinate in the third plot represents $\frac{1}{N} \sum_{n=1}^{N} \log |\tilde{x}_{n;(i,j)}|$. In the fourth plot, we visualize $\left\{ \frac{1}{N} \sum_{n=1}^{N} \log |\tilde{x}_{n;(i,i)}| \right\}_{i=0,\ldots,d-1}$.

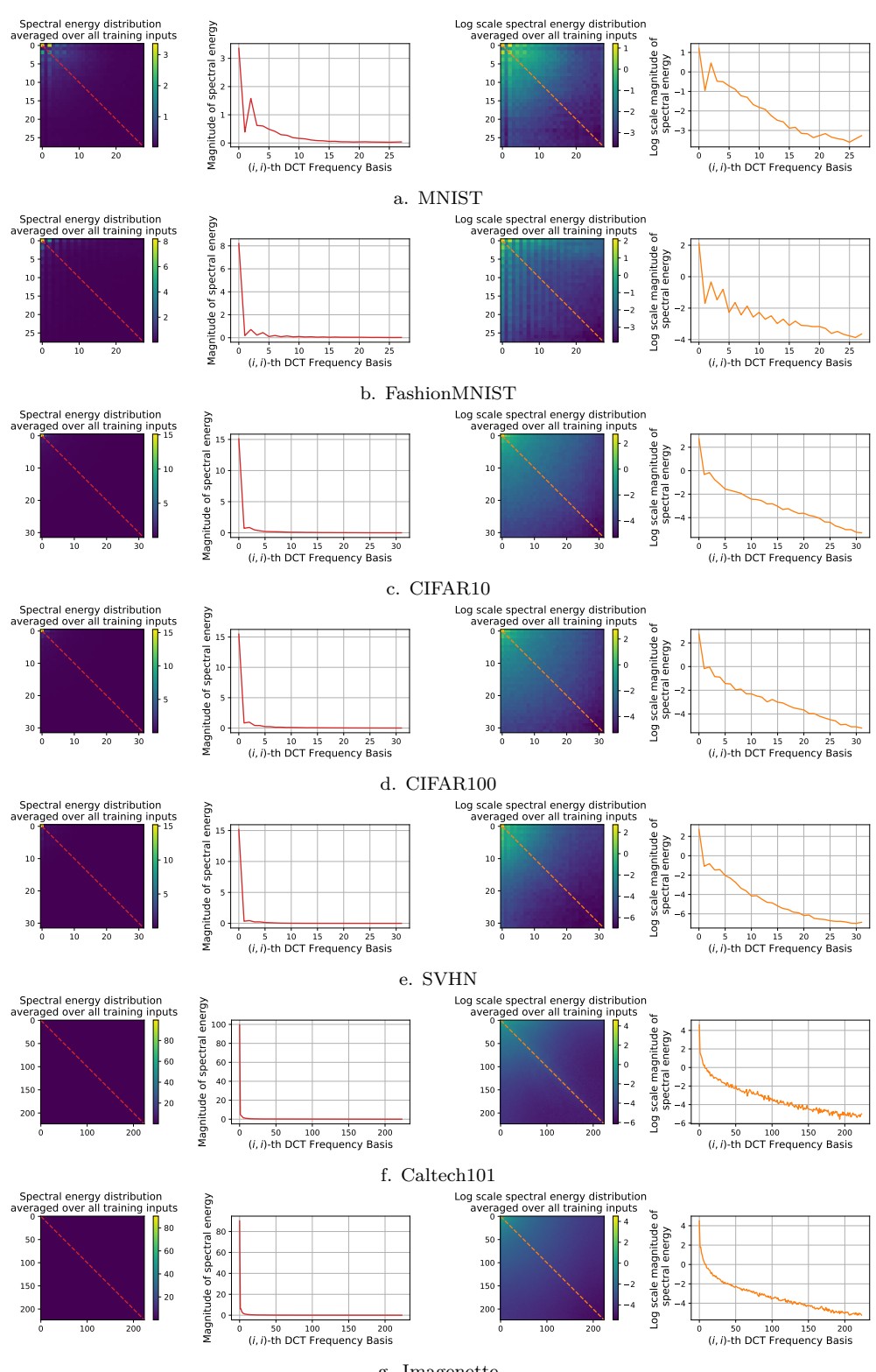

Figure 8: **Illustration of the spectral energy distribution in natural data.** Distribution of the spectral energy heavily concentrates at low frequencies and decays exponentially towards higher frequencies.

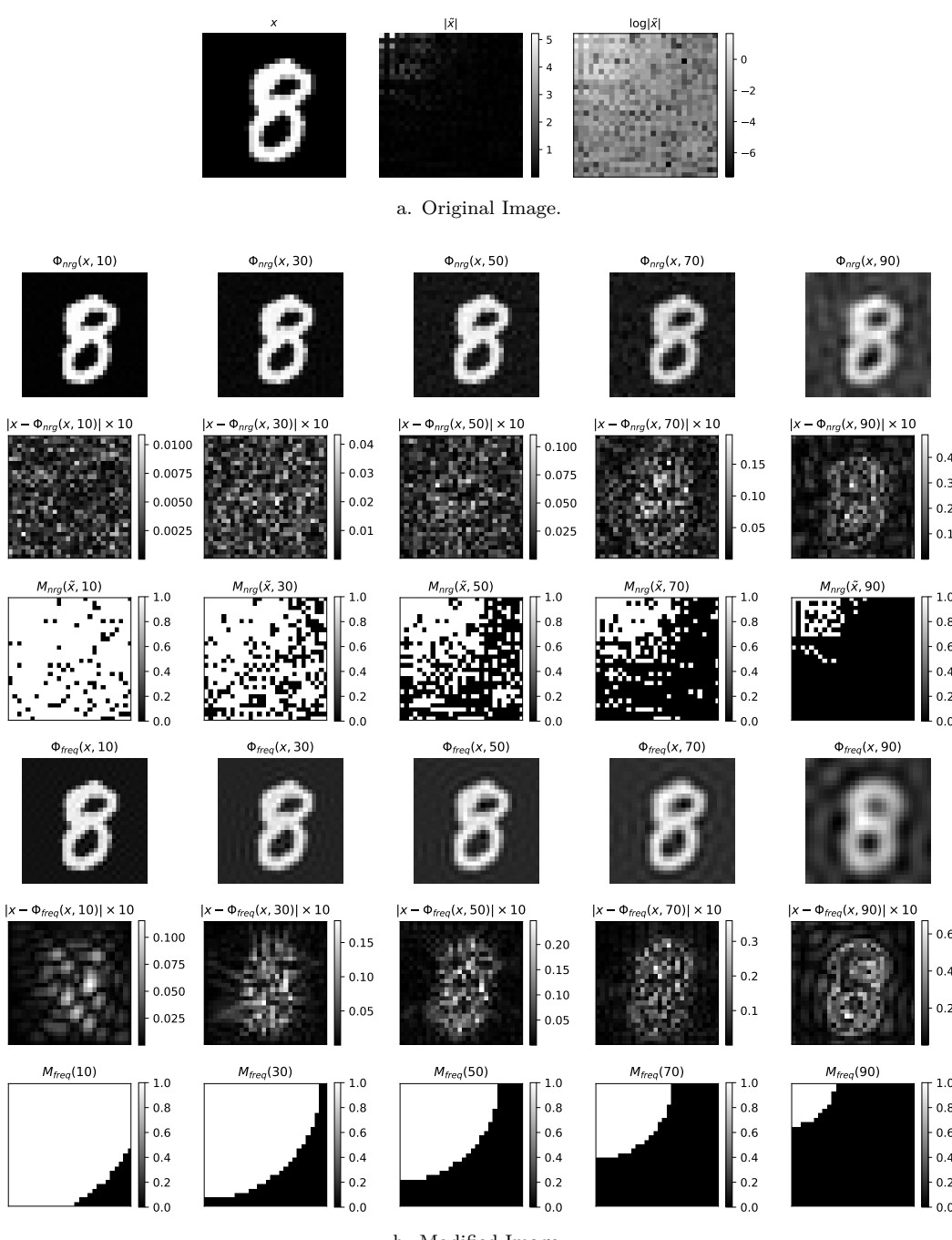

Figure 9: **Examples of modified images used in Observation I. (MNIST)** We use a threshold value of `threshold = {10, 30, 50, 70, 90}` to modify images based on its magnitude of DCT basis and their freqequency basis. In a), we show the original image $x$ and the magnitude of its DCT basis $|\tilde{x}|$ in both linear and log scale. In b), we show images modified by removing DCT basis vectors whose magnitudes are in the bottom `threshold` percentage (row 1), the differences between the modified images and the original image (row 2), the binary mask used to remove the DCT basis: black means removed (row 3), images modified by removing high-frequency DCT basis vectors (row 4), the differences between the modified images and the original image (row 5) and the binary mask used to remove the DCT basis: black means removed (row 6). Notice that the masks in row 6 only depends on the dimension of the images, whereas the masks in row 3 differs from images to images.

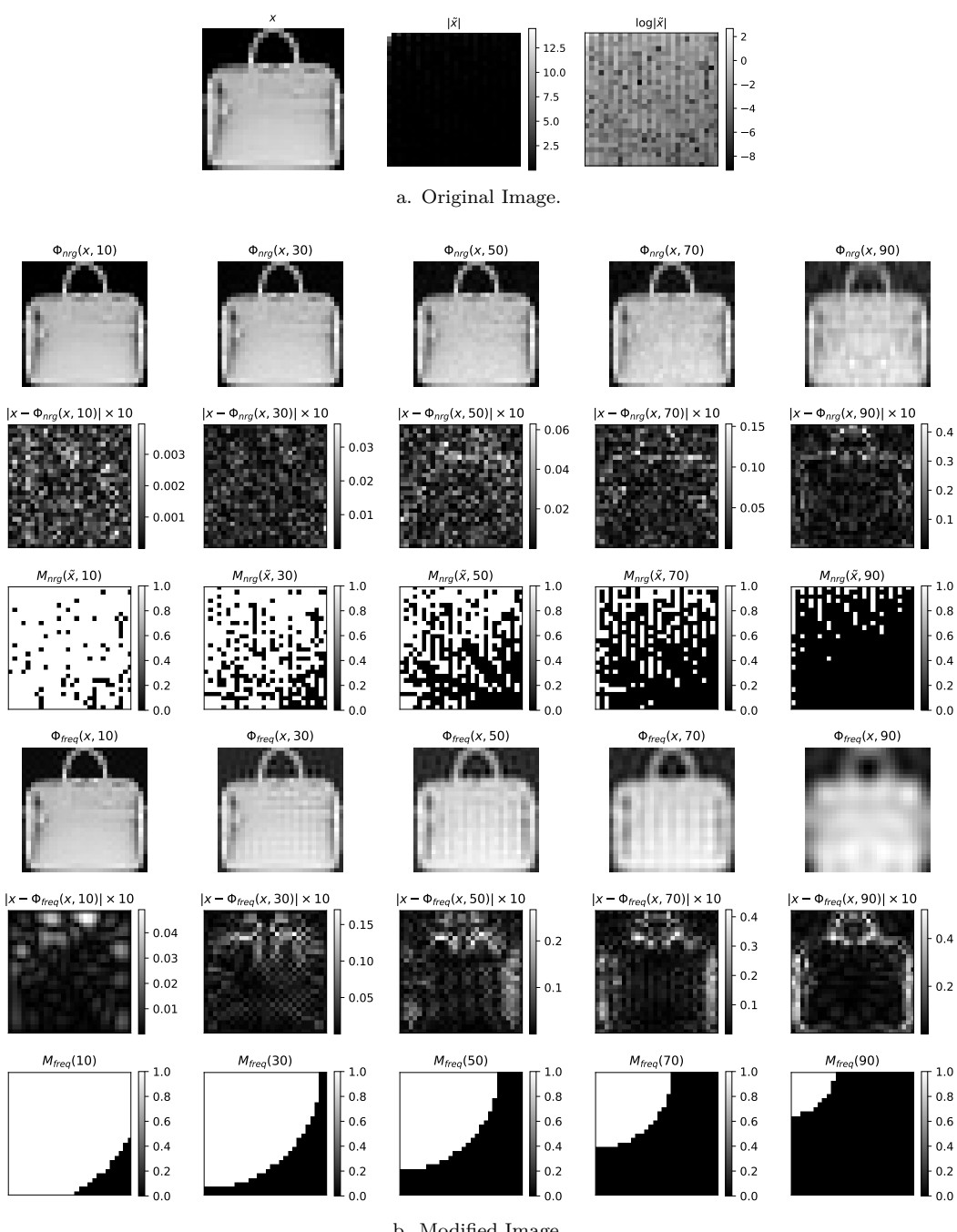

Figure 10: **Examples of modified images used in Observation I. (FashionMNIST)** We use a threshold value of `threshold = {10, 30, 50, 70, 90}` to modify images based on its magnitude of DCT basis and their freqequency basis. In a), we show the original image $x$ and the magnitude of its DCT basis $|\tilde{x}|$ in both linear and log scale. In b), we show images modified by removing DCT basis vectors whose magnitudes are in the bottom `threshold` percentage (row 1), the differences between the modified images and the original image (row 2), the binary mask used to remove the DCT basis: black means removed (row 3), images modified by removing high-frequency DCT basis vectors (row 4), the differences between the modified images and the original image (row 5) and the binary mask used to remove the DCT basis: black means removed (row 6). Notice that the masks in row 6 only depends on the dimension of the images, whereas the masks in row 3 differs from images to images.

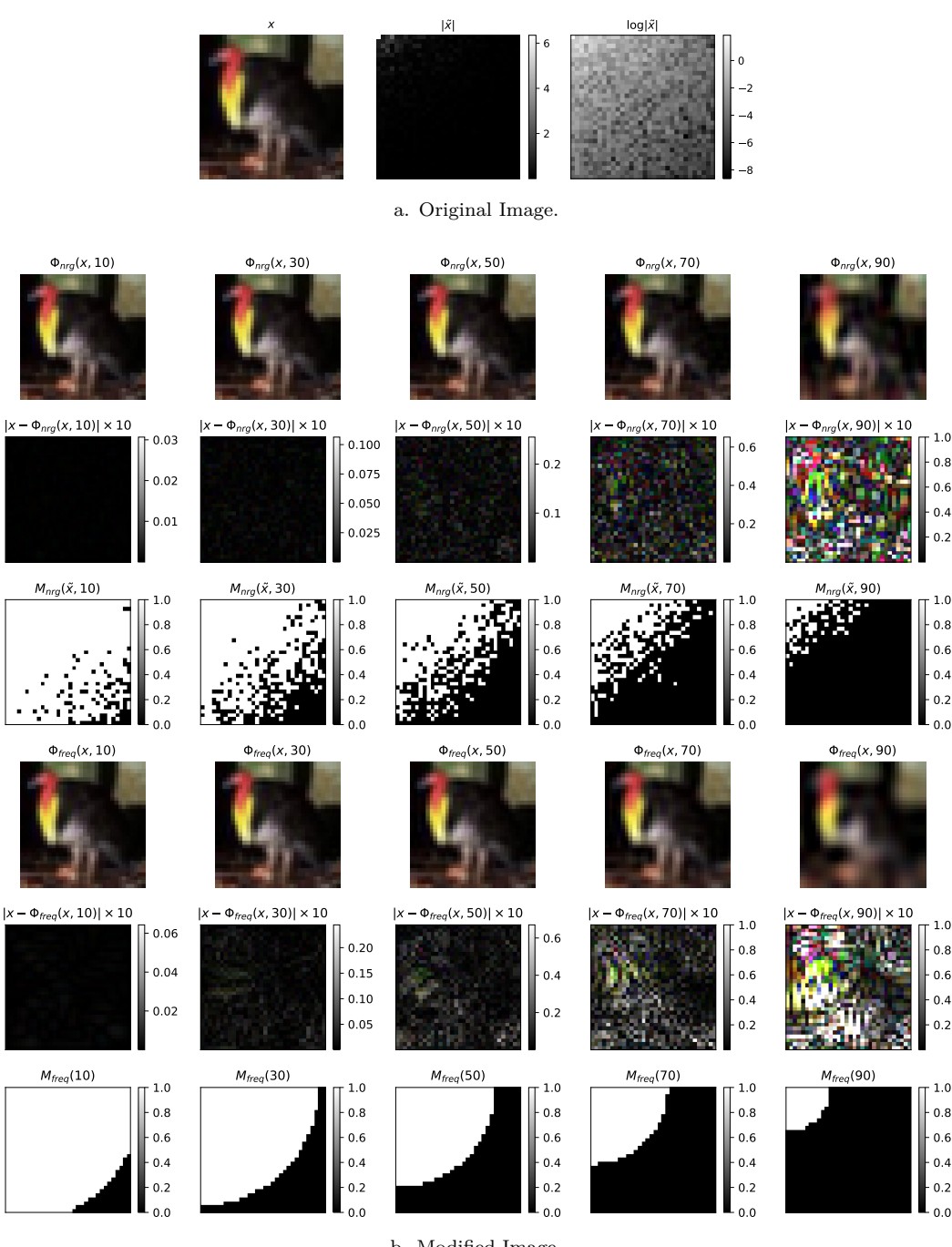

Figure 11: **Examples of modified images used in Observation I. (CIFAR10)** We use a threshold value of `threshold` $= \{10, 30, 50, 70, 90\}$ to modify images based on its magnitude of DCT basis and their freqequency basis. In a), we show the original image $x$ and the magnitude of its DCT basis $|\tilde{x}|$ in both linear and log scale. In b), we show images modified by removing DCT basis vectors whose magnitudes are in the bottom `threshold` percentage (row 1), the differences between the modified images and the original image (row 2), the binary mask used to remove the DCT basis: black means removed (row 3), images modified by removing high-frequency DCT basis vectors (row 4), the differences between the modified images and the original image (row 5) and the binary mask used to remove the DCT basis: black means removed (row 6). Notice that the masks in row 6 only depends on the dimension of the images, whereas the masks in row 3 differs from images to images.

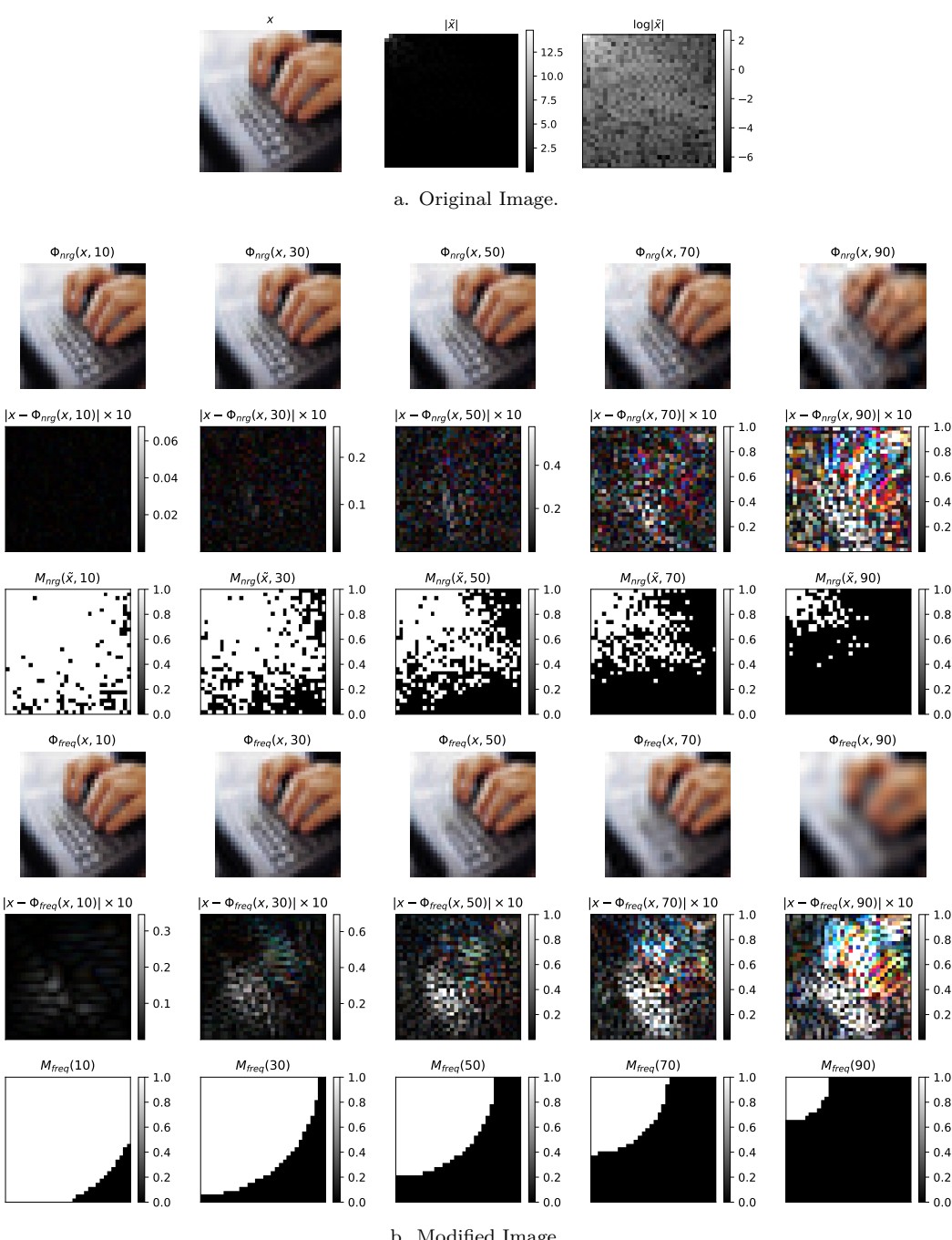

Figure 12: **Examples of modified images used in Observation I. (CIFAR100)** We use a threshold value of `threshold = {10, 30, 50, 70, 90}` to modify images based on its magnitude of DCT basis and their freqequency basis. In a), we show the original image $x$ and the magnitude of its DCT basis $|\tilde{x}|$ in both linear and log scale. In b), we show images modified by removing DCT basis vectors whose magnitudes are in the bottom `threshold` percentage (row 1), the differences between the modified images and the original image (row 2), the binary mask used to remove the DCT basis: black means removed (row 3), images modified by removing high-frequency DCT basis vectors (row 4), the differences between the modified images and the original image (row 5) and the binary mask used to remove the DCT basis: black means removed (row 6). Notice that the masks in row 6 only depends on the dimension of the images, whereas the masks in row 3 differs from images to images.

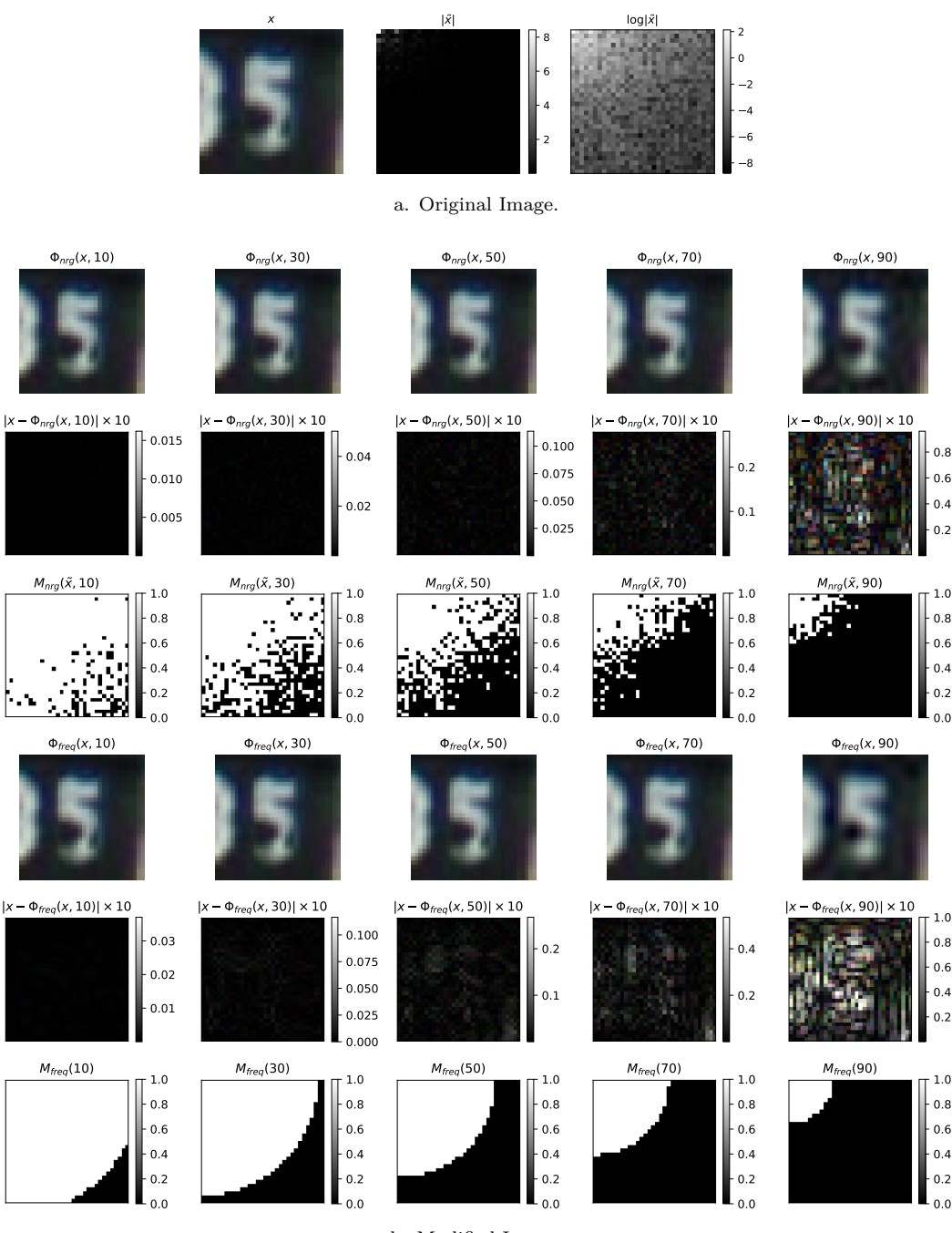

Figure 13: **Examples of modified images used in Observation I. (SVHN)** We use a threshold value of `threshold = {10, 30, 50, 70, 90}` to modify images based on its magnitude of DCT basis and their freqequency basis. In a), we show the original image $x$ and the magnitude of its DCT basis $|\tilde{x}|$ in both linear and log scale. In b), we show images modified by removing DCT basis vectors whose magnitudes are in the bottom `threshold` percentage (row 1), the differences between the modified images and the original image (row 2), the binary mask used to remove the DCT basis: black means removed (row 3), images modified by removing high-frequency DCT basis vectors (row 4), the differences between the modified images and the original image (row 5) and the binary mask used to remove the DCT basis: black means removed (row 6). Notice that the masks in row 6 only depends on the dimension of the images, whereas the masks in row 3 differs from images to images.

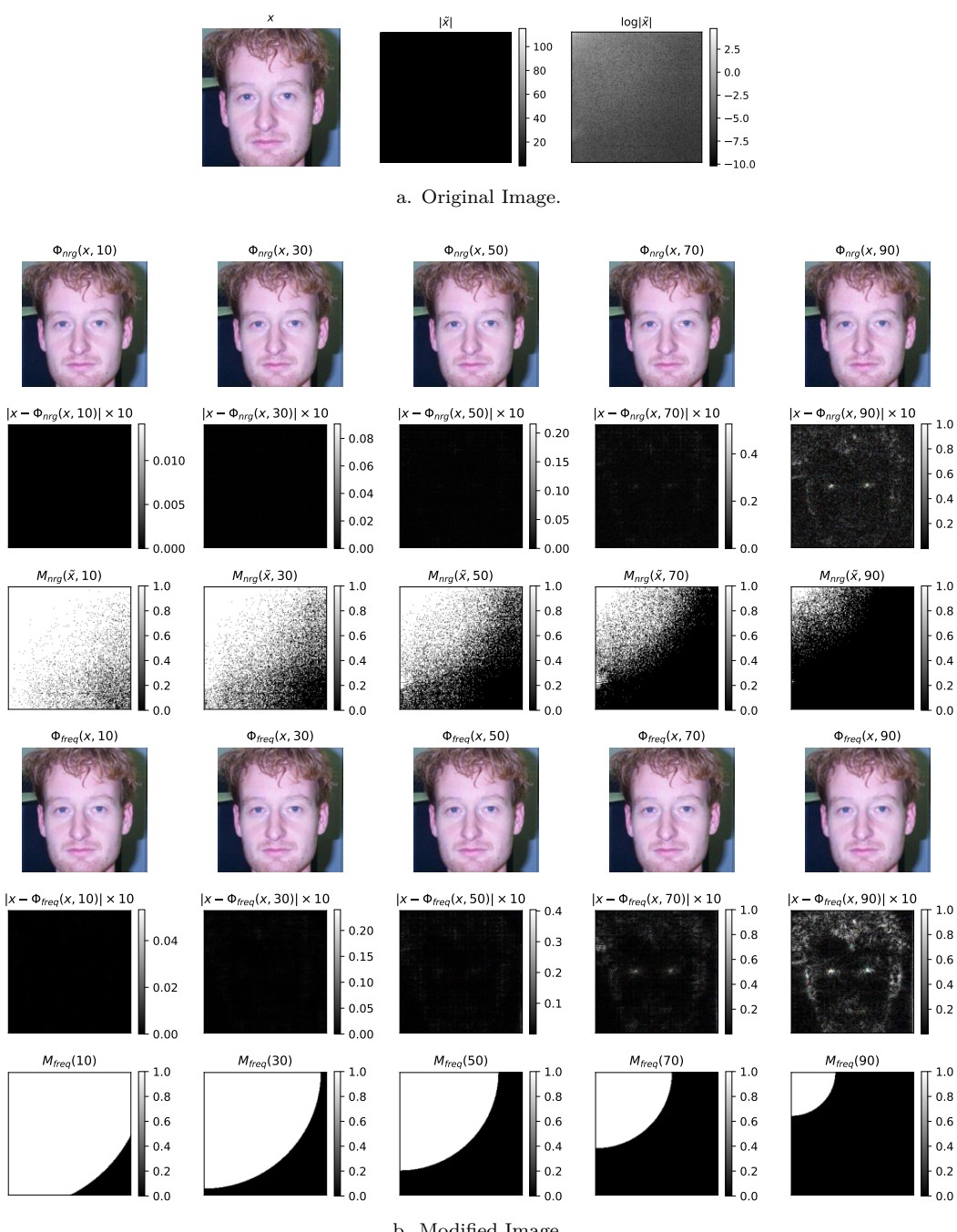

Figure 14: **Examples of modified images used in Observation I. (Caltech101)** We use a threshold value of `threshold` = $\{10, 30, 50, 70, 90\}$ to modify images based on its magnitude of DCT basis and their freqequency basis. In a), we show the original image $x$ and the magnitude of its DCT basis $|\tilde{x}|$ in both linear and log scale. In b), we show images modified by removing DCT basis vectors whose magnitudes are in the bottom `threshold` percentage (row 1), the differences between the modified images and the original image (row 2), the binary mask used to remove the DCT basis: black means removed (row 3), images modified by removing high-frequency DCT basis vectors (row 4), the differences between the modified images and the original image (row 5) and the binary mask used to remove the DCT basis: black means removed (row 6). Notice that the masks in row 6 only depends on the dimension of the images, whereas the masks in row 3 differs from images to images.

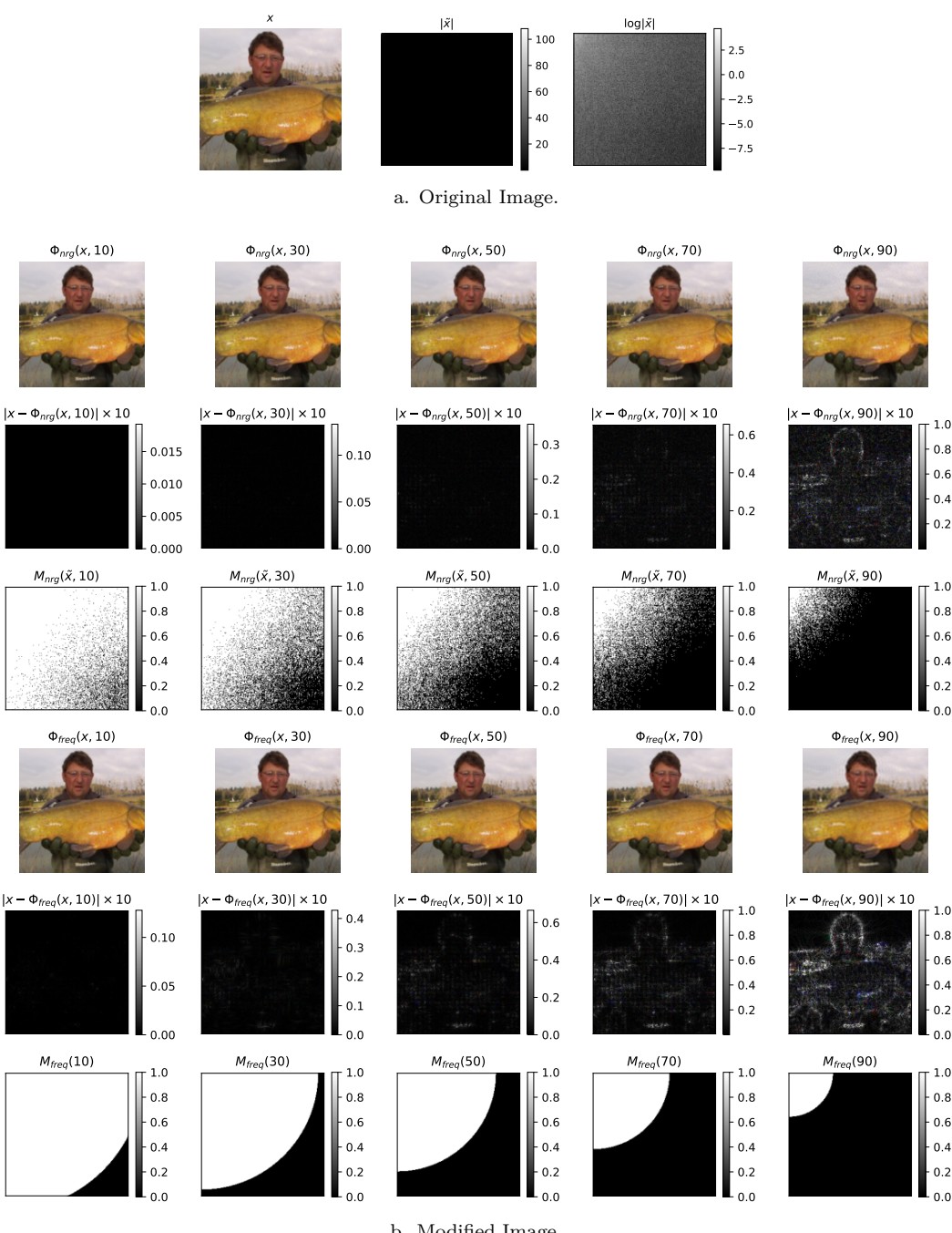

Figure 15: **Examples of modified images used in Observation I. (Imagenette)** We use a threshold value of $\texttt{threshold} = \{10, 30, 50, 70, 90\}$ to modify images based on its magnitude of DCT basis and their freqequency basis. In a), we show the original image $x$ and the magnitude of its DCT basis $|\tilde{x}|$ in both linear and log scale. In b), we show images modified by removing DCT basis vectors whose magnitudes are in the bottom $\texttt{threshold}$ percentage (row 1), the differences between the modified images and the original image (row 2), the binary mask used to remove the DCT basis: black means removed (row 3), images modified by removing high-frequency DCT basis vectors (row 4), the differences between the modified images and the original image (row 5) and the binary mask used to remove the DCT basis: black means removed (row 6). Notice that the masks in row 6 only depends on the dimension of the images, whereas the masks in row 3 differs from images to images.

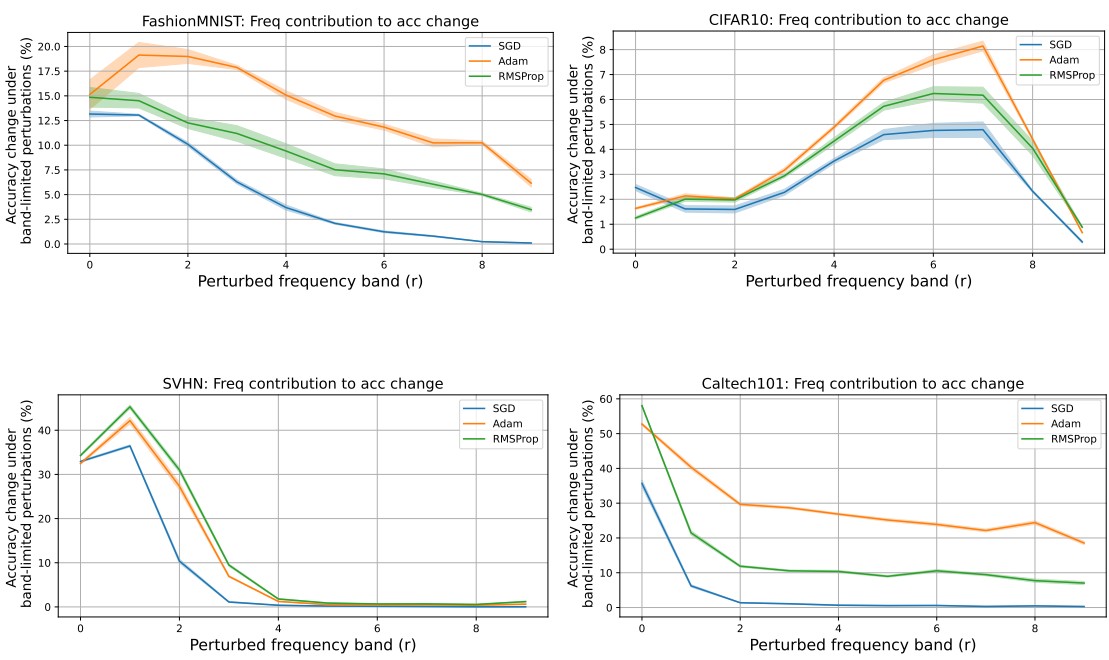

Figure 16: **The effect of band-limited Gaussian perturbations on the model (additional figures).** Perturbations from the lowest band, i.e., $\Delta x^{(0)}$, have a similar effect on all the models, despite being trained by different algorithms and exhibiting different robustness properties. On the other hand, models' responses vary significantly when the perturbation focuses on higher frequency bands.

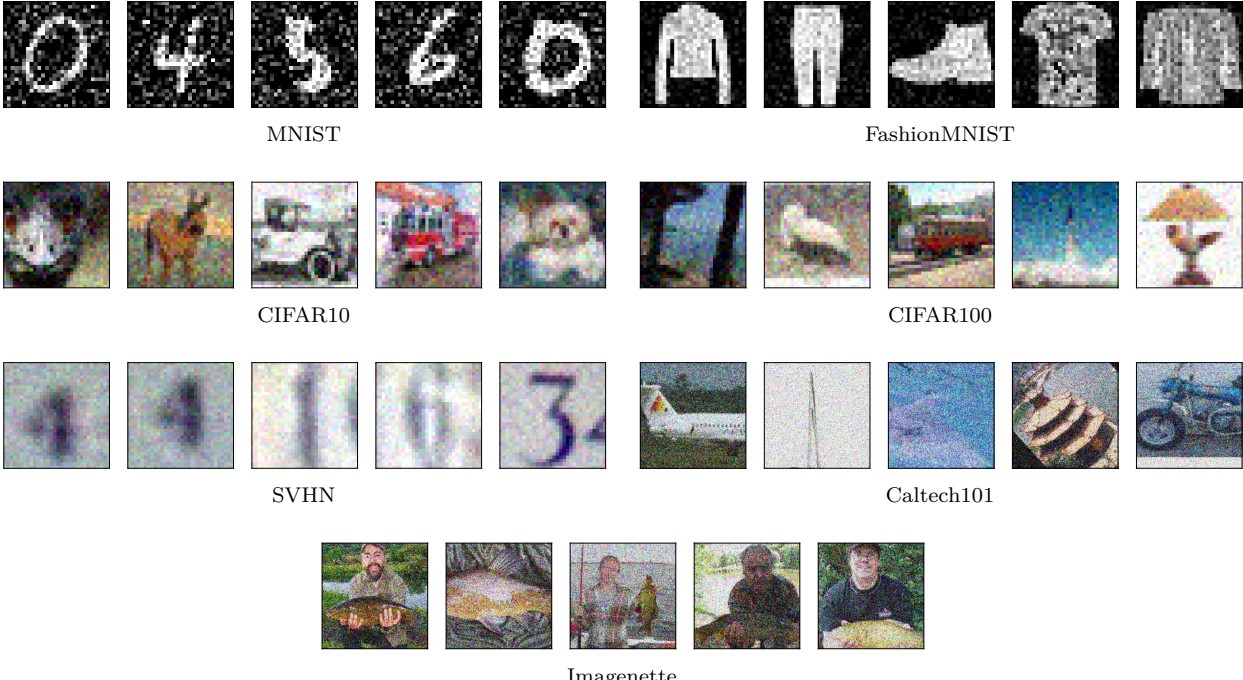

Figure 17: **Images perturbed by additive Gaussian white noise with different variance.** For each dataset, we select the largest variance value from Table 3.

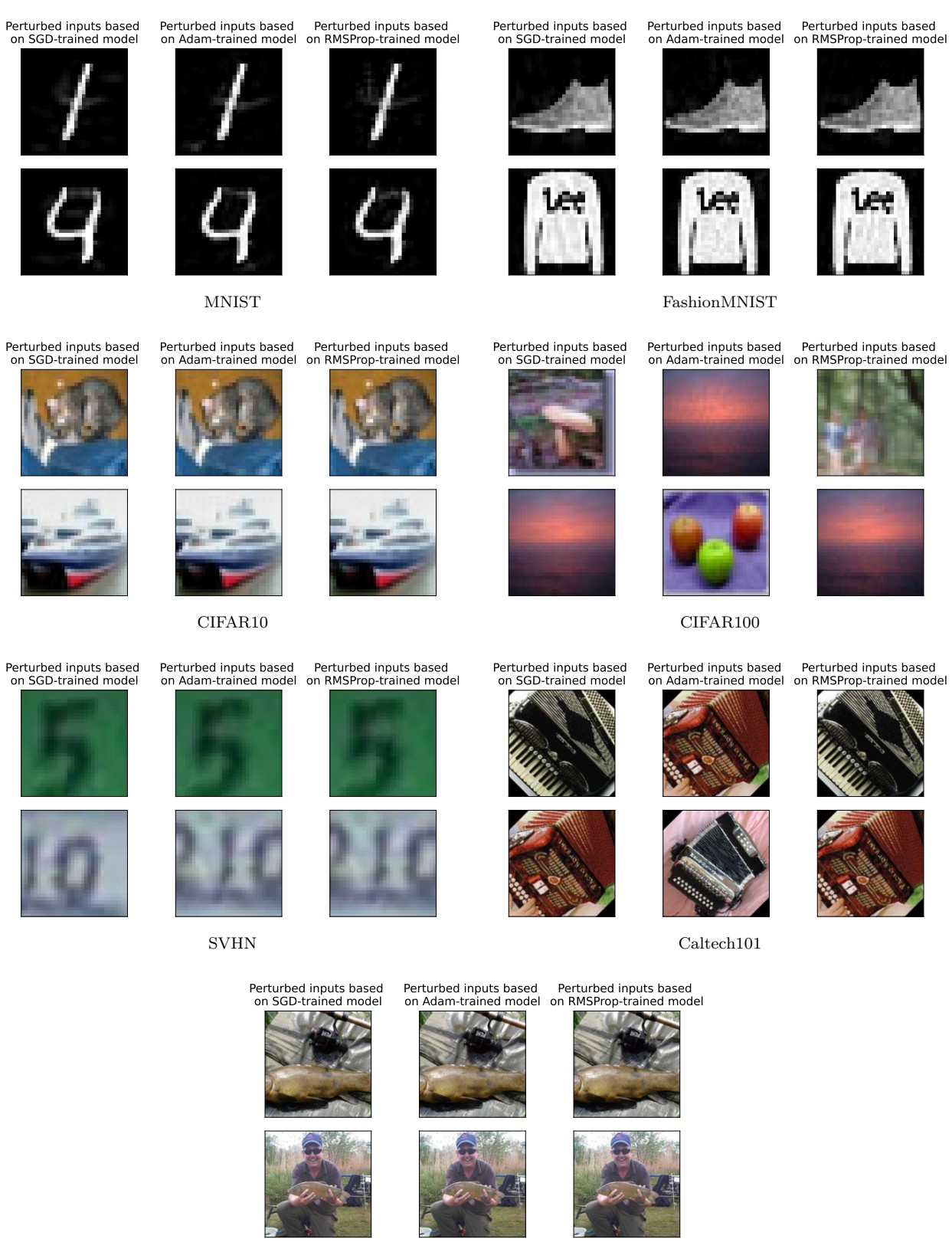

Figure 18: **Images perturbed by $\ell_2$-norm bounded adversarial perturbation (Croce & Hein, 2020).** We select the largest $\epsilon$ value from Table 3 to generate $\ell_2$ bounded perturbations for images in each dataset. We also compare perturbations generated using models trained by different algorithms.

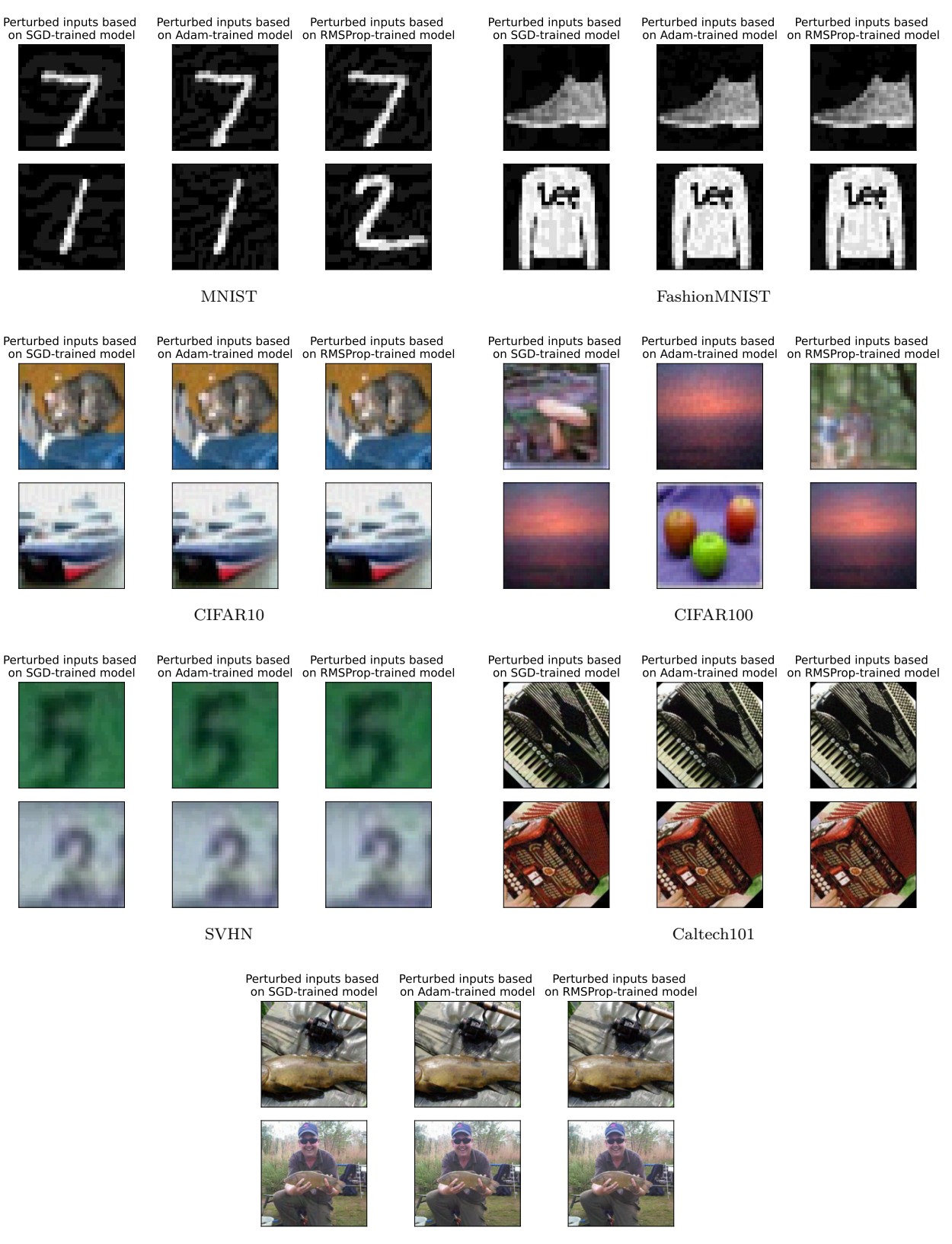

Figure 19: **Images perturbed by $\ell_\infty$-norm bounded adversarial perturbation (Croce & Hein, 2020).** We select the largest $\epsilon$ value from Table 3 to generate $\ell_\infty$ bounded perturbations for images in each dataset. We also compare perturbations generated using models trained by different algorithms.

