# OpenReview forum: "Understanding the robustness difference between stochastic gradient descent and adaptive gradient methods"
_TMLR — Accepted by TMLR_

### Review · Reviewer_rUpq · 2023-09-10

**Summary Of Contributions:**

This work studies how the frequency characteristics of the dataset affect the robustness of models trained by SGD and adaptive gradient methods. This work first observes that the robustness is related to irrelevant frequencies, which are mainly high frequencies of the data, and then shows that different learning algorithms have different mechanisms when handling these data frequencies. The experiments support the findings and also indicate that regularization can further improve performance.

**Audience:**

Yes

**Broader Impact Concerns:**

No such concerns.

**Claims And Evidence:**

Yes

**Requested Changes:**

1. Figure 5a looks strange to me at first. I thought the error for $\tilde{e}_0(t)$ and $\tilde{e}_1(t)$ should be decreasing, but it seems not. Later I understood that it was because $\tilde{w}_0(0)$ is smaller $\tilde{w}^*_0$, and $\tilde{w}_1(0)$ is smaller $\tilde{w}^*_1$. I suggest changing the y-axis to $|\tilde{e}_i(t)|$, where $i=0,1,2$ to avoid this confusion.
2. For the relationship between the weight norm and the robustness, I am not sure whether this result is novel or not. I only know that such regularization can avoid overfitting. Is the improvement of robustness related to the reduction of overfitting? Some clarification of how such discussion differs from existing knowledge is helpful.
3. For the last term of Equation 9, I thought it should be $<\Delta \tilde{x}, \tilde{w}-\tilde{w}^*>$ instead of $<\Delta \tilde{x}, \tilde{w}>$?
4. In this sentence, "Finally, in the deep learning setting, we showed that models trained by SGD have a noticeably larger Lipschitz constant than those trained by Adam and RMSProp", at the end of Section 6, I think it is a typo. SGD is shown to have a smaller Lipschitz constant as in Table 1, isn't it?

**Strengths And Weaknesses:**

Strengths:
1. This paper is very well-written. The logic of observations-> theory->experiments is very clear and attractive to readers.
2. The perspective of studying the robustness of algorithms by frequencies of data is interesting to me, although I am not very familiar with state-of-the-art research on robustness.

Weaknesses:
Some minor points are not very clear and may be incorrect. Please see the section of Requested Changes.

---

> ### Author Response · Authors · 2023-10-16
>
> We are thankful to the reviewer for the careful reading of the paper and the helpful comments. We are honored that the reviewer acknowledged the clarity and appeal of our paper. Below, we provide our detailed responses to the requested changes, and we will revise the paper accordingly.
>
> **Response to Requested Change 1:** We have enhanced the clarity of Figure 5a by updating the y-axis to represent $|\tilde{e}_i(t)|$. This modification aims to provide a more clear visualization of the error dynamics.
>
> **Response to Requested Change 2:**  Our response to the second request change is twofold. First, regarding the novelty of the result and its relation to existing work: Several previous works have studied the relationship between weight norms and model adversarial robustness. For instance, Galloway et al. empirically demonstrated the limitations of $\ell_2$ weight decay in countering adversarial examples [1], while Guo et al. explored the use of $\ell_0$ weight norms for enhancing robustness [2]. In contrast to these approaches that primarily focus on enhancing robustness, our paper takes a different perspective. Our goal is to provide insights into why SGD and adaptive gradient methods can yield models with different robustness, and we investigate how the frequency-domain characteristics of the data contribute to these differences in model robustness. To the best of our knowledge, this is a novel contribution to the field of studying neural network adversarial robustness.
>
> Second, to answer “Is the improvement of robustness related to the reduction of overfitting?”: Our work does not provide evidence on the relationship between robustness and overfitting. However, through theoretical analysis using synthetic dataset, we demonstrate that solutions obtained through GD and signGD yield near-zero generalization error, indicating that they do not exhibit overfitting. Nevertheless, the adversarial risks of those solutions are different.
>
> On CIFAR-100, we observe that training with SGD/Adam/RMSProp results in models with approximately 0 loss and 100% accuracy on the training data, but the model optimized under SGD has slightly lower test accuracy. From the perspective of overfitting, the SGD-trained models are actually more overfitted to the training set compared to the Adam/RMSProp-trained models, while the robustness gap is quite significant as shown in FIgure 1.
>
> **Response to Requested Change 3:** The complete derivation of Eq 9, which characterizes the adversarial risk under an $\ell_2$-norm bounded perturbation, is available in Appendix E.2. We emphasize that this derivation is based on the squared difference between the model output for the perturbed datapoint $<\tilde{X} + \Delta \tilde{x}, \tilde{w}>$ and the true output of the original input $<\tilde{X}, \tilde{w}^*>$.
>
> **Response to Requested Change 4:** It is indeed a typo and we have corrected it in the revised version.
>
> **Reference:**
>
> [1] Galloway et al, Adversarial Training Versus Weight Decay, https://arxiv.org/abs/1804.03308
>
> [2] Guo et al, Sparse DNNs with Improved Adversarial Robustness, NeurIPS 2018

---

> > ### Comment · Reviewer_rUpq · 2023-10-28
> >
> > Thank you for the response. My main concerns are addressed.

---

### Review · Reviewer_RKpr · 2023-09-24

**Summary Of Contributions:**

This paper aims to understand the phenomenon of models trained with different optimizers (SGD, Adam, RMSProp) exhibiting significant differences in robustness against various types of noise, including Gaussian noise and adversarial perturbations. To this end, the paper explores the connection between different optimizers and robustness through the lens of frequency representation. Specifically, the authors make the following claims: 1) There exist irrelevant frequencies in natural data; 2) the sensitivity along the irrelevant frequencies leads to differences in robustness; 3) SGD, and signGD as a special case of adaptive gradient methods (Adam and RMSProp), converge to different solutions with respect to different frequencies, hence leading to different levels of robustness. These claims are observed empirically, and (3) is analyzed theoretically based on a linear model. To further extend claim (3) from a linear model to deep neural networks (DNNs), the authors also investigate the relationship between the Lipschitz constant and robustness in DNNs. Overall, this paper provides a comprehensive insight into the connection between different optimizers, frequency representation, and weight norm in the context of robustness.

**Audience:**

Yes

**Claims And Evidence:**

Yes

**Requested Changes:**

1. Please reconsider the claim on contributions in Section 3: which claims are supported by existing work, and which claims are completely new discoveries.
2. Please justify why disabling BN layers or adding experiments with BN enabled.
3. Please justify the validity of using weight norm as a proxy for analyzing DNNs.
4. Please discuss the scalability of the theory in terms of AT and SAM.

**Strengths And Weaknesses:**

## Strengths

1. This paper is clearly written and well organized.
2. The problem addressed in this paper is interesting. To the best of my knowledge, the gap in robustness under different optimizers is somewhat overlooked in this area, and this paper may attract interested audiences.
3. The claims are supported by sufficient empirical and/or theoretical evidence. For empirical validation, the experiments are conducted on various datasets and illustrated with multiple figures.
4. Although the theoretical analysis is conducted on a simple linear model, the author also attempts to extend the theoretical evidence to DNNs.
5. The insight delivered by this paper is of significance because it can help the community better understand how optimizers affect robustness and further design better optimizers for improved robustness.

## Weaknesses

1. Exploring adversarial robustness with frequency is not a completely new idea [1,2]. In Section 3, the authors claim that there are irrelevant frequencies that exist in natural data, and the higher end of the frequency spectrum has a strong impact on robustness. However, some of the conclusions can already be drawn from [1,2]. Therefore, I suggest the authors narrow their contribution by clarifying which claims are supported by existing work, and which claims are completely new discoveries.
2. It is still a bit confusing why the authors disable batch normalization (BN) in experiments. In Appendix B, the authors claim that "the effect of BN on model robustness is still an active research topic, therefore it is important to remove such a factor from our analysis and focus on how the choice of optimization affects robustness". However, in my opinion, enabling BN is also fair for comparison among different settings. Furthermore, there are also many papers that understand different factors for adversarial robustness simply leaving BN enabled as default, including but not limited to [3,4].
3. The theoretical analysis is based on a linear **regression** model while this paper mainly discusses the robustness under **classification** task. I think there may exist some solution for bridging this gap.
4. Following 3: the extension from linear model to DNNs is based on weight norm as a proxy. The reason behind this, "the minimum norm solution is the most robust one. That is, a smaller weight norm implies better robustness. This suggests a connection between the weight norm and model robustness", shows a confusing relation between weight norm, irrelevant frequency representation, and robustness. Specifically, what is the main factor discussed in this paper, the frequency representation or weight norm? I don't think mixing everything together gives a clear explanation.
5. This paper only discusses the phenomenon under standard training. However, adversarial training (AT) [5], which has been considered as one of the most effective approaches to improve adversarial robustness of the model, also exhibits this property. In [6], it is shown that AT with SGD optimizer leads to better adversarial robustness than other optimizers. Can the theory in this paper also explain this phenomenon?
6. Recent work [7] also shows that Sharpness-Aware Minimization [8], which is an optimizer that regularizes the sharpness of weight loss landscape, can lead to significantly higher adversarial robustness than SGD optimizer. Can the theory in this paper also explain this phenomenon?

[1] High-frequency Component Helps Explain the Generalization of Convolutional Neural Networks. CVPR

[2] A Fourier perspective on model robustness in computer vision. NeurIPS

[3] Exploring Architectural Ingredients of Adversarially Robust Deep Neural Networks. NeurIPS

[4] Data Augmentation Can Improve Robustness. NeurIPS

[5] Towards deep learning models resistant to adversarial attacks. ICLR

[6] Bag of tricks for adversarial training. ICLR

[7] Sharpness-Aware Minimization Alone can Improve Adversarial Robustness. ICML Workshop

[8] Sharpness-Aware Minimization for Efficiently Improving Generalization. ICLR

---

> ### Author Response · Authors · 2023-10-16
>
> We are thankful to the reviewer for the careful reading of the paper and the helpful comments. We are honored that the reviewer acknowledged the clarity and novelty of our paper. Below, we provide our detailed responses to the requested changes, and we will revise the paper accordingly.
>
> **Response to Requested Change 1:** We appreciate the reviewer's suggestion to consider the relevance of [1,2] in our work and to distinguish our contribution from theirs. A clear distinction between our work and [1,2] is that they considered all features (frequencies) useful for generalization. Some features are robust because they align with human perception, while others are non-robust features and models become vulnerable to perturbations when their decision-making relies on such non-robust statistics in the dataset. Our work, on the other hand, emphasizes the presence of irrelevant information within natural datasets, which tends to be concentrated in the low-spectrum energy and high-frequency components of the data (Sec 3.1). In other words, not all features (frequencies) are useful for generalization, and the model’s different use of the irrelevant information in the data leads to differences in robustness. This, in turn, motivated the design of the synthetic dataset used in our subsequent linear analysis. In addition to focusing on frequency with high and low coefficients, our analysis also considered the spectrum energy of the dataset, which is an aspect absent in both [1,2]. This additional dimension provides a more comprehensive view of the irrelevant information in the data.
>
> **Response to Requested Change 2:** When training machine learning models, different design choices such as the usage of batch normalization (BN) or the type of augmentation can have different impacts on the robustness of the model. As such, our paper narrows the focus to one specific facet of the training process - the selection of optimizers. We aim to shed light on how this critical component influences the robustness of trained models. In the revised version of the paper, we have included additional results with BN enabled in Table 4 of Appendix C.1. In line with the outcomes presented in Table 3, the standard generalization performance of the models is similar, while the robustness difference between SGD and adaptive gradient methods still persists.
>
> **Response to Requested Change 3:** The main focus of the paper is to study the impact of optimization algorithms on model robustness when data distribution exhibits a particular characteristic. That is, it contains irrelevant information. We approach this problem through a frequency-domain perspective and demonstrate that such irrelevant information is predominantly located in the low-spectrum energy and high-frequency components of the data.
>
> Validity of using weight norm in analyzing DNN: In Eq 30, we demonstrate that given a  $\ell_p$ norm-constrained input perturbation, the output change of a single-layer ReLU-activated feedforward network can be bounded from above using the induced matrix norm of the weight. This allows us to extend the analysis developed from the linear analysis to DNN.
>
> **Response to Weakness 3:** The main reason we focused on linear regression is to create an environment where multiple standard risk minimizers exist, and one of the minimizers is robust to adversarial perturbations. While our primary focus has been on linear regression, there are indeed connections and implications that can be drawn for classification tasks. For example, both linear regression and linear classification are based on linear decision boundaries. Under GD, the gradient of both regression and classification is proportional to the magnitude of the input: $|\Delta w_i| \propto |x_i|$. As such, the insights gained from our study on robustness in linear regression can inform and benefit the understanding of similar principles in linear classification.

---

> ### Author Response · Authors · 2023-10-16
>
> **Response to Requested Change 4:** We agree with the reviewer that methods such as adversarial training (AT) and Sharpness-Aware Minimization(SAM) have shown effectiveness in improving the adversarial robustness of the model. However, we emphasize that the main focus of our paper is to understand the effect of optimizers on the robustness of models obtained in the standard training regime.
>
> In the theoretical analysis with linear models, we demonstrated that among many standard risk minimizers, there exists a unique robust standard risk minimizer $\tilde{w}^{R}$ (Eq. 8). Because the irrelevant frequencies in the data are under-constrained (Claim 3.1), neither SGD nor signGD is able to achieve such a robust standard risk minimizer.
>
> Relating our work to AT: During AT, the training data is augmented by introducing adversarial perturbations. These perturbations do not alter the true target of the input. Therefore, the AT procedure can be conceptualized as training with noise added to the irrelevant frequencies of the synthetic data.
>
> For simplicity, consider adding a centered Gaussian perturbation to the irrelevant frequencies. This means that there is only one standard risk minimizer: $|\tilde{\mathcal{W}}^*| = 1$, and it is a robust standard risk minimizer: $\tilde{w}^* = \tilde{w}^R$. Analyzing the GD dynamics described in Eq 15, we observe that weight adaptation now drives the weight associated with the irrelevant frequency toward zero. As a result, the asymptotic GD solution aligns with the robust standard risk minimizer.
>
> In Eq 22, we demonstrate that the asymptotic solution obtained through signGD converges to an $O(\eta)$ neighborhood of the standard risk minimizer. This implies that the asymptotic signGD solution along the irrelevant frequency oscillates around zero. This example, particularly within the context of linear models, sheds light on why AT with SGD can be a preferred approach in improving model robustness against adversarial perturbations.
>
> Relating our work to SAM: The SAM objective comprises two components: a sharpness-aware loss denoted as $L^{SAM}_{S}$ and an $\ell_2$ regularization term applied to the weight norm. With the problem setup described in Sec. 4.1, the sharpness-aware loss has a closed form of $\ell(x,w) + \rho^2 ||x||_q^2$ where $\frac{1}{p} + \frac{1}{q} = 1$, resulting in parameter gradients identical to the one from the squared loss. This shows that training with the sharpness-aware loss is ineffective in improving the adversarial robustness of the linear model. However, in Section 4, we demonstrate that a minimum norm standard risk minimizer proves to be the most robust standard risk minimizer. Consequently, the inclusion of the $\ell_2$ regularization term on the weight norm in SAM can indeed enhance the model's adversarial robustness.

---

> > ### Comment · Reviewer_RKpr · 2023-10-16
> > **Thanks for your response!**
> >
> > Dear authors,
> >
> > Thanks for your response. Most of my concerns are adequately addressed. Also, I acknowledge that this paper mainly focuses on the robustness of models trained in standard (non-adversarial) methods. Therefore, I agree that this paper does not necessarily discuss adversarial training (AT).
> >
> > However, as you also noted on page 4
> > >  Since SGD, Adam, and RMSProp have been the go-to optimizer in both academic and
> > industrial settings, this motivates us to understand the robustness of models trained by those algorithms
> > and built on the standard training pipelines, i.e., minimizing some losses on the original training set
> >
> > since SAM is also an adaptive gradient method that is designed for standard training (natural generalization) and deployable for industrial settings, in my opinion, it's valuable to discuss its robustness. I'm also curious how your understanding connects to the interpretation proposed in [7].
> >
> > Please revise your paper through this aspect.
> >
> > Best,

---

### Review · Reviewer_x5rV · 2023-10-05

**Summary Of Contributions:**

This paper presents a thorough study of the generalization and robustness performance of various optimizers, e.g., SGD and the adaptive ones (mainly RMSProp and Adam). The main finding is that though those aforementioned optimizers perform similarly on generalization, their robustness performance varies a lot. The main claim is that such a finding is due to the irrelevant frequencies in natural datasets (mainly image datasets). Perturbations on those irrelevant frequencies affect different optimizers in different ways. Analysis was conducted on linear models and empirical results were demonstrated on both synthetic and real datasets to justify the claim and existence of the irrelevant frequencies.

**Audience:**

Yes

**Broader Impact Concerns:**

I don't think there is any concern about the ethical implications of the work.

**Claims And Evidence:**

Yes

**Requested Changes:**

- Adding experiments on model architectures beyond CNNs, e.g., ViT and stable diffusion.
- Extend the current experiment to text data.
- Adding discussion on the robustness and model convergence.
- Add SGD with momentum as an additional optimization method.

**Strengths And Weaknesses:**

Strengths:
- The paper is well-written and well-motivated.
- Studying the performance of various optimization methods is important.
- The proposed existence of irrelevant frequencies is promising.

Weaknesses
- It is not clear if the results of this paper generalize to various neural network architectures, i.e., only convolutional neural networks (CNNs) have been studied. What about the performance on ViTs or even Diffusion models?
- I am highly curious if a similar effect/observation can be observed for data with other modalities, e.g., language, audio, and graphs.
- The results suggest that SGD is more robust than the adaptive optimization methods, but those adaptive methods generally lead to faster model convergence. I wonder if there is any optimization method that enjoys both properties.
- How do SGD with momentum perform on the experimented tasks?

---

> ### Author Response · Authors · 2023-10-16
>
> We are thankful to the reviewer for the careful reading of the paper and the helpful comments. We are honored that the reviewer acknowledged the clarity and the motivation of our work. Below, we provide our detailed responses to the requested changes, and we will revise the paper accordingly.
>
> **Response to Requested Change 1:** We primarily focused on CNN-based architectures in our study, given their widespread usage in training image classification models. We acknowledge the recent success of Vision Transformers (ViT) [1] and appreciate the reviewer's suggestion to expand our research to include other neural network architectures.
>
> It's important to note that the dataset utilized in our paper is significantly smaller in size compared to larger datasets such as Imagenet-1k, Imagenet-21k, and JFT-300M. Recent research, such as the work by Zhu et al. [2], has shown that Vision Transformers tend to generalize poorly on small datasets when trained from scratch. In particular, Zhu et al. empirically demonstrated that ViT and ResNet learn distinct representations on small datasets while converging to similar representations on larger datasets.
>
> Therefore, we perform fine-tuning based on a pre-trained ViT-b/16 [1]. Among the datasets we considered, Imagenette is a 10-class subset of the Imagenet-1k dataset, making it especially suitable for the fine-tuning task, since the publicly available ViT checkpoint was pre-trained on Imagenet-1k. Also, it's important to note that the pre-trained models were originally trained using Adam. In our fine-tuning process, we treated ViT as a feature extractor (i.e., no weight update on the ViT), with a focus on fine-tuning the fully-connected layer. Our approach follows prior research [3] and incorporates the three different optimizers, each fine-tuned for 10 epochs. We initiated the fine-tuning process with an initial learning rate of 0.01, followed by a cosine decay learning rate schedule and a linear warmup. Throughout this process, we maintained a fixed batch size of 512.
>
> To evaluate the robustness of the fine-tuned models, we maintained the exact same perturbation strengths, including the variance of Gaussian noise and $\epsilon$ for adversarial perturbations, as used in Table 2. The results can be found in Table 5 in Appendix C.2. We draw three observations.
>
> First, all models fine-tuned with the three different optimizers achieved near 100% test accuracy, a substantial improvement from the 89% accuracy when training from scratch using PreActResNet18. This significant boost in standard generalization highlights the effectiveness of fine-tuning with ViT. Second, we observed that the fine-tuned models exhibited a notable increase in robustness to Gaussian noise. However, they showed a high vulnerability to adversarial perturbations. This observation aligns with the existing literature, where a trade-off is often present between standard accuracy and adversarial robustness. Finally, we make a similar observation on the robustness difference between models fine-tuned with the three optimizers, where models fine-tuned with SGD exhibited greater robustness to both Gaussian noise and adversarial perturbations when compared to models fine-tuned using Adam and RMSProp.
>
> Lastly, diffusion models are primarily designed for generative tasks, making them unsuitable for our specific scope of work.
>
> **Response to Requested Change 2:** We appreciate the reviewer’s high interest in exploring whether similar observations can be extended to other data modalities. In our study, we focused on the vision domain, as it is suitable for a frequency-domain analysis that allows us to gain insights into how optimizers behave in the presence of irrelevant frequencies in the dataset. Modalities such as language and graphs lack clear frequency interpretations. While there may be some degree of irrelevance in language and graph datasets, it is likely not in the same frequency-domain manner as in the vision domain.
>
> However, audio signals do offer a frequency-based interpretation. As such, we have included additional results in Table 6 in Appendix C.3, which compare the standard generalization and robustness properties of an audio classifier trained on the SpeechCommand dataset [4]. All models are trained for 200 epochs, with an initial learning rate of 0.1 and learning rate decay at epochs 100 and 150. We considered the accuracy of models under Gaussian- and adversarially-perturbed test set. Manual verification was conducted to ensure that the noisy audio phrase could still be recognizable.
>
> Results demonstrate that despite similar test accuracy, the models trained using SGD exhibit greater robustness when compared to the other two optimization methods. These insights provide valuable context to the generalizability of our initial observations, offering a more comprehensive understanding of how optimizers perform in the context of different data modalities.

---

> ### Author Response · Authors · 2023-10-16
>
> **Response to Requested Change 3:** Achieving a balance between faster convergence and model robustness is a promising avenue for future research. The theoretical analysis with linear models offers two distinct perspectives to approach this challenge: initialization and regularization.
>
> Eq 17 shows that the asymptotic solution obtained through GD is not the most robust risk minimizer. This is because the weights associated with irrelevant frequencies do not receive updates. A straightforward approach to enhance the robustness of this linear model is to initialize the weights associated with these irrelevant frequencies to 0.
>
> For the SignGD solution, our analysis shows that $|\Delta \tilde{w}_2|$ grows until $\Delta \tilde{w}_0$ begins to oscillate. This observation suggests that regularization could play a crucial role in improving the robustness of the linear model. By penalizing the weights associated with these frequencies, one can potentially achieve a more robust solution. Introducing regularization into the optimization procedure to improve robustness has been explored in [5, 6, 7].
>
> **Response to Requested Change 4:** We appreciate the reviewer's interest in the results of models trained using SGD with momentum, and we have included additional results in Table 7 of Appendix C.4. Similarly, we maintained the exact same optimization configuration as that used for generating the SGD results presented in Table 2, and the only variation is an additional momentum term with a coefficient of $\beta = 0.9$. The result shows that models optimized by both vanilla SGD and SGD with momentum (SGD-m) exhibit similar trends in terms of generalization and robustness.
>
> **Reference:**
>
> [1] Dosovitskiy et al, An Image is Worth 16x16 Words: Transformers for Image Recognition at Scale, ICLR 2021
>
> [2] Zhu et al. Understanding Why ViT Trains Badly on Small Datasets: An Intuitive Perspective
>
> [3] Steiner et al, How to train your ViT? Data, Augmentation, and Regularization in Vision Transformers, TMLR
>
> [4] Pete Warden, Speech commands: A dataset for limited-vocabulary speech recognition
>
> [5] Simon-Gabriel et al, First-order adversarial vulnerability of neural networks and input dimension. ICML2019
>
> [6] Wen et al, Towards understanding the regularization of adversarial robustness on neural networks. ICML 2020
>
> [7] Ma et al, SOAR: Second-order adversarial regularization

---

> > ### Comment · Reviewer_x5rV · 2023-11-04
> >
> > Thank you for the response, my major concerns have been addressed.

---

### Author Response · Authors · 2023-10-18
**Revision uploaded**

We would like to thank all reviewers for their helpful feedback. We have revised the submission based on the first round of reviews. In particular, results and discussions on the additional experiments are included in C.1, C.2, C.3, and C.4 of Appendix.

---

### Decision · Action_Editor_dSWj · 2023-11-04

**Recommendation:** Accept as is

**Comment:**

This paper studies the robustness of neural networks under different training algorithms, including stochastic gradient descent (SGD) and adaptive gradient methods, followed by an in-depth analysis of gradient descent (GD) and sign gradient descent (signGD) methods. The results offer new theoretical insights into understanding the effect of optimization algorithms on the robustness of neural networks. All reviewers recommend acceptance for this paper, and I concur.

I also recommend Featured Certification for this submission, as the results provide new insights into understanding the fundamentals of robustness and gradient-based training methods.

**Audience:**

Understanding the effect of optimization algorithms for model training is of broad interest to this community.

**Claims And Evidence:**

This paper provides sound theoretical analysis and sufficient empirical evaluations to support the claims.